# Relational In-Context Learning via Synthetic Pre-training with Structural Prior

**Yanbo Wang**[1]  **Jiaxuan You**[2]  **Chuan Shi**[3]  **Muhan Zhang**[1 4]

## Abstract

Relational Databases (RDBs) are the backbone of modern business, yet they lack foundation models comparable to those in text or vision. A key obstacle is that high-quality RDBs are private, scarce, and structurally heterogeneous, making internet-scale pre-training infeasible. To overcome this data scarcity, we introduce **RDB-PFN**, the first relational foundation model trained purely via **synthetic data**. Inspired by Prior-Data Fitted Networks (PFNs), where synthetic data generated from Structural Causal Models (SCMs) enables reasoning on single tables, we design a **Relational Prior Generator** to create an infinite stream of diverse RDBs from scratch. Pre-training on **over 2 million** synthetic single-table and relational tasks, RDB-PFN learns to adapt to any new database instantly via genuine **in-context learning**. Experiments show that RDB-PFN achieves strong few-shot performance on 19 real-world relational prediction tasks, outperforming state-of-the-art tabular foundation models evaluated on the same DFS-linearized inputs, while using a lightweight architecture and fast inference. The code is available at https://github.com/MuLabPKU/RDBPFN.

## 1. Introduction

**The Foundation Model Discrepancy.** Relational Databases (RDBs) serve as the bedrock of modern enterprise, storing the vast majority of the world's high-value structured data (Codd, 1970; 2007; Harrington, 2016). Yet, a stark discrepancy defines the current AI landscape: while Foundation Models (FMs) have revolutionized unstructured modalities like text and vision through massive scale (Brown et al., 2020; Achiam et al., 2023; Dosovitskiy, 2020), RDBs

remain largely untouched by this paradigm shift. In the RDB domain, the standard workflow still relies on bespoke feature engineering followed by single-table methods like Gradient Boosted Decision Trees (GBDTs) (Chen, 2016; Prokhorenkova et al., 2018; Ke et al., 2017), or task-specific architecture search using Graph Neural Networks (GNNs) (Wang et al., 2024; Robinson et al., 2024).

**The Data Wall.** The primary bottleneck preventing foundation models for RDBs is not architectural, but data-centric. Foundation models in NLP rely on the "Scaling Law", the emergence of reasoning capabilities from massive ingestion of public real-world data (Kaplan et al., 2020). This approach fails for RDB because high-quality databases are inherently private, scarce, and structurally heterogeneous (Wang et al., 2024), making the standard pre-training methodology infeasible. Recent attempts on RDB foundation models, such as Griffin (Wang et al., 2025) and RT (Ranjan et al., 2025), rely on limited open-sourced real data, which is far from the pre-training scale. Consequently, they fail to achieve universal generalization without fine-tuning.

**The PFN Insight: Learning from Synthetic Data.** To circumvent this fundamental data scarcity, we turn to the emerging paradigm of **Prior-Data Fitted Networks (PFNs)** (Müller et al., 2021). PFNs leverage a counter-intuitive strategy: instead of training on scarce real-world data, they learn to approximate Bayesian Inference by pre-training on a vast corpus of synthetic tasks generated from Structural Causal Models (SCM). The core mechanism relies on the Transformer's ability to act as a "learning algorithm": by attending to the context of synthetic examples, the model learns to simulate the posterior predictive distribution. When the synthetic prior covers statistical patterns that are relevant to real tasks, the model can transfer this learned inference behavior to new datasets. This methodology has revolutionized the single-table domain: **TabPFN** (Hollmann et al., 2022) demonstrated that a Transformer trained purely on synthetic priors could outperform tuned GBDTs on small datasets. With GPU acceleration and In-Context Learning (ICL) inference, TabPFN-style methods can substantially reduce time cost compared with iterative training. Subsequent work continues to scale and refine this paradigm, shifting the core challenge from intractable data collection to solvable prior design, enabling models to generalize to entirely new schemas without a single gradient update.

[1]Institute for Artificial Intelligence, Peking University [2]University of Illinois at Urbana-Champaign [3]Institute of Computing Technology, Beijing University of Post and Telecommunication [4]State Key Laboratory of General Artificial Intelligence. Correspondence to: Muhan Zhang <muhan@pku.edu.cn>.

*Proceedings of the 43rd International Conference on Machine Learning*, Seoul, South Korea. PMLR 306, 2026. Copyright 2026 by the author(s).

**The Relational Gap.** However, the success of PFNs has essentially been confined to the single-table setting, where the prior typically assumes rows are Independent and Identically Distributed. This assumption violates the core of relational data. As highlighted by RDB benchmarks like 4DBInfer (Wang et al., 2024) and Relbench (Robinson et al., 2024), RDBs are defined by *interconnectivity*: a label in a "User" table is not merely a function of user attributes, but is often a complex aggregation of historical records in "Order" or "Click" tables. Applying standard single-table generators to RDBs fails to model these interactions. The field lacks a **Relational Prior**: a generative framework capable of synthesizing valid schema topologies, foreign key dependencies, and causal aggregations from scratch.

**Our Approach: RDB-PFN.** In this work, we introduce RDB-PFN, the first RDB foundation model built on a relational prior and pretrained purely on synthetic data. We formalize a novel generative mechanism that samples infinite streams of random schema topologies and propagates causal signals within and across tables. We prove a universality result under acyclic-schema, local Markov, and conditional-exchangeability assumptions. Based on synthetic RDBs sampled from our prior, we employ a rigorous curriculum, pre-training the model on over 2 million synthetic tasks. To isolate the importance of our relational prior, we pair this sophisticated data generation with a simple "Linearize-and-Attend" architecture, using standard Deep Feature Synthesis (DFS) (Kanter & Veeramachaneni, 2015) and a vanilla Transformer. This demonstrates that the model's relational intelligence stems from the prior's design rather than architectural complexity.

We extensively evaluated RDB-PFN on 19 real-world relational learning tasks. The results confirm that our approach delivers superior performance and efficiency: RDB-PFN outperforms both traditional GBDT baselines and general tabular foundation models in the few-shot regime, while requiring significantly fewer **parameters**, faster **inference speeds**, and orders of magnitude less **pre-training data**.

Our core contributions are summarized as follows:

- **Solving Scarcity via Synthetic Data:** Unlike traditional relational learning methods that require expensive fine-tuning on target tasks or pre-training on sensitive real-world data, RDB-PFN is trained **purely on synthetic data**. It performs **zero-gradient** inference strictly via ICL, effectively eliminating the dependency on large real-world datasets for both pre-training and adaptation.
- **Prior > Scale:** When compared to state-of-the-art Single-Table Foundation Models (augmented with DFS), RDB-PFN exceeds their performance despite using a fraction of the model size and training compute. This confirms that pre-training on a physically consistent **Relational Prior** equips the model with a structural inductive bias

that **cannot be efficiently replicated simply by scaling up general single-table data**.

## 2. Related Work

### 2.1. From Tabular to Relational Learning

Prior to the Foundation Model era, standard approaches required training models from scratch for each specific dataset.

**Single-Table Baselines.** Research on single table data has evolved through various approaches. Traditional methods, such as XGBoost (Chen, 2016), LightGBM (Ke et al., 2017), and CatBoost (Prokhorenkova et al., 2018), have been widely adopted due to their scalability. More recently, transformer-based methods like TabTransformer (Huang et al., 2020), TabNet (Arik & Pfister, 2021), FT-Transformer (Gorishniy et al., 2021), and SAINT (Somepalli et al., 2021) have leveraged attention mechanisms to capture complex relationships. Additionally, graph-based methods such as GRAPE (You et al., 2020), TabularNet (Du et al., 2021), TabGNN (Guo et al., 2021), and CARTE (Kim et al., 2024) represent tabular data as graphs to model interactions more effectively. Recent advancements have expanded beyond general architecture search to focus on refining specific components of the tabular modeling pipeline. Significant progress has been made in numerical encoding strategies (Gorishniy et al., 2022; Yarullin & Isaev, 2023), retrieval-augmented modeling that incorporates nearest-neighbor context (e.g., TabR (Gorishniy et al., 2023), ModernNCA (Ye et al., 2025)), and robust training protocols such as default-pretuning (RealMLP (Holzmüller et al., 2024)) or efficient ensembling (TabM (Gorishniy et al., 2024)). However, despite these innovations, they inherently assume a linearized feature vector and lack native mechanisms to capture the complex, multi-table topology of relational databases.

**The Relational Bridge: RDB Models.** RDBs extend the concept of single-table models by incorporating multiple interrelated tables, requiring models to capture both intra- and inter-table relationships. Early approaches, such as DFS (Kanter & Veeramachaneni, 2015) and RDB2Graph (Cvitkovic, 2020), attempted to flatten RDBs into a single table or apply GNNs to model relationships. Other works, like ATJ-Net (Bai et al., 2021) and KEN (Cvetkov-Iliev et al., 2023), use hypergraphs and knowledge graphs to model inter-table dependencies, while GFS (Zhang et al., 2023) integrates differentiable single-table models as embedding functions to preserve table structures. Some methods convert structured data into unstructured embeddings while retaining structural information (Grover & Leskovec, 2016), such as EmbDi (Cappuzzo et al., 2020) and RDF2Vec (Ristoski & Paulheim, 2016).

As RDB tasks have attracted increasing attention (Fey et al.,

2024), more comprehensive benchmarks and toolboxes have emerged. For example, 4DBInfer (Wang et al., 2024), Rel-Bench (Robinson et al., 2024; Fey et al., 2023), and Py-torchFrame (Hu et al., 2024) propose complete pipelines for converting RDBs into graph structures for GNN-based models. More recent efforts, such as ContextGNN (Yuan et al., 2024), RelGNN (Chen et al., 2025) and RelGT (Dwivedi et al., 2025), aim to design more expressive GNN architectures specifically for relational data. However, these models are still limited on individual RDB tasks.

## 2.2. Foundation Models for Structured Data

While language and vision fields have achieved scalability through massive real-world data ingestion, the structured data domain is still in search of a unified paradigm.

**Single-Table Landscape: Real vs. Synthetic.** Early attempts at tabular foundation models relied on large-scale real-world corpora, employing supervised or masked self-supervised learning (Zhu et al., 2023; Wang & Sun, 2022; Yang et al., 2023; Kim et al., 2024). However, these efforts struggled to generalize due to the lack of shared semantics across diverse tables. A paradigm shift occurred with Prior-Data Fitted Networks (PFNs) (Müller et al., 2021). TabPFN (Hollmann et al., 2022; 2025) demonstrated that training on synthetic datasets generated from Structural Causal Models allows a model to approximate Bayesian Inference, achieving state-of-the-art few-shot performance without observing real-world data—a result verified by numerous follow-up studies (Grinsztajn et al., 2025; Qu et al., 2025; Zhang et al., 2025a;b). This synthetic paradigm is now rapidly expanding to other domains, including time-series (Taga et al., 2025; Dooley et al., 2023; Hoo et al., 2025), causal discovery (Robertson et al., 2025; Balazadeh et al., 2025; Ma et al., 2025; Mahajan et al., 2024), and graph learning (Eremeev et al., 2025b;a; Hayler et al., 2025), among others.

**The Relational Gap.** Despite the success of PFNs in the single-table domain, relational modeling remains tied to the traditional Pre-training & Fine-tuning paradigm. Recent efforts like Griffin (Wang et al., 2025) and RT (Ranjan et al., 2025) rely on aggregating limited real-world repositories, employing text unification strategies and incorporating auxiliary tables. While valuable, these methods are fundamentally constrained by the scarcity of public schemas where they focus gradient-based fine-tuning to adapt to new tasks. To date, the field lacks a genuine foundation model.

## 2.3. Generative Models for Relational Databases

Existing research on RDB generation focuses mainly on *Privacy-Preserving Data Publishing*. The primary objective is to train a model on a specific private database ($D_{real}$) to produce a high-fidelity synthetic replica ($D_{syn}$).

**From Statistics to Powerful Generation Models.** Early approaches like SDV (Patki et al., 2016) and RC-TGAN (Gu-eye et al., 2023) relied on statistical copulas or GANs to clone parent-child relationships. Recently, the state-of-the-art has shifted toward **Graph-Conditional Diffusion**. Models such as ClavaDDPM (Pang et al., 2024), RelDiff (Hu-dovernik et al., 2025), and GRDM (Ketata et al., 2025) achieve superior fidelity by treating the database as a joint distribution and utilizing GNNs to iteratively denoise it.

**Incompatibility with Foundation Models.** However, these methods are not suitable for pre-training Foundation Models. First, as *Conditional Generators*, they depend on real input data, creating a circular dependency that fails to address the fundamental scarcity of RDBs. Second, their iterative sampling processes (e.g., diffusion denoising) are prohibitively slow for the high-throughput generation required to train on millions of tasks. To overcome this, we require an **Unconditional Relational Prior** capable of generating complex RDBs purely from scratch at scale.

**Positioning.** RDB-PFN is positioned differently from both supervised relational models and privacy-oriented relational data generators. Unlike GNN-based relational architectures or relational pre-training methods that rely on real databases and task-specific fine-tuning, RDB-PFN is trained primarily on synthetic relational databases sampled from an unconditional relational prior and adapts to new tasks through in-context learning. Unlike privacy-focused relational data generators, which are typically trained conditionally to mimic a particular private database, our generator is designed to produce diverse relational prediction tasks at scale for foundation-model pre-training.

## 3. Preliminaries

**Definition 3.1** (Relational Database: Schema and Instance). A RDB is specified by a **schema** and a **populated instance**.

**Schema.** The schema is a collection of tables $\mathcal{T} = \{T_1, \ldots, T_N\}$. Each table $T$ has a set of columns $\mathcal{C}(T) = \mathcal{K}^{pk}(T) \cup \mathcal{K}^{fk}(T) \cup \mathcal{A}(T)$, where $\mathcal{K}^{pk}(T)$ are **primary key (PK)** columns, $\mathcal{K}^{fk}(T)$ are **foreign key (FK)** columns, and $\mathcal{A}(T)$ are all remaining **feature columns**. Each FK column $k \in \mathcal{K}^{fk}(T)$ references a *parent table* $\mathrm{Ref}(k) \in \mathcal{T}$.

**Instance.** A database instance $\mathcal{D}$ assigns concrete rows to each table. Let $\mathcal{V}(T)$ denote the set of rows in table $T$; each row is a record containing values for all columns in $\mathcal{C}(T)$. PK values uniquely identify rows within a table, and FK values must match PK values in the referenced parent table.

**Definition 3.2** (Source vs. Dependent Tables). A table $T$ is a **source table** if it has no foreign keys, i.e., $\mathcal{K}^{fk}(T) = \emptyset$. A table $T$ is a **dependent table** if it has one or more foreign keys, i.e., $\mathcal{K}^{fk}(T) \neq \emptyset$.

**Definition 3.3** (Schema Graph and Instance Graph). We

use two levels of topology.

**Schema graph.** The schema graph is a directed graph $G_S = (\mathcal{V}_S, \mathcal{E}_S)$ with nodes $\mathcal{V}_S = \mathcal{T}$. We orient edges as **parent → child**:

$$(T_p \to T_c) \in \mathcal{E}_S \iff \exists k \in \mathcal{K}^{fk}(T_c) \text{ s.t. } \text{Ref}(k) = T_p.$$

**Instance graph.** The instance graph is a directed graph $G_{in} = (\mathcal{V}_{in}, \mathcal{E}_{in})$ whose nodes are **rows**: $\mathcal{V}_{in} = \bigcup_{T \in \mathcal{T}} \mathcal{V}(T)$. For a row $u \in \mathcal{V}(T)$, let $\text{pk}(u)$ denote its (possibly composite) primary-key value. For a foreign-key column $k \in \mathcal{K}^{fk}(T_c)$ and a child row $v \in \mathcal{V}(T_c)$, let $\text{fk}_k(v)$ denote the value stored in column $k$ of row $v$. We orient edges consistently as **parent row → child row**:

$$(u \to v) \in \mathcal{E}_{in} \iff v \in \mathcal{V}(T_c), \ u \in \mathcal{V}(T_p),$$
$$\exists k \in \mathcal{K}^{fk}(T_c) \text{ s.t. } \text{Ref}(k) = T_p \wedge \text{fk}_k(v) = \text{pk}(u).$$

We optionally associate each row-node $r$ in the instance graph $G_{in}$ with a latent structural state $Z_r \in \mathbb{R}^d$, and denote the collection by $\mathcal{Z} := \{Z_r\}_{r \in \mathcal{V}_{in}}$.

**Definition 3.4** (Relation Types and Neighborhoods). Each FK type induces a **relation type** $\tau$ (e.g., $T_p \to T_c$), and we also consider its reverse relation $\tau^{-1}$ to enable bidirectional message passing. For a row-node $v \in \mathcal{V}_{in}$, let $\mathcal{R}(v)$ be the set of relation types incident to $v$. For a relation type $\tau$, let $\mathcal{N}_\tau(v)$ denote the neighbor set of $v$ under $\tau$ (parents for $\tau$ pointing into $v$, children for $\tau$ pointing out of $v$, depending on the chosen direction).

**Definition 3.5** (Relational Prediction Task and In-Context Learning). A relational prediction task selects a **target table** $T_\star$ and a **target column** $y \in \mathcal{A}(T_\star)$ (could be computed via SQL Query). Each target row $v \in \mathcal{V}(T_\star)$ is mapped to a fixed-length feature vector $x_v$ by a deterministic linearization operator (e.g., DFS) applied to the database instance:

$$X = \text{Linearize}(\mathcal{D}, T_\star) \in \mathbb{R}^{|\mathcal{V}(T_\star)| \times p},$$

where each row of $X$ corresponds to one $v \in \mathcal{V}(T_\star)$ and $p$ is the standardized feature width.

In the ICL setting, we form a context set $\mathcal{D}_{ctx} = \{(x_i, y_i)\}_{i=1}^n$ from labeled rows of $T_\star$ and predict the label of a query row $(x_q, y_q)$ via a single forward pass:

$$P_\theta(y_q \mid x_q, \mathcal{D}_{ctx}).$$

# 4. Design Principles of Relational Prior

The efficacy of a foundation model depends on its exposure to vast and diverse datasets. To achieve this in the relational domain, we seek a *universal prior within a well-defined family* of relational database distributions: one that can synthesize diverse, logically consistent RDBs while remaining tractable to learn and sample from. However, the combinatorial space of "all possible relational databases" is intractable. Just as single-table models rely on structural assumptions (e.g., i.i.d. sampling/exchangeability) to make learning solvable, we introduce a **Relational Inductive Bias**: a small set of structural constraints that reduces the generation space while preserving the complex topological dependencies commonly observed in real-world systems.

## 4.1. Relational Assumptions

To make the generation of complex RDBs tractable, we summarize three core structural principles. These assumptions constrain the generative prior to a family of distributions that remains consistent with relational logic while avoiding the full combinatorial space of arbitrary schemas.

**Assumption 4.1** (Schema Acyclicity). The schema graph $G_S$ (Definition 3.3), oriented as **parent → child**, is a directed acyclic graph (DAG).

*Justification:* Many real-world analytic schemas (e.g., star/snowflake designs) are naturally acyclic. In our pre-processing of major benchmarks (e.g., Spider (Yu et al., 2018), 4DBInfer (Wang et al., 2024)), over 95% of schemas are acyclic. Restricting to DAGs also enables efficient topological generation of tables.

**Assumption 4.2** (Relational Markovian Locality). Conditioned on the realized structural skeleton, i.e., the instance graph $G_{in}$ and structural row states $\mathcal{Z}$, each generated attribute is locally determined. Concretely, let $A_{r,c}$ denote the value of column $c$ for row $r$, and let $\mathcal{A}_{-r,c}$ denote all other generated attributes. There exists a local parent set $\text{Pa}(A_{r,c})$ contained in a bounded $k$-hop neighborhood of $r$ in $G_{in}$ such that

$$P(A_{r,c} \mid \mathcal{A}_{-r,c}, G_{in}, \mathcal{Z}) = P(A_{r,c} \mid \text{Pa}(A_{r,c}), Z_r).$$

*Justification:* This captures empirical locality in relational systems: an entity's attributes are usually governed by its connected records and latent state, rather than by unrelated entities. This locality makes synthetic generation tractable through bounded relational aggregation.

**Assumption 4.3** (Conditional Exchangeability / Mechanism Sharing). Within any table $T$, rows are exchangeable conditional on the realized instance graph (and structural states): permuting row identities within $T$ does not change the joint distribution, provided relational links are permuted consistently. Equivalently, rows of the same table are governed by shared mechanisms that are permutation-invariant to the ordering of neighbor sets.

*Justification:* Row indices carry no semantics; dependence between rows is mediated by relational links, and shared mechanisms enable learning from variable-size sets.

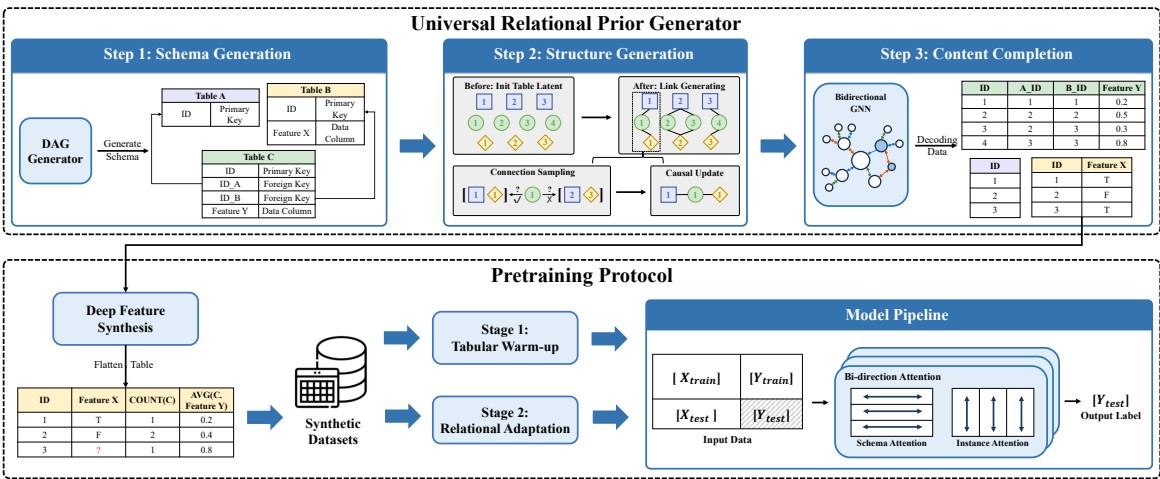

*Figure 1.* **Overview of the RDB-PFN Framework.** The top panel illustrates our **Universal Relational Prior**, which synthesizes diverse relational databases via a hierarchical decomposition: Schema (LayerDAG), Structure (Hybrid SCM), and Content (Hierarchical SCM). The bottom panel depicts the **Two-Stage Curriculum Learning** protocol, where the model first establishes a statistical backbone on single-table data before adapting to the complex topological signals of linearized relational data.

## 4.2. Constructive Decomposition and Expressivity

Under Assumptions 4.1–4.3, we factorize the database distribution into schema, structure, and content:

$$P(\mathcal{D}) = \underbrace{P(G_S)}_{\text{Schema}} \cdot \underbrace{P(\mathcal{D}_{struct}, \mathcal{Z} \mid G_S)}_{\text{Structural Skeleton}} \cdot$$
$$\underbrace{P(\mathcal{D}_{dep} \mid \mathcal{D}_{struct}, \mathcal{Z}, G_S)}_{\text{Dependent Content}}. \quad (1)$$

Here, $G_S$ is the table-level schema; $\mathcal{D}_{struct}$ contains structure-defining fields such as primary keys, foreign keys, and connectivity variables, which induce the instance graph $G_{in}$; and $\mathcal{D}_{dep}$ contains the remaining attributes generated conditioned on the realized relational structure.

**Expressivity.** This factorization motivates a modular prior with three components: a schema generator, a structural generator for keys/connectivity and latent row states, and a content generator for remaining attributes. Appendix E formalizes this construction. In Theorem E.23, we show that under the stated assumptions and additional technical conditions, the resulting prior can approximate any consistent RDB distribution within the corresponding assumption-defined family.

## 5. Method: The RDB-PFN Architecture

We present the implementation of the RDB-PFN framework. Our system comprises two components: a scalable **Data Generation Pipeline** and an ICL **Pretraining Pipeline**.

### 5.1. Data Generation

We instantiate the three-stage decomposition (Schema $\rightarrow$ Structure $\rightarrow$ Content) using specialized neural modules designed to capture the unique dependencies of each phase.

### 5.1.1. STAGE 1: SCHEMA GRAPH GENERATION

The first step synthesizes the schema graph $G_S = (\mathcal{V}_S, \mathcal{E}_S)$, where nodes are tables and directed edges are oriented as **parent $\rightarrow$ child**. We model the schema distribution $P(G_S)$ using either (i) **hand-designed topology priors** or (ii) a **learned topology model** trained on *public* schema graphs. In our experiments, we adopt LayerDAG (Li et al., 2024) as a learned schema-topology prior (Appendix C.1.1) to sample realistic DAG topologies. Crucially, regardless of how $G_S$ is obtained, all table *contents* used for pre-training are generated synthetically by our Stage 2–3 generators.

### 5.1.2. STAGE 2: STRUCTURAL GENERATION

For **Dependent Tables** (tables referencing $p$ parent tables), we employ a **Selective SCM** to link each new child row $v$ to a tuple of existing parent rows $(u^{(1)}, \ldots, u^{(p)})$.

**1. Latent Initialization:** We first sample a child state $z_v^{(0)} = \mathrm{MLP}_{init}(\epsilon)$, representing its latent characteristics.

**2. Connection Sampling via Attention:** We sample $M$ candidate parent tuples $C_j = (u_j^{(1)}, \ldots, u_j^{(p)})$, where each $u_j^{(t)}$ is an existing row from the $t$-th parent table. We maintain a dynamic embedding $e_u$ for each parent row $u$ (initialized at creation and optionally updated during generation). Each candidate tuple is embedded as

$$h_{C_j} = \mathrm{MLP}_{comb}\Big(e_{u_j^{(1)}} \oplus \cdots \oplus e_{u_j^{(p)}}\Big).$$

We then compute compatibility scores by viewing the child as a query and the tuple as a key:

$$s_j = \Big\langle z_v^{(0)} W_Q, \, h_{C_j} W_K \Big\rangle, \quad C^\star \sim \mathrm{Softmax}(s_1, \ldots, s_M).$$

**3. Causal Update:** Once the connection is established, the child integrates the chosen parents to form its final latent

state:
$$z_v = \text{MLP}_{child}\left(z_v^{(0)} \oplus h_{C^\star}\right).$$

Optionally, we apply a feedback update to the chosen parents to control the resulting topology:

$$e_u \leftarrow \text{MLP}_{fb}(e_u \oplus z_v), \qquad \forall u \in C^\star.$$

This feedback mechanism smoothly interpolates between uniform random attachment (frozen or delayed updates) and preferential attachment (immediate positive updates), enabling diverse distributions across generated schemas.

### 5.1.3. STAGE 3: CONTENT COMPLETION

The final stage generates the observable data columns $\mathcal{D}_{cols}$. By delaying feature synthesis until the topology is fixed, we ensure that every generated data point is conditioned on the complete, globally-refined structural context.

To approximate the conditional distribution $P(\mathcal{D}_{cols} \mid G_{in}, \mathcal{Z})$, we employ a **Bidirectional Graph Neural Network**. This architecture acts as the practical instantiation of the Hierarchical SCM, propagates the latent causal states across the instance graph to induce correlations.

**1. Relational Message Passing.** We initialize each row-node with its latent state $h_v^{(0)} = z_v$ and perform $K$ rounds of heterogeneous propagation on the instance graph. Let $\tau$ denote a relation type induced by an FK (and $\tau^{-1}$ its reverse), with neighbor sets $\mathcal{N}_\tau(v)$ and incident relations $\mathcal{R}(v)$ as defined in Definition 3.4. We aggregate messages separately per PK-FK edge type:

$$h_v^{(\ell+1)} = \text{Update}\left(h_v^{(\ell)}, \bigoplus_{\tau \in \mathcal{R}(v)} \underbrace{\sum_{u \in \mathcal{N}_\tau(v)} \text{MLP}_\tau\left(h_u^{(\ell)}\right)}_{\text{Permutation-invariant over neighbors}}\right)$$

**2. Universal Decoding.** After $K$ layers, the final embedding $h_v^{(K)}$ contains a globally contextualized representation of the row. A shared decoder maps this state to the values of the columns: $\{A_v\} = \text{MLP}_{dec}(h_v^{(K)})$. Continuous features are obtained through clipping/normalization transformations, while categorical features are obtained through discretization/binning, completing the synthetic database $\mathcal{D}$.

### 5.2. Architectural Implementation

Our backbone is a Transformer adapted for relational data. To bridge the gap between graph-structured RDBs and vector-based architectures, we employ a two-stage process:

**1. Graph Linearization.** Given a database instance $\mathcal{D}$ and a target table $T_\star$, we apply Deep Feature Synthesis

(DFS) (Kanter & Veeramachaneni, 2015) as a deterministic linearization operator: $X = \text{DFS}(\mathcal{D}, T_\star) \in \mathbb{R}^{|\mathcal{V}(T_\star)| \times p}$. DFS recursively aggregates relational neighborhoods via (i) *Forward Inheritance* (propagating parent attributes to children) and (ii) *Backward Aggregation* (summarizing child sets with permutation-invariant statistics), yielding a context-enriched single-table representation.

**2. Bi-Attention Reasoning.** We process this linearized input using a simplified TabPFNv2 (or NanoPFN) architecture (Hollmann et al., 2022; Pfefferle et al., 2025). The model alternates between two attention mechanisms to approximate the posterior:

- **Schema Attention (Within-same-row):** Attends across features to model inter-feature dependencies. Since DFS features in different columns are organized by different relational primitives, this attention captures relational signals expressed in the linearized representation.
- **Instance Attention (Within-same-column):** Attends across rows to perform In-Context Learning, enabling the query row to leverage patterns from labeled context rows with similar DFS-induced structural summaries.

In this initial release, we use a binary classification head rather than the more common multi-class head adopted by many tabular foundation models, because the evaluated Rel-Bench and 4DBInfer classification tasks are overwhelmingly binary. However, multiclass tasks can still be supported through a straightforward one-vs-rest extension.

### 5.3. Pretraining Protocol

We adopt a Two-Stage Curriculum to decouple statistical learning from topological reasoning:

- **Stage 1: Tabular Warm-up.** The model is pre-trained on synthetic single-table datasets. This establishes a "statistical backbone," allowing the model to master distribution matching and outlier detection without structural noise.
- **Stage 2: Relational Adaptation.** We transition to full RDBs generated by our prior. With a stable statistical foundation, the model focuses on interpreting the complex, aggregated signals from DFS, effectively learning to treat topological context as a predictive feature.

## 6. Experiments

We designed our experiments to rigorously evaluate the RDB-PFN as a Foundation Model for relational databases. We structure our analysis around three core questions:

- **RQ1 (RDB Foundation Model Capabilities):** Can RDB-PFN generalize to unseen real-world RDBs without fine-tuning? How does its computational efficiency and architectural complexity compare to existing tabular foundation models?

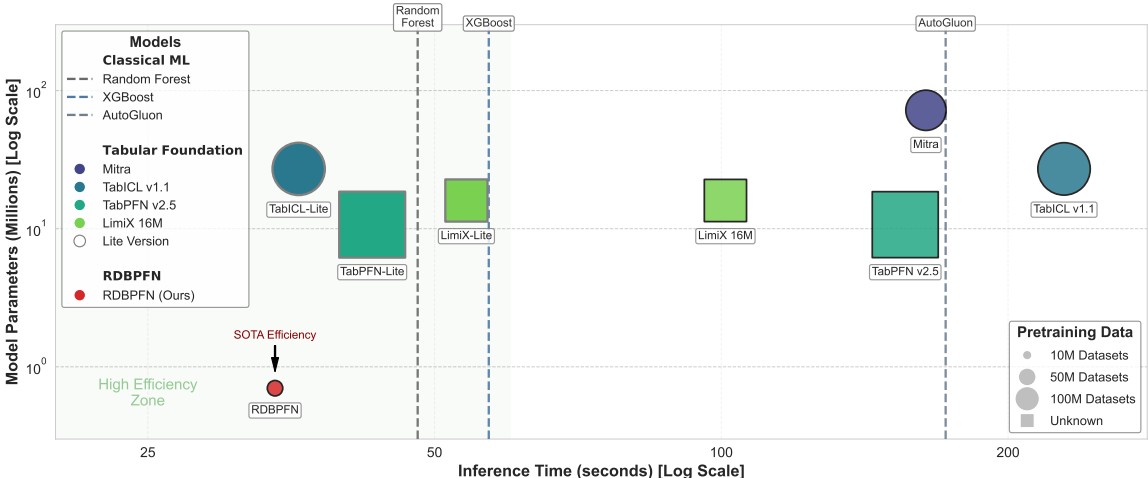

*Figure 2.* **Resource Efficiency Frontier.** Comparison of model complexity across **Inference Latency** (X-axis), **Parameter Count** (Y-axis), and **Pre-training Data Volume** (Bubble Size). "Lite" baselines denote single-estimator configurations with ensembling disabled. RDB-PFN (red point) uses a compact 0.7M-parameter backbone and is pre-trained on only ∼2M synthetic datasets. Compared with full/default tabular foundation-model baselines, it is 3.0–6.7× faster at inference; among baselines with disclosed pre-training budgets, it uses 2.5–4.4% as many pre-training datasets.

- **RQ2 (Single-Table Impact):** How does relational pre-training impact performance on standard single-table tasks? Does the model retain general tabular reasoning capabilities despite its specialized prior?
- **RQ3 (Linearized Relational Prior Analysis):** When RDBs are linearized via DFS, what distinguishes their statistical structure from standard single-tabular data?

### 6.1. Experimental Setup

**Datasets.** We evaluate relational reasoning on 19 diverse tasks curated from the 4DBInfer (Wang et al., 2024) and RelBench (Robinson et al., 2024) benchmarks. These tasks span domains including e-commerce, clinical trials, and sports analytics, varying in complexity from simple attribute prediction to complex behavioral modeling requiring temporal aggregation. Extensive prior work has established that these tasks benefit significantly from structure-aware modeling, making them a rigorous testbed for our framework. Additionally, we evaluate single-table performance using a Tabular Benchmark (Grinsztajn et al., 2022). Full dataset statistics are provided in Appendix A.

**Evaluation Protocol.** We do evaluation under a strict global few-shot ICL protocol. A small labeled support set is shared across all query rows of a task. This setting tests rapid adaptation to a new RDB under limited labels. Specifically, we evaluate the aggregated performance across a spectrum of few-shot context sizes $N \in \{64, \ldots, 1024\}$. For the single-table benchmark, we downsample each dataset to a maximum of $N = 1000$ samples. To ensure statistical robustness, we report the mean performance across 10 distinct random seeds for all tasks.

**Baselines.** To benchmark RDB-PFN as a genuine foundation model, we compare against two types of methods:

- **Single-Table Foundation Models (w/ DFS):** We benchmark against state-of-the-art ICL tabular models, including TabPFNv2.5 (Grinsztajn et al., 2025), TabI-CLv1.1 (Qu et al., 2025), Mitra (Zhang et al., 2025a), and LimiX16M (Zhang et al., 2025b). Because these architectures cannot natively process relational schemas, we provide them with the exact same linearized DFS features used by RDB-PFN to ensure a strictly fair comparison of modeling capabilities. To facilitate direct architectural comparisons, we evaluate both their default (ensemble) variants and their "Lite" (single-estimator) variants.
- **Classical Supervised Learning:** We provide reference points using Random Forest (Breiman, 2001), XG-Boost (Chen, 2016), and AutoGluon (Erickson et al., 2020). While these methods require iterative fitting (violating the zero-shot ICL constraint), they remain robust industrial baselines. Because raw library defaults often underfit and perform poorly on DFS-generated relational features, we apply lightweight, budgeted hyperparameter optimization to these baselines to ensure they remain competitive without exceeding practical runtime limits.

We also discuss graph-based RDB models such as Griffin (Wang et al., 2025) and RT (Ranjan et al., 2025) as native relational references. These methods operate under different supervised or label-access protocols from our strict global-context ICL setting, so we treat them as contextual comparisons rather than direct apples-to-apples baselines. Detailed configurations and additional protocol discussion are provided in Appendix B.

**Model variants.** We report both **RDB-PFN-Single** and the full **RDB-PFN** in analysis. The variants share the same architecture and inference procedure, and differ only in whether the relational pre-training stage is included.

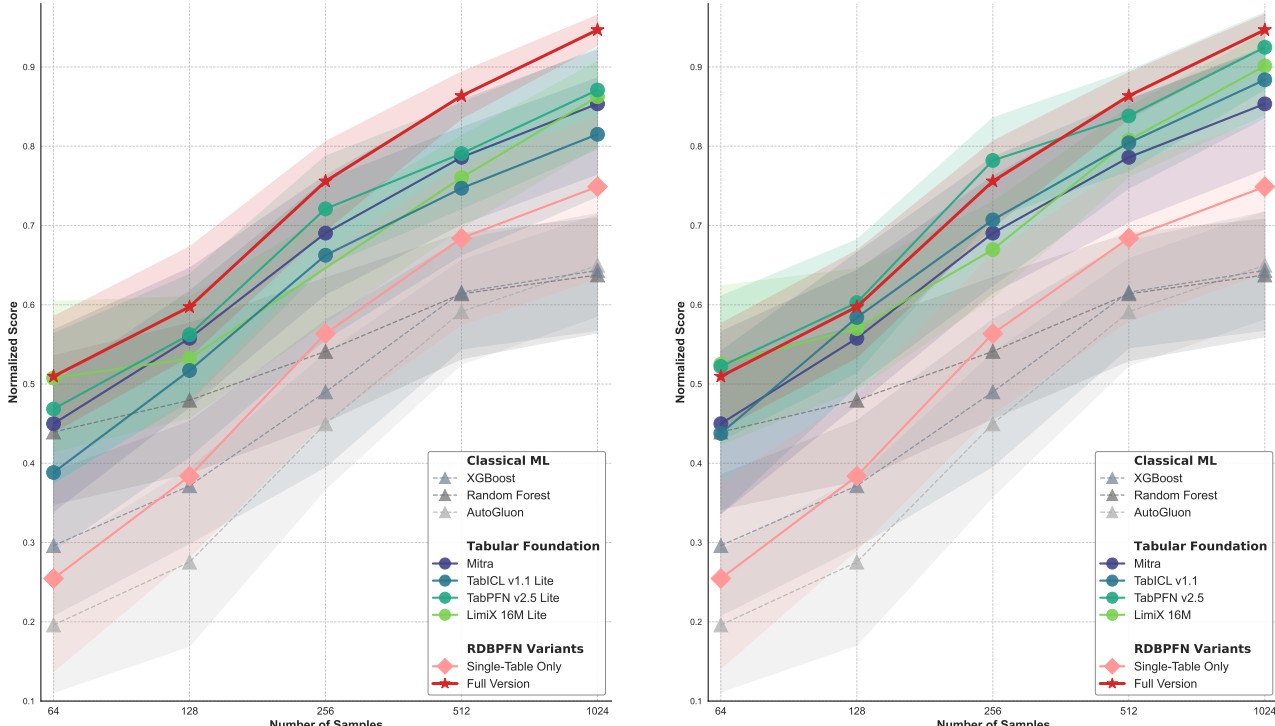

*(a)* **Lightweight Regime (Single Estimator).**  *(b)* **Standard Regime (Ensemble Enabled for other TFMs).**

*Figure 3.* **Relational Few-Shot Performance across Evaluation Protocols.** We report aggregated normalized performance across 19 relational tasks (higher is better). **(a) Single-Estimator Protocol:** all baselines are constrained to one estimator (ensembling disabled). RDB-PFN clearly surpasses all baselines. **(b) Default Protocol:** baselines run with their official default inference pipelines (which may include test-time ensembling), while RDB-PFN remains a single forward-pass estimator. RDB-PFN maintains superior average performance while offering **3.0x – 6.7x faster** inference, positioning it on the optimal frontier of the efficiency-accuracy trade-off.

### 6.2. RQ1: RDB Foundation Model Capabilities

We first establish the performance and computational profile of RDB-PFN on the 19 real-world relational tasks.

**Efficiency & Complexity Analysis.** As illustrated in Figure 2, RDB-PFN achieves a favorable efficiency–performance trade-off. The main practical gain is inference efficiency: compared with full/default tabular foundation-model baselines, RDB-PFN is approximately **3.0–6.7× faster**. This advantage is largely due to avoiding expensive default ensembling, while the smaller backbone provides an additional efficiency benefit. Compared with Lite single-estimator variants, RDB-PFN still owns a moderately faster inference.

RDB-PFN is also data-efficient: it is pre-trained on only ∼**2 million** synthetic datasets. Among baselines with disclosed pre-training budgets, this corresponds to only **2.5–4.4%** as many pre-training datasets. RDB-PFN also uses a compact **0.7M**-parameter backbone, corresponding to **1.0–6.5%** of the parameters of the compared tabular foundation models. These results suggest that a relationally matched synthetic prior can provide a useful inductive bias without requiring the scale of current single-table foundation models.

**SOTA RDB Few-Shot Performance.** Figure 3 presents the

aggregated performance across the benchmark suite.

- **Single-Model Dominance:** When restricted to a strict single-estimator setting (no ensembling), RDB-PFN explicitly outperforms all single-table baselines. This confirms that our Relational Prior yields a superior inductive bias for structural data compared to generic tabular priors.

- **The Efficiency-Performance Frontier:** While baselines can artificially boost performance through ensembling, RDB-PFN still achieves the highest overall average performance using only a single estimator. It delivers superior predictive accuracy while maintaining the rapid inference speed of a lightweight model.

### 6.3. RQ2: Single-Table Performance

To understand the trade-offs of our specialized design, we evaluated RDB-PFN on standard single-table benchmarks (Grinsztajn et al., 2022).

**Positive Transfer from Relational Pre-training.** As shown in Figure 4, RDB-PFN performs competitively in the few-shot single-table setting. More importantly, the full RDB-PFN improves over the single-table-only variant. Since these two variants share the same architecture and differ only in whether relational synthetic pre-training is included,

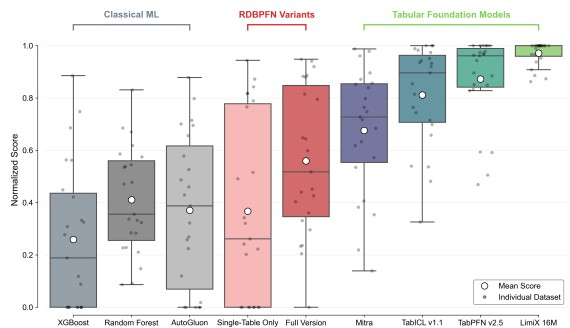

*Figure 4.* **Single-Table Performance Analysis.** We compare classical baselines, specialized tabular foundation models, and two RDB-PFN variants. The single-table benchmark serves as a transfer check rather than the primary target setting. RDB-PFN trails specialized single-table TFMs, while the continued relational synthetic pre-training improves over its single-table-only variant, suggesting a positive transfer.

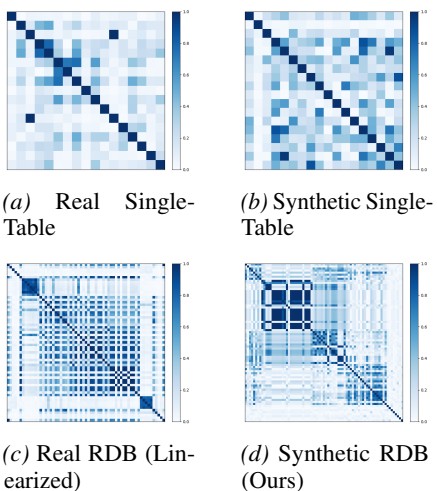

*(a)* Real Single-Table

*(b)* Synthetic Single-Table

*(c)* Real RDB (Linearized)

*(d)* Synthetic RDB (Ours)

*Figure 5.* **Structural Correlation Patterns.** Correlation heatmaps for real/synthetic single-table data and real/synthetic DFS-linearized RDBs. Linearized RDBs exhibit more block-like feature correlations induced by relational aggregations, and our synthetic RDB prior reproduces a similar qualitative pattern.

this result suggests that linearized RDBs provide useful additional pre-training variation rather than causing harmful overspecialization to relational inputs.

**The "Specialization Gap".** RDB-PFN still trails specialized single-table foundation models. This is expected: our model is optimized for relational prediction, uses a simpler single-table pre-training stage, and has fewer parameters than several specialized single-table TFMs. We therefore interpret the single-table result as a scope and transfer check, while the main contribution remains the learned relational prior for RDB tasks.

### 6.4. RQ3: Analysis of the Linearized Relational Prior

We hypothesize that RDB-PFN outperforms single-table baselines because it captures a distinct topological signature inherent to relational data. For instance, the DFS process creates clusters of highly correlated columns (e.g., the `Sum`

and `Mean` aggregations of the same parent table), which manifest as distinct statistical patterns that standard tabular priors fail to anticipate.

**Visualization Analysis.** Figure 5 compares the correlation matrices of representative datasets. Both real and synthetic single-table datasets exhibit diffuse, unstructured correlation patterns. Conversely, both real and synthetic RDBs display a prominent **Block-Diagonal Structure**, where dense blocks correspond to correlated feature families derived from parent relations. This visual alignment strongly suggests that our synthetic generator effectively models the structural manifold of real-world relational data, allowing the network to internalize these dependencies during pretraining.

**Additional analysis and stress tests.** Beyond the main few-shot comparison, we conduct additional analyses to test the robustness of our conclusions. First, we strengthen the classical baselines with budgeted XGBoost tuning and a stronger AutoGluon preset, and also report PR-AUC for imbalanced binary tasks; both checks preserve the qualitative comparison. Second, we study adaptation and scale beyond strict ICL: target-task fine-tuning further improves performance, cross-task fine-tuning gives smaller but positive gains, and chunked-support ensembling shows continued benefits from larger effective support sizes. Third, ablations over the relational prior and backbone show that simplifying the schema generator, link-generation mechanism, temporal component, locality assumption, or interleaved attention design generally degrades performance. Baseline configurations and protocol discussions are provided in Appendix B; additional per-task results and ablation details are provided in Appendix D.

## 7. Conclusion and Limitations

We presented RDB-PFN, a foundation model for RDBs trained entirely on synthetic data. By replacing the standard i.i.d. single-table assumption with a relational synthetic prior, RDB-PFN enables PFN-style in-context learning on relational prediction tasks. Across real-world relational benchmarks, it achieves strong few-shot performance while remaining compact and efficient.

The current work focuses on a deliberately constrained setting: binary relational classification with few-shot in-context adaptation through a shared global support set. This setting matches the dominant task type in existing public relational benchmarks and highlights the low-label, low-engineering regime targeted by RDB-PFN. In practice, relational learning scenarios can be broader, involving more labeled data for continued adaptation, more diverse task types, and richer ways to use database structure beyond Deep Feature Synthesis. We view these directions as promising next steps for extending synthetic relational pre-training to a wider range of database learning scenarios.

## Acknowledgements

This work is supported by National Natural Science Foundation of China (62550138, 62276003).

## Impact Statement

This paper presents work whose goal is to advance the field of Machine Learning. There are many potential societal consequences of our work, none which we feel must be specifically highlighted here.

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

# A. Dataset Details & Evaluation Protocol

## A.1. Relational Benchmark Statistics (4DBInfer & RelBench)

We utilize a suite of 19 diverse predictive tasks derived from 13 distinct relational databases sourced from **4DBInfer** (Wang et al., 2024) and **RelBench** (Robinson et al., 2024). This collection represents a comprehensive cross-section of real-world industrial scenarios, including e-commerce, social networks, and medical research. The selected benchmarks exhibit significant diversity across three key dimensions:

- **Scale:** The datasets range from small-scale scientific studies (e.g., *Rel-Trial* with ∼5.8M rows) to massive industrial logs (e.g., *AVS* with ∼350M rows), testing the model's scalability.

- **Topological Complexity:** The schema complexity varies from simple star schemas (e.g., *Amazon* with 3 tables) to deep, multi-hop snowflake schemas (e.g., *Rel-Trial* with 15 tables and 140 columns).

- **Target Distribution:** The tasks cover various difficulties, including highly imbalanced task types.

*Table 1.* Statistics of relational database datasets.

| Dataset | Tables | Columns | Rows |
|---|---|---|---|
| Amazon | 3 | 15 | 24,291,489 |
| AVS | 3 | 24 | 349,967,371 |
| Diginetica | 5 | 28 | 3,672,396 |
| Outbrain | 8 | 31 | 4,778,954 |
| Retailrocket | 3 | 11 | 23,033,676 |
| Stackexchange | 7 | 49 | 5,399,818 |
| Rel-amazon | 3 | 15 | 24,291,489 |
| Rel-avito | 8 | 43 | 20,679,117 |
| Rel-event | 5 | 128 | 41,328,337 |
| Rel-f1 | 9 | 77 | 97606 |
| Rel-hm | 3 | 37 | 33,265,846 |
| Rel-stack | 7 | 51 | 38,109,828 |
| Rel-trial | 15 | 140 | 5,852,157 |

*Table 2.* Statistics of relational database tasks.

| Dataset | Task Description | #Train / #Val / #Test |
|---|---|---|
| Amazon | User Churn Prediction | 1,045,568 / 149,205 / 152,486 |
| AVS | Customer Retention Prediction | 109,341 / 24,261 / 26,455 |
| Diginetica | Click-through-rate Prediction | 108,570 / 6,262 / 5,058 |
| Outbrain | Click-through-rate Prediction | 69,709 / 8,715 / 8,718 |
| RetailRocket | Conversion-rate Prediction | 80,008 / 9,995 / 9,997 |
| Stackexchange | User Churn Prediction | 142,877 / 88,164 / 105,612 |
| Stackexchange | Post Popularity Prediction | 308,698 / 38,587 / 38,588 |
| Rel-amazon | User Churn Prediction | 4,732,555 / 409,792 / 351,885 |
| Rel-amazon | Item Churn Prediction | 2,559,264 / 177,689 / 166,842 |
| Rel-avito | User Clicks Prediction | 59,454 / 21,183 / 47,996 |
| Rel-avito | User Visits Prediction | 86,619 / 29,979 / 36,129 |
| Rel-event | User Repeat Prediction | 3,842 / 268 / 246 |
| Rel-event | User Ignore Prediction | 19,239 / 4,185 / 4,010 |
| Rel-f1 | Driver DNF Prediction | 11,411 / 566 / 702 |
| Rel-f1 | Driver Top3 Prediction | 1,353 / 588 / 726 |
| Rel-hm | User Churn Prediction | 3,871,410 / 76,556 / 74,575 |
| Rel-stack | User Engagement Prediction | 1,360,850 / 85,838 / 88,137 |
| Rel-stack | User Badge Prediction | 3,386,276 / 247,398 / 255,360 |
| Rel-trial | Study Outcome Prediction | 11,994 / 960 / 825 |

## A.2. Single-Table Benchmark Details

To verify the backward compatibility of RDB-PFN with standard tabular tasks, we evaluate on a subset of the **Tabular Benchmark** proposed by Grinsztajn et al. (2022). While our primary focus is relational reasoning, this evaluation ensures that our architecture maintains competitive performance on "flat" feature matrices.

We selected 23 diverse classification datasets ranging from small-scale tasks (e.g., *Bioresponse*) to larger industrial logs (e.g., *Higgs*, *Covertype*). Table 3 details the characteristics of these datasets.

*Table 3.* Statistics of the Single-Table Classification Datasets used for verification. **(Num)** and **(Cat)** denote two variants of the same dataset: one containing exclusively numerical features and one containing categorical features, respectively.

| Dataset | # Samples | # Feats | Dataset | # Samples | # Feats |
|---|---|---|---|---|---|
| Bioresponse | 3,434 | 419 | Default-Credit (Num) | 13,272 | 20 |
| Diabetes130US | 71,090 | 7 | Default-Credit (Cat) | 13,272 | 21 |
| Higgs | 940,160 | 24 | Electricity (Num) | 38,474 | 7 |
| MagicTelescope | 13,376 | 10 | Electricity (Cat) | 38,474 | 8 |
| MiniBooNE | 72,998 | 50 | Eye_Movements (Num) | 7,608 | 20 |
| Albert | 58,252 | 31 | Eye_Movements (Cat) | 7,608 | 23 |
| Bank-Marketing | 10,578 | 7 | HELOC | 10,000 | 22 |
| California | 20,634 | 8 | House_16H | 13,488 | 16 |
| Compas-Two-Years | 4,966 | 11 | Jannis | 57,580 | 54 |
| Covertype (Num) | 566,602 | 10 | Pol | 10,082 | 26 |
| Covertype (Cat) | 423,680 | 54 | Road-Safety | 111,762 | 32 |
| Credit | 16,714 | 10 | | | |

## A.3. Evaluation Protocol

To ensure statistical rigor, we adhere to a standardized few-shot evaluation protocol across all experiments.

**RDB Few-Shot Evaluation.** For relational tasks, we evaluate the model's ability to learn in-context across a spectrum of data availability regimes.

- **Context Sizes ($N_{shot}$):** We iterate through context lengths of $k \in \{64, 128, 256, 512, 1024\}$.

- **Robustness:** For each task and each $N_{shot}$, we perform inference over **10 distinct random seeds**. In each run, the context examples are sampled uniformly at random from the training split. We report the mean and standard deviation of the performance metric across these 10 folds to account for variance in context quality.

**Single-Table Verification.** For the single-table benchmarks, we adopt a lightweight, fixed-budget evaluation protocol.

- **Setup:** We fix the dataset size to $N = 1000$ samples, downsampling larger datasets as necessary.

- **Repeated Trials:** Using the downsampled datasets, we apply a 70%/30% train-test split. We then train and evaluate the model, repeating this process 10 times to ensure statistical stability.

**Task Standardization & Metrics.** In this work, we focus primarily on Binary Classification, which constitutes the vast majority of high-value industrial relational problems (e.g., Churn, CTR, Fraud).

- **Scope:** Consequently, all selected benchmarks are binary classification tasks. Multi-class targets (if any) can be binarized (e.g., "Top-1 vs. Rest") to align with the current architecture's output head.

- **Metric:** We report ROC-AUC as the primary metric.

# B. Baseline Configurations & Analysis

### B.1. Classical Supervised Learning Baselines

To represent the Industrial Standard, we evaluate three widely-used algorithms. Crucially, to ensure a fair comparison of modeling capacity, all baselines are fed the exact same linearized DFS features as RDB-PFN.

- **AutoGluon** (Erickson et al., 2020): We use the "medium_quality" preset as an efficient default that balances predictive performance and training cost. To provide a stronger AutoML baseline, we additionally evaluate the "best_quality" preset with a 5-minute time budget per task.

- **Random Forest** (Breiman, 2001): We observed that the standard configuration with 100 estimators led to rapid inference but underfit complex relational features. To provide a stronger baseline while maintaining reasonable speed, we increase the capacity by setting the number of estimators to 500.

- **XGBoost** (Chen, 2016): Similarly, standard configurations often resulted in premature convergence and suboptimal performance. We therefore use a strengthened fixed configuration with 5000 estimators, learning rate 0.01, maximum depth 12, and both "subsample" and "colsample_bytree" set to 0.8. We also evaluate a tuned variant, where hyperparameters are selected on an internal validation split generated from the available training data.

### B.2. Single-Table Foundation Model Configurations

All single-table foundation models are evaluated using the same DFS-Linearized inputs. For models that support ensembling, we evaluate both the **Default** (ensemble) configuration and a **Lite** (single-estimator) configuration to facilitate a direct comparison with RDB-PFN's efficiency.

- **TabPFNv2.5** (Grinsztajn et al., 2025): We use the official "tabpfn-v2.5" checkpoint (`tabpfn-v2.5-classifier-v2.5_default.ckpt`).
  - *Lite Variant:* We disable the default ensembling mechanism (setting $N_{ens} = 1$).

- **TabICLv1.1** (Qu et al., 2025): We utilize the official default checkpoint.
  - *Lite Variant:* We evaluate the model without its standard ensemble aggregation.

- **Mitra** (Zhang et al., 2025a): We use the checkpoint provided via the AutoGluon integration. Note that the default configuration for Mitra already utilizes a single estimator; therefore, no separate "Lite" variant is required.

- **LimiX16M** (Zhang et al., 2025b): We use the official checkpoint.
  - *Preprocessing Adjustment:* We observed that the default SVD preprocessing caused numerical instability and crashes due to the higher noise variance in linearized RDB features. Consequently, we disable SVD preprocessing for all experiments.
  - *Lite Variant:* We evaluate the model with ensembling disabled.

### B.3. Graph-Based RDB Foundation Models

We also investigated Graph-Based RDB foundation models. Unlike single-table approaches, these models ingest raw relational schemas directly, theoretically offering a higher potential to capture complex structural topologies without the information loss inherent in flattening. However, they currently face significant challenges regarding training complexity and scalability.

We examined two leading methods: **Griffin** (Wang et al., 2025) and **RT** (Ranjan et al., 2025). Both primarily focus on supervised fine-tuning and may incorporate label semantics in ways that diverge from our strict in-context learning protocol.

- **Griffin:** This framework's few-shot evaluation is limited to a subset of datasets and specific shot counts ($N \in \{512, 4096\}$). Furthermore, its primary focus is on *transfer learning* via fine-tuning on domain-specific corpora, rather than providing a single, universal inference engine. Since Griffin provides four distinct model variants rather than a unified foundation model, a direct comparison is challenging.

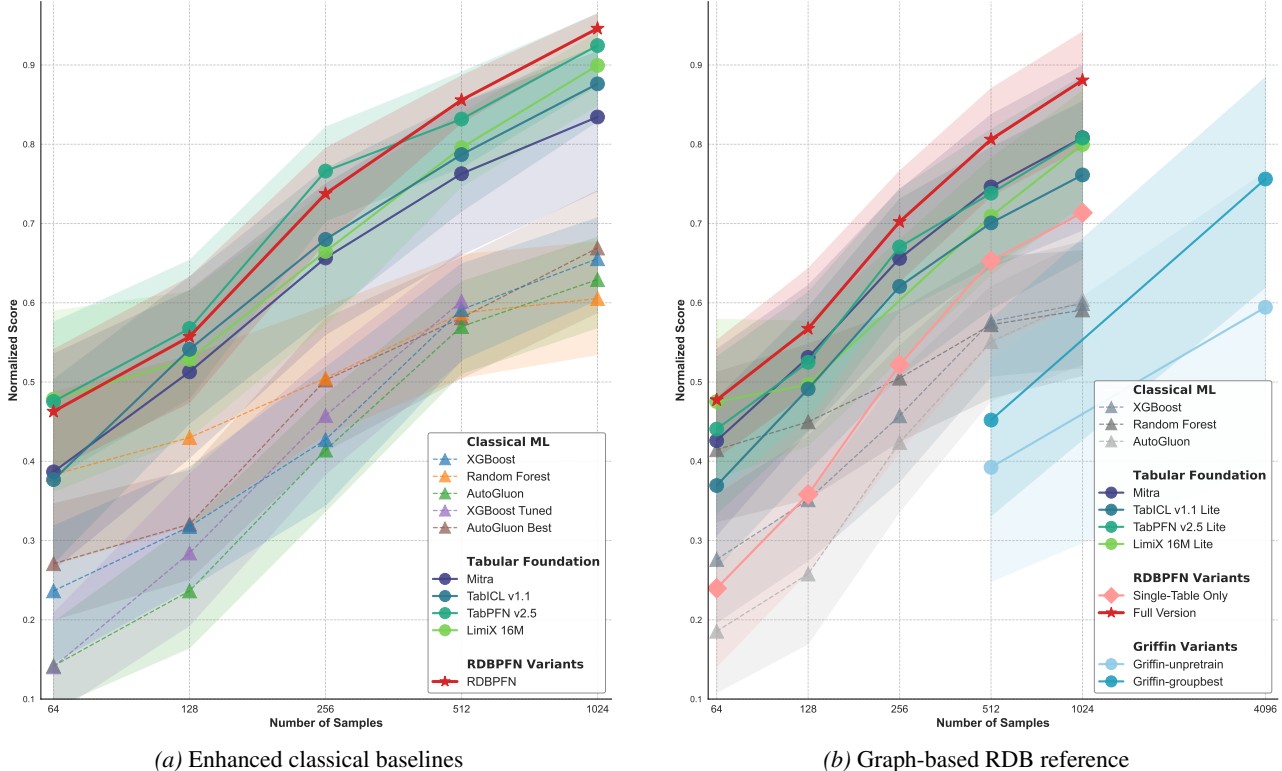

*(a)* Enhanced classical baselines

*(b)* Graph-based RDB reference

*Figure 6.* **Additional Baseline Analysis.** Left: comparison after strengthening classical baselines with tuned XGBoost and AutoGluon using the `best_quality` preset under a 5-minute time budget. Right: comparison with Griffin on the subset of overlapping tasks, using an idealized reference that selects the best-performing Griffin checkpoint for each task group. These additional comparisons preserve the qualitative conclusion while clarifying the strength and protocol differences of the baselines.

*Comparison Strategy:* To approximate a comparison, we adopted an "Ensemble" strategy for Griffin: for each task, we selected the best-performing Griffin checkpoint from its respective domain group. Even against this idealized baseline, RDB-PFN consistently achieves superior performance across the intersecting tasks, demonstrating the robustness of our universal prior (shown in figure 6).

- **RT:** RT studies a different zero-shot relational protocol. Although it does not update model weights using target labels, it materializes label information into the database as an auxiliary table during inference. In the reported RT setting, labels from the training split are included in this table, so the sampled relational context can expose target-specific historical labels or labels of nearby connected entities. In contrast, our strict ICL protocol uses a fixed global support set shared across test predictions and does not inject per-target labels into the relational graph. Thus, adapting RT to our **global-context protocol** or adapting RDB-PFN to RT's **local graph-conditioned protocol** would change the accessible supervision, not only the computational cost. We therefore discuss RT as a related relational reference rather than reporting it as a direct same-protocol baseline.

## C. Implementation Details

### C.1. Data Generation Model Details

#### C.1.1. STAGE 1: SCHEMA GENERATION WITH LAYERDAG

To approximate the complex distribution of realistic database schemas $P(G_S)$, we employ LayerDAG (Li et al., 2024), an autoregressive discrete diffusion model. LayerDAG is well suited for relational schema synthesis because it decomposes DAG generation into a sequence of bipartite layers, a structure that naturally enforces the acyclic dependencies required for valid Foreign Key (FK) joins. The generation process proceeds autoregressively as follows:

- **Layerwise Decomposition:** The model views the schema graph as a topological sequence of layers $\mathcal{V}^{(1)}, \ldots, \mathcal{V}^{(L)}$. The first layer $\mathcal{V}^{(1)}$ consists of independent "Source Tables" (root nodes), while subsequent layers contain dependent tables that reference previous layers.

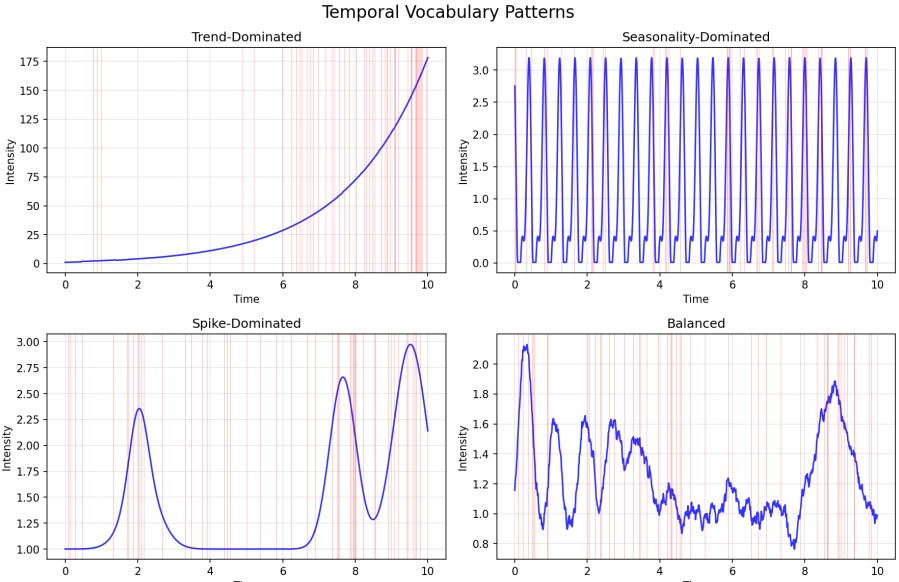

*Figure 7.* **Diversity of Generated Temporal Patterns.** We visualize the row density over time for four distinct synthetic tables. By composing primitives from our Temporal Vocabulary (Trend, Seasonality, Spike), the prior generates complex, non-i.i.d. distributions.

---

**Algorithm 1** Selective SCM for Foreign-Key Generation (one dependent table)

---

**Require:** Parent row sets $\{U^{(1)}, \ldots, U^{(p)}\}$, #child rows $n$, candidate size $M$, params $\psi$
**Ensure:** FK assignments and latent states $\{z_v\}$
 1: **for** $i = 1$ to $n$ **do**
 2:     Sample child initialization $z^{(0)} \leftarrow \mathrm{MLP}_{init}(\epsilon)$
 3:     Sample candidate parent tuples $\{C_j\}_{j=1}^M$, each $C_j = (u_j^{(1)}, \ldots, u_j^{(p)})$
 4:     Tuple embed: $h_{C_j} \leftarrow \mathrm{MLP}_{comb}(e_{u_j^{(1)}} \oplus \cdots \oplus e_{u_j^{(p)}})$
 5:     Score: $s_j \leftarrow \langle z^{(0)} W_Q, h_{C_j} W_K \rangle$
 6:     Sample $C^\star \sim \mathrm{Categorical}(\mathrm{Softmax}(s_1, \ldots, s_M))$ and set FKs accordingly
 7:     Update child state: $z_v \leftarrow \mathrm{MLP}_{child}(z^{(0)} \oplus h_{C^\star})$
 8:     Optional feedback: for each selected parent $u \in C^\star$, update $e_u \leftarrow \mathrm{MLP}_{fb}(e_u \oplus z_v)$
 9: **end for**

---

- **Conditional Diffusion:** At each step $l$, the model conditions on the partial graph generated so far, $G^{(\leq l-1)}$, to generate the next bipartite layer. A discrete diffusion process jointly synthesizes both the table nodes (metadata/attributes) and the FK edges, ensuring that every generated connection respects the logical constraints of the schema.

We pre-train this module on a small corpus of real-world database schemas (Yu et al., 2018; Li et al., 2023). This allows our relational prior to sample realistic industrial topologies, ranging from star schemas to deep snowflake structures, which then serve as schema skeletons for synthetic data generation. To test whether this learned schema model is necessary, we also compare against a simpler DAG generator in Appendix D.3. The LayerDAG-based prior gives a slight but consistent improvement, suggesting that realistic schema topology is useful but not the only source of the model's performance.

### C.1.2. STAGE 2: STRUCTURAL GENERATION DETAILS

The structural generation stage relies on a selective SCM, where the foreign-key connection probability is parameterized. The pseudocode for generating one dependent table is shown in Algorithm 1.

In the default attention-based sampler, $W_Q$ and $W_K$ are learnable linear projections initialized with the standard Kaiming-uniform initialization. The selected parent tuple is sampled stochastically from the softmax distribution rather than chosen by an argmax, preserving diversity in the generated structural skeleton. We also implement simpler link-generation variants, including fixed-probability sampling and concatenation followed by MLP scoring; their ablation results are reported in Appendix D.3.

In addition, to simulate the diversity of real-world database topologies, which range from nearly uniform random graphs to highly skewed scale-free networks, we implement a **Hybrid Sampling Strategy** and a **Temporal Latent Initialization** mechanism.

**1. Topological Control via Hybrid Sampling.** As posited in the main text, the distribution of node degrees (e.g., the number of Orders associated with a User) can be governed by the parent update frequency. We implement this practically by mixing two generation modes:

- **Mode A: Parallel Generation (Frozen State).** In this mode, we generate child rows without updating parent embeddings at intermediate steps. We sample candidate parent tuples for all child rows in parallel, based on a static snapshot of the parent states. This approximates a more uniform random graph process, since high-degree parents do not gain an immediate advantage during batch generation.

- **Mode B: Sequential Generation (Dynamic Feedback).** In this mode, we generate children in small mini-batches. After each batch, we apply the feedback update only to the selected parents $u \in C^\star$. This increases the future selection probability of chosen parents and induces preferential-attachment-like behavior, leading to more long-tailed degree distributions.

By modulating the mixing ratio between Mode A and Mode B, together with the magnitude and frequency of the feedback update, we can continuously interpolate between more uniform and more power-law-like degree distributions.

**2. Temporal Latent Initialization.** Real-world data is rarely i.i.d.; rather, it often exhibits temporal dependencies. To capture this, we augment the latent initialization step. Instead of sampling purely from a standard Gaussian distribution, the initial child state $z^{(0)}$ can be conditioned on timestamp-derived temporal features from a **Temporal Vocabulary**:

- **Signal Primitives:** We define a library of three temporal signals:
    1. **Trend:** Linear or non-linear drift over time (e.g., a growing user base).
    2. **Seasonality:** Cyclic patterns (e.g., weekly or monthly spending habits).
    3. **Spike:** Sparse, high-magnitude events (e.g., Black Friday sales).

- **Composition:** For each table, we sample a random mixture of these primitives to form a temporal signature. This signature affects both timestamp generation and latent initialization, ensuring that generated rows reflect diverse temporal evolutions rather than static noise distributions (see Figure 7).

We further include an ablation that removes the temporal component in Appendix D.3, which tests whether these timestamp-driven signals contribute to downstream relational prediction.

**C.2. Main Model Details**

C.2.1. MODEL ARCHITECTURE

We employ a 6-layer Bidirectional Transformer architecture, optimized for high-throughput inference in resource-constrained environments.

- **Hyperparameters:** The model utilizes an embedding dimension of $d_{model} = 128$ and 4 parallel attention heads.

- **Efficiency:** This lightweight configuration allows for rapid deployment while retaining sufficient capacity to resolve complex relational patterns via the attention mechanism.

C.2.2. DFS LINEARIZATION CONFIGURATION

To linearize the relational graph into a sequence compatible with the Transformer, we apply Deep Feature Synthesis (DFS) using a restricted, robust set of aggregation primitives.

- **One-to-Many (Aggregation):** We utilize {`Mean`,`Max`,`Min`,`Count`,`Mode`} to summarize child records.

- **One-to-One (Transformation):** We employ standard identity mappings to propagate attributes from parent tables directly to child rows.

### C.2.3. TRAINING PROTOCOL

We adopt a Two-Stage Curriculum Learning strategy to stabilize training and progressively introduce relational complexity.

- **Stage 1: Tabular Warm-up (Feature Reasoning).**
  The model is initially trained on $600k$ synthetic *single-table* datasets. This phase focuses on mastering fundamental statistical properties and feature interactions.

  - *Data Dimensions:* Fixed context size of 600 rows $\times$ 18 columns.
  - *Hardware:* Single NVIDIA RTX 4090 GPU.
  - *Optimization:* We use the **Schedule-Free AdamW** optimizer (Defazio et al., 2024; Loshchilov & Hutter, 2017) with a learning rate of $lr = $ 5e-4.

- **Stage 2: Relational Adaptation Pre-training (Structural Reasoning).**
  We continue training on a mixed corpus of approximately $1.8$ million synthetic datasets, combining single-table data with complex RDBs generated by our Relational Prior. This phase adapts the model to the structural modality of aggregated features. To efficiently process the expanded corpus and increased structural complexity, training for this stage is distributed across 8 NVIDIA RTX 4090 GPUs.
  **1. Dataset Composition.** The training mix is stratified as follows:

  - **Single-Table Tasks:** $\sim$600k datasets.
  - **Relational Tasks (RDBs):** $\sim$1.2M datasets, comprising:

    * Small Prior (1-hop DFS): $\sim$800k.
    * Small Prior (2-hop DFS): $\sim$200k.
    * Large Prior (1-hop DFS): $\sim$200k.

  **2. Feature Standardization (Over-generate & Subsample).**
  Since DFS produces feature sets of variable length depending on the schema depth, we employ a standardization strategy to maintain consistent input dimensions. We initially over-compute features by generating 60 columns for 1-hop tasks and 90 columns for 2-hop tasks, and subsequently downsample them to a fixed width of 30 columns. This ensures all Stage 2 tasks share a uniform shape of 600 rows $\times$ 30 columns.
  **3. Task Augmentation.**
  To maximize data utility, we employ a multi-target sampling strategy. For each generated schema, rather than selecting a single target column, we randomly sample 6 distinct columns to serve as prediction targets. This effectively multiplies the available training instances, forcing the model to reason about different dependency directions within the same structural context.
  **4. Row Sampling and Split Construction.** For each generated RDB, we first run DFS on the full database instance and then uniformly sample rows from the resulting target table to form a fixed-size task table. Unless otherwise specified, each task contains 600 rows after downsampling. We vary task difficulty by changing the train/test split ratio.

## D. Additional Experimental Analyses

In this section, we provide additional experiments that complement the main few-shot comparison. These analyses test whether the main conclusions are robust to alternative metrics, adaptation protocols, larger effective context sizes, model-side design choices, relational-prior simplifications, and DFS configurations.

### D.1. PR-AUC on Imbalanced Tasks

The main paper reports ROC-AUC following common practice in relational prediction benchmarks. However, several tasks are class-imbalanced, where PR-AUC can provide a more sensitive view of minority-class prediction quality. We therefore re-evaluate the main binary classification benchmarks using PR-AUC. As shown in Figure 8, the comparison under PR-AUC is broadly consistent with the ROC-AUC results. This suggests that the observed gains are not only due to improvements on easy majority-class rankings, but also remain visible under a metric more sensitive to positive-class retrieval.

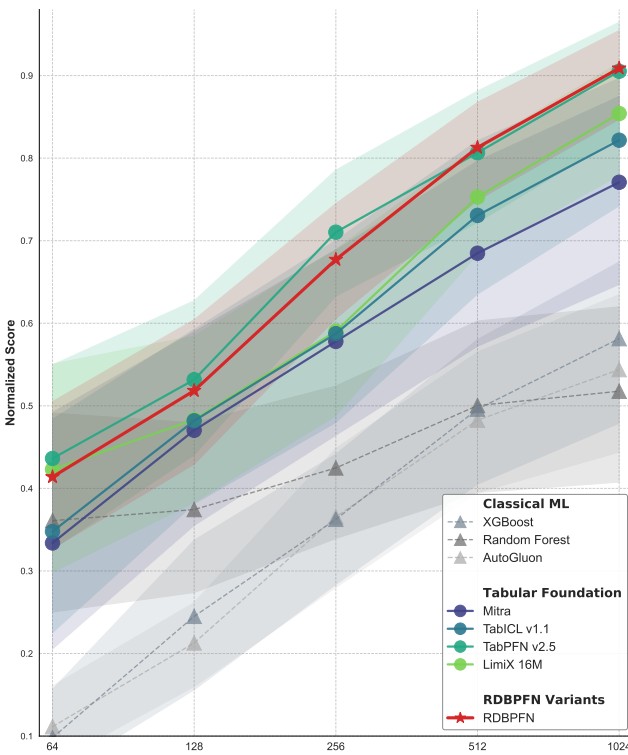

*Figure 8.* **Main benchmark results under PR-AUC.** We report PR-AUC to complement ROC-AUC on imbalanced binary classification tasks. The qualitative comparison remains similar to the ROC-AUC results in the main text.

## D.2. Adaptation and Effective Context Scaling

Our main evaluation focuses on strict few-shot in-context learning, where the model adapts to a new relational task through a fixed labeled context without updating parameters. We additionally evaluate whether RDB-PFN can benefit from real relational data when gradient-based adaptation is allowed, and whether it can use larger effective support sets through chunked-support ensembling.

For adaptation, we compare three settings: strict ICL, target-task fine-tuning, and cross-task fine-tuning. In target-task fine-tuning, the model is fine-tuned using labeled examples from the same task that will later be evaluated. This measures the benefit of task-specific adaptation when additional labels from the target task are available. In cross-task fine-tuning, when evaluating a particular target task, we exclude that task from the fine-tuning data and fine-tune only on tasks from other datasets. This setting is intended to test whether real relational tasks provide transferable adaptation signal beyond the synthetic pre-training distribution, without directly training on the evaluated task itself. Fine-tuning is run for 1,000 steps with 1,024 samples per step. We also evaluate TabPFNv2.5 under target-task fine-tuning using its official lightweight fine-tuning configuration. Figure 9a shows that target-task fine-tuning further improves performance over strict ICL, while cross-task fine-tuning provides a smaller but positive gain.

For effective context scaling, we split a larger support set into multiple chunks, run RDB-PFN separately on each chunk, and average the predicted probabilities. Figure 9b shows that increasing the effective support size from 1,024 to 8,192 generally improves performance, with diminishing gains and increased runtime at larger support sizes. This indicates a practical performance-runtime trade-off and highlights context size as an important limitation of the current ICL setup.

## D.3. Relational Prior and Backbone Ablations

We ablate both the synthetic relational prior and the model backbone. For efficiency, we do not go through the full pretraining pipeline. Instead, we use the full data for single-table warmup and a subset of RDB-pretrain (20k RDBs) with a similar task-enhancement method that randomly selects columns as the task.

For the relational prior, we compare the full prior against simplified variants that modify schema generation, link generation,

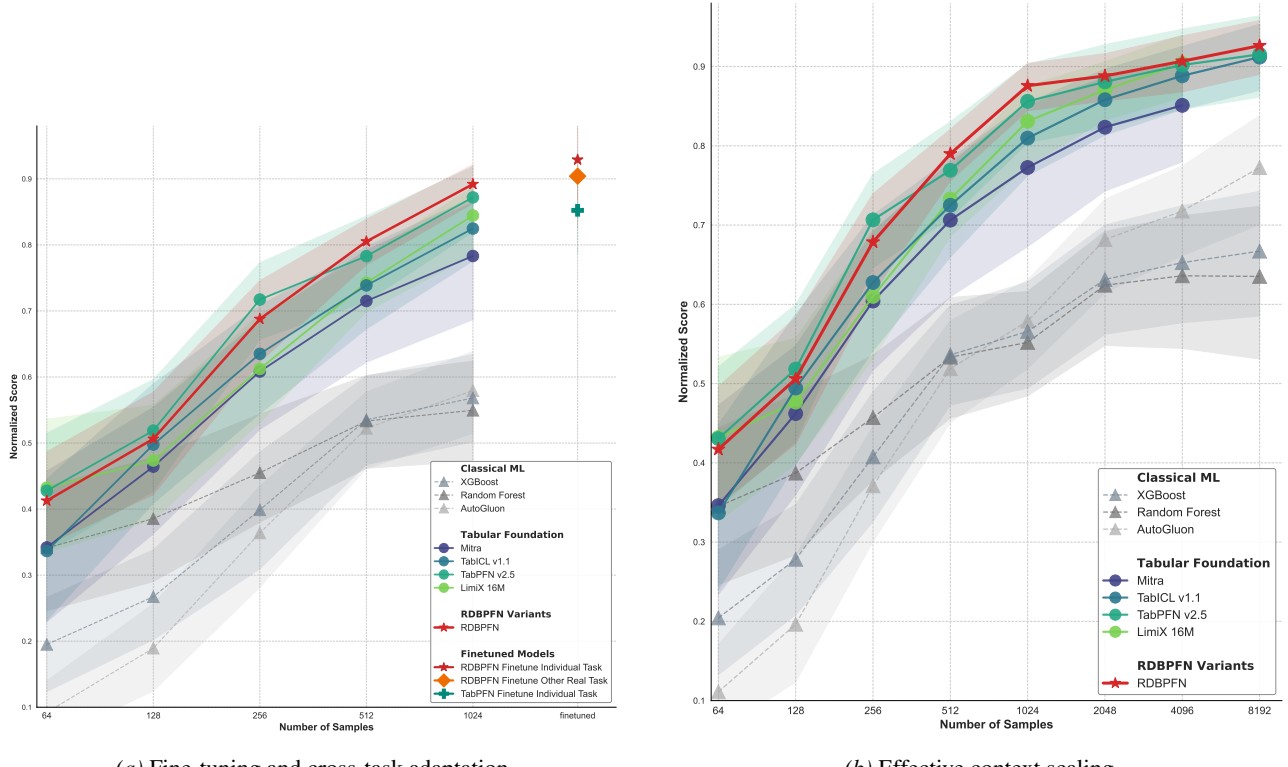

*(a)* Fine-tuning and cross-task adaptation
*(b)* Effective context scaling

*Figure 9.* **Adaptation and larger effective support.** (a) Target-task fine-tuning further improves over strict ICL, while cross-task fine-tuning gives a smaller but positive gain. (b) Chunked-support ensembling increases the effective support size beyond the pre-training context budget, improving performance with an expected runtime trade-off.

temporal modeling, and locality assumptions. The ablated variants include replacing LayerDAG with a simpler DAG generator, replacing the attention-based link sampler with simpler edge-generation variants, removing the temporal component, and weakening the locality assumption by allowing more global or randomly selected distant dependencies. As shown in Figure 10a, the full relational prior performs best overall, suggesting that these components provide a useful inductive bias for relational prediction.

For the backbone, we compare the default interleaved row–column attention design against reduced-size variants and weaker blockwise attention variants. The model-side ablations include smaller RDB-PFN variants, a blockwise row-first / column-second variant, and a blockwise column-first / row-second variant. Figure 10b shows that reducing model size changes performance in the expected direction, while replacing interleaved attention with blockwise variants causes a larger drop on relational tasks. This suggests that the attention design is important for making DFS-linearized relational signals usable.

### D.4. DFS Sensitivity

DFS is used as a practical relational-to-tabular interface for RDB-PFN and for single-table foundation-model baselines. Since DFS determines what relational information is exposed to the model, we conduct a lightweight sensitivity check over both hop depth and aggregation functions. Specifically, we compare the default 2-hop DFS configuration against a simplified 1-hop variant, and further compare the default 2-hop aggregation set with restricted 2-hop aggregation subsets. All results are evaluated on a subset of smaller relational tasks with context size 1024.

As shown in Table 4, the default 2-hop configuration improves performance on most tasks relative to 1-hop DFS, indicating that broader relational neighborhoods are often useful. The restricted aggregation variants also change performance, sometimes improving individual tasks but generally trailing the default configuration on average. This suggests that DFS is a useful but non-optimal interface: it enables shared comparison with tabular foundation models, but the choice of hop depth, aggregation functions, and feature budget can affect downstream performance.

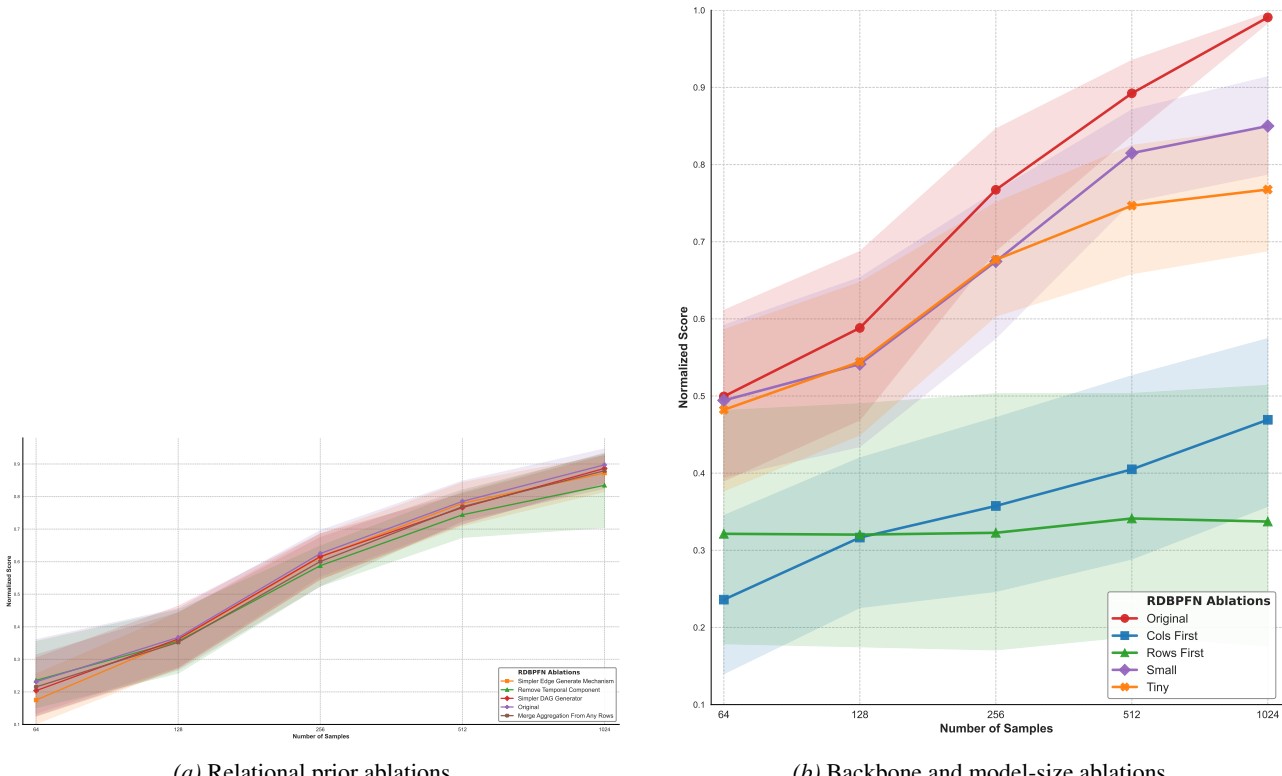

*(a)* Relational prior ablations        *(b)* Backbone and model-size ablations

*Figure 10.* **Relational prior and backbone ablations.** (a) Simplifying components of the relational synthetic prior generally degrades performance. (b) The default interleaved attention design outperforms weaker blockwise variants, while reduced-size models follow the expected capacity trend.

## E. Proof

**Proof roadmap.** Although the generative pipeline operates chronologically as *Schema → Structure → Content* (Eq. equation 1), our completeness proof is organized by *lemma dependency* rather than execution order. We first establish the completeness of **Content Completion** (Stage 3), since it reduces to approximating local conditional mechanisms that obey structured, permutation-invariant aggregation over relational neighborhoods. This yields a foundational universality result for our **Hierarchical SCM** architecture. We then leverage this result to prove the completeness of **Structural Generation** (Stage 2), which reuses the same hierarchical aggregation but additionally requires an autoregressive *selection* mechanism to generate foreign-key links (i.e., to route each new child row to appropriate parent rows). Finally, combining the universality of the schema generator (Stage 1) with the completeness of Stages 2–3 gives Theorem E.23.

**From main-text assumptions to proof assumptions.** The main text introduces three principles: schema acyclicity (A4.1), relational Markovian locality (A4.2), and conditional exchangeability (A4.3). The appendix instantiates them as follows:

- **Acyclicity ⇒ explicit generation order.** A4.1 implies that tables can be generated in a topological order, formalized as AE.17 for Stage 2.

- **Locality ⇒ bounded-hop parent sets.** A4.2 becomes (i) cell-level local parent sets for content generation (AE.9, Stage 3) and (ii) row-level local dependence for structural link/latent generation (AE.18, Stage 2).

- **Exchangeability ⇒ shared, permutation-invariant mechanisms.** A4.3 is realized as mechanism sharing by feature type and hierarchy-respecting, permutation-invariant aggregation (AE.11–AE.12, Stage 3) and by table/relation-level sharing for structure generation (AE.19, Stage 2).

In addition, the appendix introduces a small number of technical conditions that are not meant as new domain claims but as *operational regularity assumptions* enabling a tractable hierarchical factorization (e.g., feature-type ordering AE.7 and typewise conditional independence AE.8; deterministic PK encoding AE.16). These conditions specify *how* the high-level

*Table 4.* **DFS configuration sensitivity.** We evaluate RDB-PFN at context size 1024 under different DFS configurations on small relational tasks. **DFS-1** uses 1-hop relational features, while **DFS-2** is the default 2-hop setting. **DFS-2-MaxMinMode** and **DFS-2-MeanCount** keep the 2-hop neighborhood but restrict the aggregation functions to the indicated subsets. The default 2-hop configuration performs best on most tasks, suggesting that both relational depth and aggregation choice affect performance, although the sensitivity is task-dependent.

| DFS Config | Diginetica Ctr | Outbrain Small Ctr | Rel F1 Driver Dnf | Rel F1 Driver Top3 | Rel Hm User Churn | Stackexchange Churn | Stackexchange Upvote | Average |
|---|---|---|---|---|---|---|---|---|
| DFS-1 | 0.5225 | 0.4914 | 0.4955 | 0.5445 | 0.5294 | 0.5592 | 0.8555 | 0.5711 |
| DFS-2 | 0.7004 | 0.5352 | 0.7188 | 0.8115 | 0.6648 | 0.8477 | 0.8527 | 0.7330 |
| DFS-2-MaxMinMode | 0.6029 | 0.5363 | 0.7222 | 0.8148 | 0.6212 | 0.8327 | 0.8447 | 0.7107 |
| DFS-2-MeanCount | 0.6135 | 0.4906 | 0.7195 | 0.8205 | 0.6708 | 0.8419 | 0.8419 | 0.7141 |

principles are realized in our modular generators.

## E.1. Preliminaries

We adopt the schema/instance notation and graph constructions from Section 3, in particular Definitions 3.1–3.3. Note that in Section 3 we used $v \in \mathcal{V}_{in}$ for a row-node; in the appendix we reserve $v = (r, c)$ (Definition E.1) for a *cell instance* and use $r$ for row or row-nodes to avoid clashes.

**Definition E.1** (Cell Instances and Values). For the proofs, we work at the **cell-instance** level. We reserve $v = (r, c)$ to denote a cell instance, where $r \in \mathcal{V}(T)$ is a row and $c \in \mathcal{C}(T)$ is a column of the same table. Let

$$\mathcal{V} := \{(r, c) : T \in \mathcal{T}, \ r \in \mathcal{V}(T), \ c \in \mathcal{C}(T)\}$$

be the set of all cell instances in the database instance. We denote the value stored at cell instance $v = (r, c)$ by the random variable $A_v$.

We also use the canonical projection maps:

$$\text{RowOf}(v) = r, \quad \text{ColumnOf}(v) = c, \quad \text{TableOf}(v) = T \text{ where } T \text{ is the unique table such that } r \in \mathcal{V}(T).$$

**Definition E.2** (Structural vs. Dependent Columns and Cell Instances). Fix a schema $G_S$ and row sets $\{\mathcal{V}(T)\}_{T \in \mathcal{T}}$. Let $\mathcal{V}$ be the set of all cell instances $v = (r, c)$.

We define the *structural column types* as key columns plus any additional connectivity-defining columns:

$$\mathcal{C}^{struct} := \left( \bigcup_{T \in \mathcal{T}} \mathcal{K}^{pk}(T) \right) \cup \left( \bigcup_{T \in \mathcal{T}} \mathcal{K}^{fk}(T) \right) \cup \mathcal{C}^{conn},$$

where $\mathcal{C}^{conn}$ denotes any (optional) non-key columns whose values are used by the structural generator to determine connectivity (and is empty if no such columns are used). And define the corresponding structural cell instances

$$\mathcal{V}^{struct} := \{(r, c) \in \mathcal{V} : c \in \mathcal{C}^{struct}\}.$$

All remaining *feature* columns are *dependent columns*:

$$\mathcal{A}^{dep} := \bigcup_{T \in \mathcal{T}} \mathcal{A}(T), \qquad \mathcal{V}^{dep} := \{(r, c) \in \mathcal{V} : c \in \mathcal{A}^{dep}\}.$$

We denote the realized values on structural cells by $\mathcal{D}_{struct} := \{A_v\}_{v \in \mathcal{V}^{struct}}$ and on dependent cells by $\mathcal{D}_{dep} := \{A_v\}_{v \in \mathcal{V}^{dep}}$.

**Definition E.3** (Structural Latent States). In addition to key columns, Stage 2 may generate a latent state $Z_r \in \mathbb{R}^d$ for each row-node $r \in \mathcal{V}_{in}$, representing structural characteristics used for subsequent content generation. We denote the collection of all latent row states by $\mathcal{Z} := \{Z_r\}_{r \in \mathcal{V}_{in}}$.

## E.2. Single-Table Completeness

We begin with the foundational case of a single table. Let the database contain only one table $T$ with columns $\mathcal{C}(T) = \{c_1, \ldots, c_m\}$. Let $\mathcal{V}(T)$ denote its (finite) set of rows. For any row $r \in \mathcal{V}(T)$ and column $c \in \mathcal{C}(T)$, the corresponding cell instance is $v = (r, c)$ and its value is denoted by $A_v$ (Definition E.1).

To describe the data-generating process, we introduce a *generic row* random vector

$$\mathbf{A} := (A_1, \ldots, A_m), \qquad \text{where } A_k \text{ represents the value of column } c_k \text{ in a random row.}$$

Thus, a realized row $r$ corresponds to one sample $\mathbf{a}_r = (a_{r,1}, \ldots, a_{r,m})$ from the joint distribution $P(\mathbf{A})$. Equivalently, for each column $c_k \in \mathcal{C}(T)$ we have $A_{(r,c_k)} = a_{r,k}$, and $A_k$ denotes the random value of column $c_k$ in a generic row.

**Assumption E.4** (Row i.i.d.). Rows in table $T$ are independent and identically distributed draws from a fixed row distribution:

$$\mathbf{a}_r \sim_{i.i.d.} P(\mathbf{A}) \qquad \text{for all } r \in \mathcal{V}(T).$$

**Justification** Assumption E.4 formalizes the standard single-table modeling view used throughout classical statistics and most tabular learning benchmarks: a table is treated as a multiset of records drawn from a common population distribution. This perspective is also implicit in early tabular foundation model constructions such as prior-data fitted networks (e.g., TabPFNv1, TabICL), where a *dataset-level* generative mechanism is sampled once (e.g., an SCM with parameters $\theta_{\text{SCM}}$), and then individual rows are generated independently by drawing fresh exogenous noise for each row. In this view, the table exhibits a *global mechanism* shared across all rows within the dataset, while randomness arises from per-row noise, making rows exchangeable and (under the simplest setting) i.i.d. We emphasize that this i.i.d. assumption is a *baseline* used to build intuition and establish a clean completeness result. Many real-world tables deviate from i.i.d. sampling due to temporal ordering, distribution shift, repeated measurements, or group-level correlations, and are already attempted to be handled by later work (e.g., TabPFNv2). However, these extensions are orthogonal to the core message here: in the simplest and most widely used single-table setting, an SCM-style shared mechanism with independent per-row noise yields the i.i.d. formulation captured by Assumption E.4.

**Lemma E.5** (Universal measurable conditional sampler + approximation transfer). *Let $(X, Y)$ be random variables with $Y$ taking values in a standard Borel space (e.g., $\mathbb{R}^d$, any countable set, any finite set, or a product of these). Then there exists a Borel measurable function $f$ and an independent $U \sim \text{Unif}(0, 1)$ such that*

$$Y \overset{d}{=} f(X, U).$$

*Moreover, if $\hat{f}_n$ is any sequence of measurable functions such that $\hat{f}_n(X, U) \to f(X, U)$ in probability under the joint law of $(X, U)$, then*

$$\hat{f}_n(X, U) \Rightarrow f(X, U).$$

*Proof sketch.* The first statement is a standard randomization lemma for regular conditional distributions on standard Borel spaces: sample $Y$ by applying a measurable map to $(X, U)$. The second statement follows from the Continuous Mapping Theorem / Slutsky-style arguments on pushforward measures: convergence in probability of the outputs implies weak convergence of the output laws. □

**Theorem E.6** (Single-Table SCM Completeness). *Fix any joint distribution $P(A_1, \ldots, A_m)$ satisfying Assumption E.4, and fix any ordering $(A_1, \ldots, A_m)$. Assume each $A_k$ takes values in a standard Borel space (so that a conditional CDF and generalized inverse are well-defined; e.g., discrete sets or subsets of $\mathbb{R}$). Then there exist measurable functions*

$$f_k : \text{val}(A_{<k}) \times [0, 1] \to \text{val}(A_k), \qquad k = 1, \ldots, m,$$

*and independent noise variables $U_k \sim \text{Unif}(0, 1)$ such that the SCM*

$$A_k = f_k(A_{<k}, U_k), \qquad k = 1, \ldots, m,$$

*induces exactly the joint distribution $P(A_1, \ldots, A_m)$.*

*Moreover, if $\hat{f}_k$ are chosen from universal approximator classes (for the relevant notion of approximation under the input distribution), then the SCM with $\hat{f}_k$ can approximate $P(A_1, \ldots, A_m)$ to arbitrary precision in distribution.*

*Proof sketch.* Fix an ordering $(A_1, \ldots, A_m)$. By the chain rule,

$$P(A_1, \ldots, A_m) = \prod_{k=1}^{m} P(A_k \mid A_{<k}).$$

Apply Lemma E.5 to each conditional: for every $k$ there exists a measurable $f_k$ and independent $U_k \sim \text{Unif}(0, 1)$ such that

$$\text{A}_k \stackrel{d}{=} f_k(\text{A}_{<k}, U_k).$$

Composing these samplers yields an SCM that reproduces the target joint distribution.

For approximation, choose $\hat{f}_k$ from a universal approximator class so that $\hat{f}_k(\text{A}_{<k}, U_k) \to f_k(\text{A}_{<k}, U_k)$ in probability under the induced input law. Iterating Lemma E.5 (approximation transfer) over $k = 1, \ldots, m$ implies that the induced joint distribution converges to $P(\text{A}_1, \ldots, \text{A}_m)$. □

## E.3. Multi-Table Completeness

We now extend the single-table analysis to a full relational database with multiple tables. Our goal is to characterize and approximate the joint distribution of all cell values in an RDB.

In the single-table setting, the set of variables (columns) is fixed. In the relational setting, however, the set of variables and their allowed link structure depend on the schema. Formally, the schema graph $G_S$ (Definition 3.3) specifies which tables exist and which foreign-key relations are permitted. Given $G_S$ and the row sets $\{\mathcal{V}(T)\}_{T \in \mathcal{T}}$, the set of cell instances $\mathcal{V} = \{(r, c) : r \in \mathcal{V}(T), c \in \mathcal{C}(T)\}$ is well-defined (Definition E.1). We therefore consider the conditional joint distribution over all cell values,

$$P(\{A_v\}_{v \in \mathcal{V}} \mid G_S).$$

**Three-stage decomposition.** We instantiate the same *Schema → Structure → Content* decomposition as in the main text (Eq. equation 1), and adopt consistent stage numbering:

- **Stage 1 (Schema):** sample a schema graph $G_S$.

- **Stage 2 (Structure):** generate the *structural skeleton* of the instance, including key columns $\mathcal{D}_{struct} = \{A_v\}_{v \in \mathcal{V}^{struct}}$ (Definition E.2) and, optionally, latent row states $\mathcal{Z} = \{Z_r\}_{r \in \mathcal{V}_{in}}$ (Definition E.3). The realized structure induces (or equivalently includes) the instance graph $G_{in}$.

- **Stage 3 (Content):** generate all remaining feature values $\mathcal{D}_{dep} = \{A_v\}_{v \in \mathcal{V}^{dep}}$ conditioned on the realized structure (and $G_S$).

Accordingly, by the chain rule we write the database distribution as

$$P(\mathcal{D}) = \underbrace{P(G_S)}_{\text{Stage 1: Schema}} \cdot \underbrace{P(\mathcal{D}_{struct}, \mathcal{Z} \mid G_S)}_{\text{Stage 2: Structure}} \cdot \underbrace{P(\mathcal{D}_{dep} \mid \mathcal{D}_{struct}, \mathcal{Z}, G_S)}_{\text{Stage 3: Content}}.$$

**Stage 1 (schema) as a modular prior.** We treat $P(G_S)$ as an external schema prior (hand-designed or learned). Since our focus is on instance generation conditioned on $G_S$, we do not further analyze Stage 1 beyond assuming it can represent distributions over finite schema DAGs.

**What remains to prove.** We focus on the completeness of Stage 3 (Content) and Stage 2 (Structure). Following the roadmap stated earlier, we prove Stage 3 first because it reduces to approximating local conditional mechanisms with structured aggregation over relational neighborhoods; we then reuse this result inside the Stage 2 proof, which introduces an additional selection mechanism for generating foreign-key links.

### E.3.1. STAGE 3: CONDITIONAL FEATURE GENERATION (CONTENT COMPLETION)

**Setting.** Fix a schema $G_S$. Suppose Stage 2 has generated the structural skeleton $\mathcal{D}_{struct} = \{A_v\}_{v \in \mathcal{V}^{struct}}$ (Definition E.2) and latent row states $\mathcal{Z} = \{Z_r\}_{r \in \mathcal{V}_{in}}$ (Definition E.3). Given $(G_S, \mathcal{D}_{struct})$, the instance graph $G_{in}$ is induced by referential integrity (Definition 3.3). Stage 3 models the conditional distribution of all dependent feature values:

$$P(\{A_v\}_{v \in \mathcal{V}^{dep}} \mid \mathcal{D}_{struct}, \mathcal{Z}, G_S).$$

For brevity, let

$$C_2 := (\mathcal{D}_{struct}, \mathcal{Z}, G_S) \quad \text{(equivalently, } C_2 = (G_{in}, \mathcal{Z}, G_S)\text{)}.$$

We use the dependent feature types $\mathcal{A}^{dep}$ and dependent cell instances $\mathcal{V}^{dep}$ as defined in Definition E.2. For each feature type $a \in \mathcal{A}^{dep}$ define its instance set

$$\mathcal{V}_a^{dep} := \{v \in \mathcal{V}^{dep} : \text{ColumnOf}(v) = a\}.$$

A naive autoregressive factorization over all $v \in \mathcal{V}^{dep}$ is generally intractable. We therefore introduce a hierarchical factorization based on feature types, with locality defined relative to the fixed instance graph $G_{in}$ and the structural states $\mathcal{Z}$.

**Assumption E.7** (Feature-Type Ordering). There exists a global topological ordering $\mathcal{O} = (a_1, \dots, a_M)$ of all dependent feature types $\mathcal{A}^{dep}$.

**Assumption E.8** (Typewise Conditional Independence). Given $C_2$ and all dependent features from preceding types $\{A_u\}_{u \in \mathcal{V}_{<a}^{dep}}$ (where $a' \prec a$ denotes precedence in the global order $\mathcal{O}$, and $\mathcal{V}_{<a}^{dep} := \bigcup_{a' \prec a} \mathcal{V}_{a'}^{dep}$), the dependent feature instances of the current type $a$ are conditionally independent:

$$P\Big(\{A_v\}_{v \in \mathcal{V}_a^{dep}} \mid C_2, \{A_u\}_{u \in \mathcal{V}_{<a}^{dep}}\Big) = \prod_{v \in \mathcal{V}_a^{dep}} P\Big(A_v \mid C_2, \{A_u\}_{u \in \mathcal{V}_{<a}^{dep}}\Big).$$

**Assumption E.9** (Structured Local Dependency). For each dependent cell instance $v \in \mathcal{V}^{dep}$, there exists a parent set $Pa(v) \subseteq \mathcal{V}^{struct} \cup \mathcal{V}_{<\text{ColumnOf}(v)}^{dep}$ such that

$$P\Big(A_v \mid C_2, \{A_u\}_{u \in \mathcal{V}_{<\text{ColumnOf}(v)}^{dep}}\Big) = P\Big(A_v \mid \{A_u\}_{u \in Pa(v)}, Z_{\text{RowOf}(v)}\Big).$$

Moreover, $Pa(v)$ is relationally local: every $u \in Pa(v)$ lies either in the same row as $v$ or in a row within $k$ hops of $\text{RowOf}(v)$ in $G_{in}$ (for a fixed constant $k$). Here $\mathcal{V}_{<\text{ColumnOf}(v)}^{dep}$ denotes the union of dependent cell instances whose *feature types* precede $\text{ColumnOf}(v)$ in the global order $\mathcal{O}$.

**Corollary E.10** (Hierarchical Factorization for Stage 3). *Under Assumptions E.7–E.9, the Stage 3 conditional distribution factorizes as*

$$P(\{A_v\}_{v \in \mathcal{V}^{dep}} \mid C_2) = \prod_{a \in \mathcal{O}} \prod_{v \in \mathcal{V}_a^{dep}} P\Big(A_v \mid \{A_u\}_{u \in Pa(v)}, Z_{\text{RowOf}(v)}\Big).$$

*Proof.* By the chain rule applied over the feature-type order $\mathcal{O}$ (Assumption E.7),

$$P(\{A_v\}_{v \in \mathcal{V}^{dep}} \mid C_2) = \prod_{a \in \mathcal{O}} P\Big(\{A_v\}_{v \in \mathcal{V}_a^{dep}} \,\Big|\, C_2, \{A_u\}_{u \in \mathcal{V}_{<a}^{dep}}\Big).$$

By typewise conditional independence (Assumption E.8),

$$P\Big(\{A_v\}_{v \in \mathcal{V}_a^{dep}} \,\Big|\, C_2, \{A_u\}_{u \in \mathcal{V}_{<a}^{dep}}\Big) = \prod_{v \in \mathcal{V}_a^{dep}} P\Big(A_v \,\Big|\, C_2, \{A_u\}_{u \in \mathcal{V}_{<a}^{dep}}\Big).$$

Finally, by structured local dependency (Assumption E.9), for each $v \in \mathcal{V}_a^{dep}$ we have

$$P\Big(A_v \,\Big|\, C_2, \{A_u\}_{u \in \mathcal{V}_{<a}^{dep}}\Big) = P\Big(A_v \,\Big|\, \{A_u\}_{u \in Pa(v)}, Z_{\text{RowOf}(v)}\Big),$$

which yields the stated factorization. $\square$

**Assumption E.11** (Mechanism Sharing by Type). All dependent feature instances of the same type share an identical conditional mechanism. That is, for each $a \in \mathcal{A}^{dep}$, there exists a conditional distribution $K_a$ such that

$$P\big(A_v \mid \{A_u\}_{u \in Pa(v)}, Z_{\text{RowOf}(v)}\big) = K_a\big(A_v \mid \{A_u\}_{u \in Pa(v)}, Z_{\text{RowOf}(v)}\big) \qquad \forall v \in \mathcal{V}_a^{dep}.$$

**Assumption E.12** (Hierarchical Parent Processing)**.** The mechanism $K_a$ must process the parent set $Pa(v)$ in a way that respects the RDB hierarchy. Let $R_v := \{\text{RowOf}(u) : u \in Pa(v)\}$ be the set of unique parent rows and $T_v := \{\text{TableOf}(u) : u \in Pa(v)\}$ the set of unique parent tables.

- **(a) Intra-row coherence (ordered).** For each parent row $r \in R_v$, the parent cells from that row, $\{u \in Pa(v) : \text{RowOf}(u) = r\}$, must be processed in a fixed canonical order. The row latent $Z_r$ may be injected as an additional "row token" in this ordered processing.

- **(b) Inter-row invariance (unordered).** Within each parent table $T \in T_v$, the mechanism must be permutation-invariant to the order of row-level representations coming from rows $r \in \mathcal{V}(T) \cap R_v$.

- **(c) Inter-table coherence (ordered).** The mechanism must process the table-level representations for tables in $T_v$ in a fixed canonical order (e.g., a fixed schema order or a topological order induced by $G_S$).

**Corollary E.13** (Hierarchical SCM Architecture with Latents)**.** *Under Assumptions E.11 and E.12, each mechanism $K_a$ can be realized by an SCM $f_a$ that computes $A_v$ via bottom-up hierarchical aggregation over $Pa(v)$, with row-latent injection:*

$$A_v = f_a(Pa(v), Z_{\text{RowOf}(v)}, U_v) = f'_a(z_v, U_v), \tag{2}$$

$$z_r = h^{\text{row}}\left(\text{Concat}\left(Z_r, [A_u : u \in Pa(v), \text{RowOf}(u) = r]\right)\right), \tag{3}$$

$$z_T = \text{Agg}^{\text{table}}\left(\{z_r : r \in R_v, r \in \mathcal{V}(T)\}\right), \tag{4}$$

$$z_v = h^{\text{global}}\left(\text{Concat}\left(\{z_T : T \in T_v\}, Z_{\text{RowOf}(v)}\right)\right). \tag{5}$$

*Here $h^{\text{row}}$ and $h^{\text{global}}$ are order-sensitive functions (e.g., MLPs over concatenations), and $\text{Agg}^{\text{table}}$ is permutation-invariant (e.g., Deep Sets).*

*Proof.* This is constructive. Eq. equation 3 enforces ordered intra-row processing (AE.12a) and allows injecting the row latent $Z_r$ as additional row-level context. Eq. equation 4 aggregates row representations within each table in a permutation-invariant manner (AE.12b). Eq. equation 5 processes table representations in canonical order (AE.12c) and can additionally condition on the target row latent $Z_{\text{RowOf}(v)}$. □

**Theorem E.14** (Completeness of Stage 3 (Content Completion))**.** *Let $P(\{A_v\}_{v \in \mathcal{V}^{dep}} \mid C_2)$ satisfy Assumptions E.7–E.12. Then there exist hierarchical SCMs $\{\hat{f}_a\}_{a \in \mathcal{A}^{dep}}$ of the form in Corollary E.13 and independent noises $U_v \sim \text{Unif}(0, 1)$ such that*

$$A_v = \hat{f}_{\text{ColumnOf}(v)}(Pa(v), Z_{\text{RowOf}(v)}, U_v), \qquad v \in \mathcal{V}^{dep},$$

*approximates $P(\{A_v\}_{v \in \mathcal{V}^{dep}} \mid C_2)$ arbitrarily well in distribution.*

*Proof.* **Step 1 (Reduction to local mechanisms).** By Corollary E.10, it suffices to approximate each local conditional kernel

$$K_a\left(A_v \mid \{A_u\}_{u \in Pa(v)}, Z_{\text{RowOf}(v)}\right) \qquad (v \in \mathcal{V}_a^{dep}).$$

**Step 2 (Sampling representation + approximation transfer).** Fix a type $a$ and a cell instance $v \in \mathcal{V}_a^{dep}$. Let

$$X_v := \left(\{A_u\}_{u \in Pa(v)}, Z_{\text{RowOf}(v)}\right), \qquad Y_v := A_v.$$

By Lemma E.5, there exists a measurable sampler $f_a$ and an independent $U_v \sim \text{Unif}(0, 1)$ such that

$$A_v \overset{d}{=} f_a(X_v, U_v).$$

Moreover, if $\hat{f}_a(X_v, U_v) \to f_a(X_v, U_v)$ in probability under the induced input law of $X_v$, then $\hat{f}_a(X_v, U_v) \Rightarrow f_a(X_v, U_v)$.

**Step 3 (Existence of $\hat{f}_a$ within the hierarchical SCM class).** Assumptions E.11–E.12 restrict $f_a$ to be *hierarchy-respecting* in its dependence on $Pa(v)$: ordered within each parent row, permutation-invariant over parent rows within a table, and ordered across tables. The architecture in Corollary E.13 is universal for this function class because: (i) $h^{\text{row}}$ and $h^{\text{global}}$ are chosen from universal approximator classes for order-sensitive maps, and (ii) $\text{Agg}^{\text{table}}$ is chosen from

a universal permutation-invariant set-function class. Therefore, there exists a sequence $\hat{f}_a$ in this architecture such that $\hat{f}_a(X_v, U_v) \to f_a(X_v, U_v)$ in probability under the induced input law.

**Step 4 (Joint convergence of Stage 3).** Approximating each factor in the product of Corollary E.10 and using independence of $\{U_v\}$ yields convergence of the induced joint Stage 3 conditional distribution. □

**Connection to the practical GNN decoder.** The Stage 3 proof is stated in terms of a hierarchy-respecting conditional mechanism $K_a$. In the main model, we implement this mechanism by a bidirectional relational GNN over $G_{in}$ with a shared decoder. This choice is compatible with the proof: message-passing layers compute permutation-equivariant summaries of $k$-hop neighborhoods, and the decoder realizes the final conditional sampling step.

**Proposition E.15** (GNN as an instantiation of Stage 3 mechanisms). *For any fixed hop radius $k$, a $k$-layer relational message passing network with permutation-invariant aggregation can* approximate *the bounded-hop, permutation-equivariant neighborhood summaries required by Corollary E.13, under standard expressivity conditions on the per-relation MLPs and an injective set aggregator.*

*Proof sketch.* Both constructions compute row-wise representations by composing (i) order-sensitive intra-row processing and (ii) permutation-invariant aggregation across sets of neighbors, repeated over a bounded-hop neighborhood. Universal approximation of the constituent MLPs yields approximation of the resulting neighborhood-to-representation map, and the decoder then approximates the conditional output distribution using standard inverse-transform/categorical sampling. □

### E.3.2. STAGE 2: STRUCTURAL GENERATION (KEYS, LINKS, AND LATENT STATES)

**Setting.** Fix a schema $G_S$. Stage 2 models $P(\mathcal{D}_{struct}, \mathcal{Z} \mid G_S)$, where $\mathcal{D}_{struct}$ are structural cells (Definition E.2) and $\mathcal{Z} = \{Z_r\}_{r \in \mathcal{V}_{in}}$ are optional row latents (Definition E.3). For convenience we denote the key-cell subsets

$$\mathcal{V}^{pk} := \{(r,c) \in \mathcal{V} : c \in \bigcup_{T \in \mathcal{T}} \mathcal{K}^{pk}(T)\}, \quad \mathcal{V}^{fk} := \{(r,c) \in \mathcal{V} : c \in \bigcup_{T \in \mathcal{T}} \mathcal{K}^{fk}(T)\}, \quad \mathcal{V}^{conn} := \{(r,c) \in \mathcal{V} : c \in \mathcal{C}^{conn}\},$$

so that $\mathcal{V}^{struct} = \mathcal{V}^{pk} \cup \mathcal{V}^{fk} \cup \mathcal{V}^{conn}$. To keep notation light, the completeness argument below is written for the common case $\mathcal{C}^{conn} = \emptyset$ (i.e., $\mathcal{V}^{conn} = \emptyset$); the extension to $\mathcal{C}^{conn} \neq \emptyset$ follows by including $\mathcal{V}^{conn}$ among the generated structural cells and in the structural context $\mathrm{Ctx}(\cdot)$ defined below.

Stage 3 conditions on a fixed realized structure and models feature values via local parent sets $Pa(v)$ at the cell level. Stage 2, in contrast, must model the *structure itself* (FK links), which naturally introduces (i) a schema-respecting generation order (to satisfy referential integrity) and (ii) possible *competition/degree effects* among parent rows. Accordingly, Stage 2 mirrors Stage 3's five-assumption style (ordering, locality, sharing, structured processing), but replaces Stage 3's within-type conditional independence with a restricted dependence on a permutation-invariant competition summary.

**Assumption E.16** (Deterministic Primary Keys). For each table $T$ with $n_T := |\mathcal{V}(T)|$ rows, primary key values are a deterministic, known injective encoding of row identity. Concretely, there exists a known injective map $\mathrm{PKEnc}_T : \{1, \ldots, n_T\} \to \mathrm{val}(\mathcal{K}^{pk}(T))$ such that the PK cells of the $i$-th row equal $\mathrm{PKEnc}_T(i)$. Hence PK values carry no causal/content information beyond identifying rows.

Under Assumption E.16, Stage 2 reduces to modeling the joint distribution of (i) all foreign-key cells (equivalently, the induced instance graph $G_{in}$), and (ii) the latent row states $\mathcal{Z}$:

$$P(\mathcal{D}_{struct}, \mathcal{Z} \mid G_S) = P(\{A_v\}_{v \in \mathcal{V}^{fk}}, \mathcal{Z} \mid G_S), \quad \text{with} \quad G_{in} = g(G_S, \mathcal{D}_{struct}).$$

For each dependent table $T$, fix a row order $\pi_T = (r_{T,1}, \ldots, r_{T,n_T})$. If $T$ has parent tables $\mathrm{Par}(T) = \{T_p^{(1)}, \ldots, T_p^{(p)}\}$, define for each row $r_{T,k}$ the selected parent-row tuple

$$U_{T,k} = (u_{T,k}^{(1)}, \ldots, u_{T,k}^{(p)}), \qquad u_{T,k}^{(j)} \in \mathcal{V}(T_p^{(j)}).$$

For source tables (no FKs), $U_{T,k}$ is taken to be empty. For a dependent table $T$ with parents $\mathrm{Par}(T) = \{T_p^{(1)}, \ldots, T_p^{(p)}\}$, define the feasible parent-tuple set

$$\mathcal{C}_T := \mathcal{V}(T_p^{(1)}) \times \cdots \times \mathcal{V}(T_p^{(p)}),$$

so that $U_{T,k} \in \mathcal{C}_T$. (For finite instances, $\mathcal{C}_T$ is finite.)

**Assumption E.17** (Table Topological Order). There exists a topological ordering of tables $\text{Topo}(G_S) = (T^{(1)}, \ldots, T^{(N)})$ such that every foreign-key edge in $G_S$ points from an earlier table to a later table.

Assumption E.17 implies a valid procedural generation order: parent tables can be sampled before child tables, and each dependent row selects parent rows and copies their PKs into FK cells. For a source table (no FKs), we generate its rows' latent states from exogenous noise. For a dependent table $T_c$ with parent tables $\text{Par}(T_c) = \{T_p^{(1)}, \ldots, T_p^{(p)}\}$, each new child row selects a tuple of parent rows $(u^{(1)}, \ldots, u^{(p)})$ and sets its FK values accordingly. This selection is the structural "routing" mechanism.

**Assumption E.18** (Structured Local Dependency). Fix a dependent table $T$ with row order $\pi_T = (r_{T,1}, \ldots, r_{T,n_T})$ and parent tables $\text{Par}(T) = \{T_p^{(1)}, \ldots, T_p^{(p)}\}$.

For each step $k$, let $\mathcal{C}_{T,k} \subseteq \mathcal{C}_T$ be a finite candidate-tuple set, where $\mathcal{C}_T = \mathcal{V}(T_p^{(1)}) \times \cdots \times \mathcal{V}(T_p^{(p)})$.

Define:

- **(Pre-selection context parents).** A set $Pa^{ctx}(T, k)$ of structural cell instances drawn only from already-generated tables/rows (tables earlier than $T$, and rows $\{r_{T,j}\}_{j<k}$ in $T$), restricted to a bounded-hop neighborhood (radius $k_0$) in the partially constructed instance graph.

- **(Candidate-local parents).** For each candidate tuple $c \in \mathcal{C}_{T,k}$, a set $Pa^{cand}(c)$ of structural cell instances drawn from the rows in $c$ and their bounded-hop neighborhoods (radius $k_0$) in the partially constructed instance graph (again, only using already-generated variables).

- **(Embeddings).** A context embedding $\text{Ctx}_{T,k} = \Phi_T^{ctx}(\{A_u\}_{u \in Pa^{ctx}(T,k)})$ and candidate embeddings $\phi_{T,k}(c) = \Phi_T^{cand}(\{A_u\}_{u \in Pa^{cand}(c)})$.

Let $S_{T,k-1} = S(\{U_{T,j}\}_{j<k})$ be a permutation-invariant competition/degree summary of previous selections.

**(Markov restriction).** The local conditional depends on the past only through these pre-selection quantities:

$$P(U_{T,k}, Z_{r_{T,k}} \mid \text{all previously generated variables}) = P\Big(U_{T,k}, Z_{r_{T,k}} \,\Big|\, \text{Ctx}_{T,k}, \{\phi_{T,k}(c)\}_{c \in \mathcal{C}_{T,k}}, S_{T,k-1}\Big).$$

For source tables, interpret $U_{T,k}$, $\mathcal{C}_{T,k}$, and $S_{T,k-1}$ as empty and keep only $P(Z_{r_{T,k}} \mid \text{Ctx}_{T,k})$.

**Assumption E.19** (Mechanism Sharing by Table/Relation Pattern). For each table $T$, all rows share the same conditional mechanism for producing $(U_{T,k}, Z_{r_{T,k}})$ from their pre-selection inputs. Concretely, there exists a (table-specific) kernel $K_T$ such that for all $k$,

$$P\Big(U_{T,k}, Z_{r_{T,k}} \,\Big|\, \text{Ctx}_{T,k}, \{\phi_{T,k}(c)\}_{c \in \mathcal{C}_{T,k}}, S_{T,k-1}\Big) = K_T\Big(U_{T,k}, Z_{r_{T,k}} \,\Big|\, \text{Ctx}_{T,k}, \{\phi_{T,k}(c)\}_{c \in \mathcal{C}_{T,k}}, S_{T,k-1}\Big).$$

**Assumption E.20** (Hierarchy-Respecting Processing for Structure). The kernel $K_T$ processes (i) the context parents $Pa^{ctx}(T, k)$ and (ii) each candidate parent set $Pa^{cand}(c)$ in a hierarchy-respecting way:

- **(Row-level order).** Within any row, structural cells (and optional row latents) are processed in a fixed canonical order.

- **(Within-table set invariance).** Within any table, sets of row representations are aggregated permutation-invariantly.

- **(Across-table order).** Tables are processed in a fixed canonical order (e.g., schema/topological order).

- **(Candidate-tuple role order).** Within a candidate tuple $c = (u^{(1)}, \ldots, u^{(p)})$, the $p$ parent roles are processed in the canonical FK-role order induced by the schema (so the tuple embedding is sensitive to role).

- **(Candidate-set symmetry).** The dependence of $K_T$ on the candidate set $\{\phi_{T,k}(c)\}_{c \in \mathcal{C}_{T,k}}$ is permutation-equivariant with respect to reordering candidates (e.g., implemented by scoring each candidate with a shared function and normalizing).

**Corollary E.21** (Stage 2 Factorization (Structure)). *Under Assumptions E.16, E.17, E.18, and E.19, the Stage 2 distribution admits an autoregressive factorization over tables and (within each table) rows:*

$$P(\{A_v\}_{v \in \mathcal{V}^{fk}}, \mathcal{Z} \mid G_S) = \prod_{T \in \text{Topo}(G_S)} \prod_{k=1}^{n_T} P\Big(Z_{r_{T,k}}, U_{T,k} \,\Big|\, \text{Ctx}_{T,k}, \{\phi_{T,k}(c)\}_{c \in \mathcal{C}_{T,k}}, S_{T,k-1}\Big),$$

*where $U_{T,k}$, $\mathcal{C}_{T,k}$ and $S_{T,k-1}$ are empty for source tables.*

*Proof.* Fix a topological table order $\mathrm{Topo}(G_S) = (T^{(1)}, \ldots, T^{(N)})$ (Assumption E.17). For each table $T^{(i)}$, fix a row order $\pi_{T^{(i)}} = (r_{T^{(i)},1}, \ldots, r_{T^{(i)},n_i})$. Under Assumption E.16, primary keys are deterministic, so the Stage 2 randomness is fully captured by

$$Y_{i,k} \;:=\; (U_{T^{(i)},k},\, Z_{r_{T^{(i)},k}}),$$

where $U_{T^{(i)},k}$ is empty for source tables.

**Step 1 (Repeated chain rule over the nested order).** Applying the chain rule repeatedly over the nested order "tables then rows" gives

$$P(\{A_v\}_{v \in \mathcal{V}^{fk}}, \mathcal{Z} \mid G_S) = P(\{Y_{i,k}\}_{i,k} \mid G_S) = \prod_{i=1}^{N} \prod_{k=1}^{n_i} P(Y_{i,k} \mid G_S,\, \text{all variables generated before } (i,k))\,.$$

**Step 2 (Apply the candidate-based Markov restriction).** By Assumption E.18, for each dependent-table step $(i,k)$ the conditional distribution of $Y_{i,k} = (U_{T^{(i)},k}, Z_{r_{T^{(i)},k}})$ given all previously generated variables depends on the past only through: (i) the pre-selection context embedding $\mathrm{Ctx}_{T^{(i)},k}$, (ii) the candidate set $\mathcal{C}_{T^{(i)},k}$ and its candidate embeddings $\{\phi_{T^{(i)},k}(c)\}_{c \in \mathcal{C}_{T^{(i)},k}}$, and (iii) the within-table competition summary $S_{T^{(i)},k-1} = S(\{U_{T^{(i)},j}\}_{j<k})$. Hence,

$$P(Y_{i,k} \mid G_S,\, \text{all variables generated before } (i,k)) = P\Big(Y_{i,k} \,\Big|\, \mathrm{Ctx}_{T^{(i)},k}, \{\phi_{T^{(i)},k}(c)\}_{c \in \mathcal{C}_{T^{(i)},k}}, S_{T^{(i)},k-1}\Big),$$

with the convention that for source tables $U_{T^{(i)},k}$, $\mathcal{C}_{T^{(i)},k}$, and $S_{T^{(i)},k-1}$ are empty, so the right-hand side reduces to conditioning on $\mathrm{Ctx}_{T^{(i)},k}$ only.

**Step 3 (Substitute into the chain rule product).** Substituting the above identity into the product from Step 1 yields

$$P(\{A_v\}_{v \in \mathcal{V}^{fk}}, \mathcal{Z} \mid G_S) = \prod_{i=1}^{N} \prod_{k=1}^{n_i} P\Big(Y_{i,k} \,\Big|\, \mathrm{Ctx}_{T^{(i)},k}, \{\phi_{T^{(i)},k}(c)\}_{c \in \mathcal{C}_{T^{(i)},k}}, S_{T^{(i)},k-1}\Big),$$

which is exactly the claimed factorization after re-indexing $(i,k)$ back to $(T,k)$. $\square$

**A universal selective SCM.** We now give a constructive parameterization that can realize the Stage 2 factorization (Corollary E.21) while respecting Assumption E.20. Fix a dependent table $T$ and row step $k$ with candidate set $\mathcal{C}_{T,k} \subseteq \mathcal{C}_T$ (in the idealized proof one may take $\mathcal{C}_{T,k} = \mathcal{C}_T$).

For each row $r_{T,k}$ we: (i) sample an initial latent $Z_{r_{T,k}}^{(0)}$ from exogenous noise; (ii) compute a hierarchy-respecting *pre-selection* context embedding

$$z_{T,k}^{ctx} := \mathrm{Ctx}_{T,k} \;=\; \Phi_T^{ctx}\big(\{A_u\}_{u \in Pa^{ctx}(T,k)}\big)\,;$$

(iii) compute candidate-tuple embeddings for each $c \in \mathcal{C}_{T,k}$ by

$$\phi_{T,k}(c) \;=\; \Phi_T^{cand}\big(\{A_u\}_{u \in Pa^{cand}(c)}\big)\,;$$

(iv) score and sample a parent tuple $U_{T,k} \in \mathcal{C}_{T,k}$ using a shared scoring function $g$ and a uniform selector:

$$\ell_c = g\Big(Z_{r_{T,k}}^{(0)},\, z_{T,k}^{ctx},\, \phi_{T,k}(c),\, S_{T,k-1}\Big), \qquad c \in \mathcal{C}_{T,k},$$

$$U_{T,k} = \mathrm{Sample}\big(\mathrm{Softmax}(\{\ell_c\}_{c \in \mathcal{C}_{T,k}}),\, U_{T,k}^{sel}\big), \quad U_{T,k}^{sel} \sim \mathrm{Unif}(0,1);$$

we then set the FK cell values of $r_{T,k}$ to match the selected parent rows' PKs (referential integrity); and (v) update the latent state via a universal map $\psi$:

$$Z_{r_{T,k}} = \psi\Big(Z_{r_{T,k}}^{(0)},\, z_{T,k}^{ctx},\, \phi_{T,k}(U_{T,k}),\, U_{T,k}^{lat}\Big), \quad U_{T,k}^{lat} \sim \mathrm{Unif}(0,1),$$

where $U_{T,k}^{lat}$ is independent noise.

For a source table $T$ (no FKs), the mechanism omits the selection step and outputs

$$Z_{r_{T,k}} = \psi_T^{src}\big(\Phi_T^{ctx}(\{A_u\}_{u \in Pa^{ctx}(T,k)}),\, U_{T,k}^{lat}\big), \quad U_{T,k}^{lat} \sim \mathrm{Unif}(0,1),$$

with a table-shared map $\psi_T^{src}$.

**Theorem E.22** (Completeness of Stage 2 (Structure))**.** *Fix a schema $G_S$. Let $P(\mathcal{D}_{struct}, \mathcal{Z} \mid G_S)$ be any structural distribution satisfying Assumptions E.16, E.17, E.18, E.19, E.20. Then there exists a parameter setting of the above selective SCM such that the induced distribution over $(\mathcal{D}_{struct}, \mathcal{Z})$ approximates $P(\mathcal{D}_{struct}, \mathcal{Z} \mid G_S)$ arbitrarily well in distribution.*

*Proof.* **Step 1 (Reduction to local kernels).** By Corollary E.21 (candidate-based factorization), it suffices to approximate each local conditional

$$P\Big(Z_{r_{T,k}}, U_{T,k} \;\Big|\; \mathrm{Ctx}_{T,k}, \{\phi_{T,k}(c)\}_{c \in \mathcal{C}_{T,k}}, S_{T,k-1}\Big),$$

with the convention that for source tables $U_{T,k}$, $\mathcal{C}_{T,k}$, and $S_{T,k-1}$ are empty.

**Step 2 (Sampling representation + approximation transfer).** Fix a dependent table $T$ and step $k$. Define the (pre-selection) input

$$X_{T,k} := \Big(\mathrm{Ctx}_{T,k}, \{\phi_{T,k}(c)\}_{c \in \mathcal{C}_{T,k}}, S_{T,k-1}\Big), \qquad Y_{T,k} := (U_{T,k}, Z_{r_{T,k}}).$$

Since $\mathcal{C}_{T,k}$ is finite and $Z_{r_{T,k}} \in \mathbb{R}^d$, the product space $\mathcal{C}_{T,k} \times \mathbb{R}^d$ is standard Borel. By Lemma E.5, there exists a measurable sampler $f_T$ and an independent $U_{T,k}^{(gen)} \sim \mathrm{Unif}(0,1)$ such that

$$(U_{T,k}, Z_{r_{T,k}}) \stackrel{d}{=} f_T(X_{T,k}, U_{T,k}^{(gen)}).$$

Thus, if our model class contains $\hat{f}_T$ with $\hat{f}_T(X_{T,k}, U_{T,k}^{(gen)}) \to f_T(X_{T,k}, U_{T,k}^{(gen)})$ in probability under the induced input law, then the induced conditional law converges in distribution.

**Step 3 (Realizability by the structural SCM parameterization).** Write the target local kernel as

$$P(U_{T,k}, Z_{r_{T,k}} \mid X_{T,k}) = P(U_{T,k} \mid X_{T,k}) \cdot P(Z_{r_{T,k}} \mid X_{T,k}, U_{T,k}).$$

Because $\mathcal{C}_{T,k}$ is finite, any categorical distribution $P(U_{T,k} \mid X_{T,k})$ can be represented by logits $\{\ell_c(X_{T,k})\}_{c \in \mathcal{C}_{T,k}}$ via

$$P(U_{T,k} = c \mid X_{T,k}) = \mathrm{Softmax}(\{\ell_{c'}(X_{T,k})\}_{c' \in \mathcal{C}_{T,k}})_c,$$

e.g., by taking $\ell_c(X_{T,k}) = \log P(U_{T,k} = c \mid X_{T,k})$ (up to an additive constant). Choosing the scoring network $g$ from a universal approximator class over its inputs and using the candidate embeddings $\phi_{T,k}(c)$ yields arbitrarily accurate approximation of these logits under the induced input law.

Conditioned on $(X_{T,k}, U_{T,k})$, the latent distribution $P(Z_{r_{T,k}} \mid X_{T,k}, U_{T,k})$ admits a measurable sampler from uniform noise by Lemma E.5. Choosing the update map $\psi$ from a universal approximator class yields an arbitrarily accurate approximation of this sampler in probability under the induced input law.

Finally, Assumption E.20 restricts dependence on the structural information to hierarchy-respecting processing. Choosing $\Phi_T^{ctx}$ and $\Phi_T^{cand}$ from universal hierarchy-respecting function classes yields arbitrarily accurate approximations of $\mathrm{Ctx}_{T,k}$ and $\phi_{T,k}(c)$ as required.

**Step 4 (Conclude).** Combining (i) approximation of the categorical selection kernel for $U_{T,k}$ and (ii) approximation of the conditional sampler for $Z_{r_{T,k}}$ yields a sequence of structural SCM parameters whose induced local conditionals converge in distribution. Substituting these approximations into the Stage 2 factorization and using independent per-step noise variables gives convergence of the induced Stage 2 distribution to the target. □

E.3.3. UNIVERSALITY OF THE THREE-STAGE CONSTRUCTION

**Theorem E.23** (Universality of the three-stage construction)**.** *Let $\mathcal{P}_{RDB}$ be the family of relational database distributions $P(\mathcal{D})$ that factorize as*

$$P(\mathcal{D}) = P(G_S)\, P(\mathcal{D}_{struct}, \mathcal{Z} \mid G_S)\, P(\mathcal{D}_{dep} \mid \mathcal{D}_{struct}, \mathcal{Z}, G_S),$$

*and satisfy the Stage 2 assumptions and Stage 3 assumptions stated above.*

*Assume: (i) the schema model family $\{\hat{P}_{\theta_S}(G_S)\}$ is dense in distributions over finite schema DAGs; (ii) for each fixed schema $G_S$, the Stage 2 family $\{\hat{P}_{\theta_2}(\mathcal{D}_{struct}, \mathcal{Z} \mid G_S)\}$ is dense in the admissible $P(\mathcal{D}_{struct}, \mathcal{Z} \mid G_S)$; and (iii) for each fixed $(G_S, \mathcal{D}_{struct}, \mathcal{Z})$, the Stage 3 family $\{\hat{P}_{\theta_3}(\mathcal{D}_{dep} \mid \mathcal{D}_{struct}, \mathcal{Z}, G_S)\}$ is dense in the admissible $P(\mathcal{D}_{dep} \mid \mathcal{D}_{struct}, \mathcal{Z}, G_S)$.*

*Then the composite family*

$$\hat{P}_\theta(\mathcal{D}) = \hat{P}_{\theta_S}(G_S)\,\hat{P}_{\theta_2}(\mathcal{D}_{struct}, \mathcal{Z} \mid G_S)\,\hat{P}_{\theta_3}(\mathcal{D}_{dep} \mid \mathcal{D}_{struct}, \mathcal{Z}, G_S)$$

*is dense in* $\mathcal{P}_{RDB}$ *(in distribution).*

*Proof sketch.* Fix any target $P \in \mathcal{P}_{RDB}$ and $\varepsilon > 0$.

Choose $\theta_S$ so that $\hat{P}_{\theta_S}(G_S) \Rightarrow P(G_S)$. Next, for each fixed schema $G_S$, choose $\theta_2(G_S)$ so that $\hat{P}_{\theta_2(G_S)}(\mathcal{D}_{struct}, \mathcal{Z} \mid G_S) \Rightarrow P(\mathcal{D}_{struct}, \mathcal{Z} \mid G_S)$. Finally, for each fixed $(G_S, \mathcal{D}_{struct}, \mathcal{Z})$, choose $\theta_3(G_S, \mathcal{D}_{struct}, \mathcal{Z})$ so that $\hat{P}_{\theta_3(G_S, \mathcal{D}_{struct}, \mathcal{Z})}(\mathcal{D}_{dep} \mid \mathcal{D}_{struct}, \mathcal{Z}, G_S) \Rightarrow P(\mathcal{D}_{dep} \mid \mathcal{D}_{struct}, \mathcal{Z}, G_S)$.

The joint law is the iterated mixture

$$P(\mathcal{D}) = \int P(G_S) \int P(\mathcal{D}_{struct}, \mathcal{Z} \mid G_S)\,P(\mathcal{D}_{dep} \mid \mathcal{D}_{struct}, \mathcal{Z}, G_S),$$

and $\hat{P}_\theta(\mathcal{D})$ is defined by the same iterated mixture with each factor replaced by its approximation. Since weak convergence is preserved under forming such mixtures on standard Borel spaces, $\hat{P}_\theta(\mathcal{D}) \Rightarrow P(\mathcal{D})$. $\qquad\square$

# F. Raw Results

For completeness, we provide raw results for the additional analyses. Table 5 include raw results of the efficiency figure 2. Tables 6–10 report ROC-AUC across context sizes, corresponding to main figure results. Tables 12–16 report PR-AUC across context sizes. Table 17 reports target-task and cross-task fine-tuning. Tables 18–20 report chunked-support scaling. Tables 21–25 and Tables 26–30 report relational-prior and backbone ablations, respectively.

*Table 5.* Efficiency and complexity comparison of baseline models. Total inference time is calculated as the cumulative time required to evaluate all 19 relational benchmark tasks using a fixed 500-shot context ($N = 500$).

| Model | Parameters (M) | Pretraining Data (M) | Inference Time (s) |
|---|---|---|---|
| *Our Model* | | | |
| **RDB-PFN** | **0.7** | **2** | **34** |
| *Tabular Foundation Models (Full / Ensemble)* | | | |
| TabICL v1.1 | 27.1 | 80 | 229 |
| Mitra | 72 | 45 | 164 |
| TabPFN v2.5 | 10.7 | Undisclosed | 156 |
| LimiX 16M | 16 | Undisclosed | 101 |
| *Tabular Foundation Models (Lite / Single-Estimator)* | | | |
| TabICL-Lite | 27.1 | 80 | 36 |
| TabPFN-Lite | 10.7 | Undisclosed | 43 |
| LimiX-Lite | 16 | Undisclosed | 54 |
| *Classical Baselines* | | | |
| AutoGluon | - | - | 172 |
| XGBoost | - | - | 57 |
| Random Forest | - | - | 48 |

*Table 6.* Raw Results for Context Size 64

**Part 1: First 10 Datasets**

| Model | Amazon Churn | Avs Repeater | Diginetica Ctr | Outbrain Small Ctr | Rel Amazon Item Churn | Rel Amazon User Churn | Rel Avito User Clicks | Rel Avito User Visits | Rel Event User Ignore | Rel Event User Repeat |
|---|---|---|---|---|---|---|---|---|---|---|
| Random Forest | 0.6036 | 0.5115 | 0.5792 | 0.5073 | 0.6530 | 0.5628 | 0.5743 | 0.4824 | 0.7326 | 0.6032 |
| XGBoost | 0.5757 | 0.5049 | 0.5000 | 0.5109 | 0.6417 | 0.5505 | 0.4884 | 0.5357 | 0.7317 | 0.6279 |
| XGBoost-Tuned | 0.5871 | 0.5099 | 0.5000 | 0.5046 | 0.6369 | 0.5271 | 0.4922 | 0.5230 | 0.7134 | 0.5968 |
| AutoGluon-Medium | 0.5818 | 0.5059 | 0.5000 | 0.5038 | 0.6516 | 0.5339 | 0.4979 | 0.5254 | 0.7169 | 0.6170 |
| AutoGluon-Best | 0.6176 | 0.5059 | 0.5709 | 0.5003 | 0.6551 | 0.5544 | 0.5090 | 0.5045 | 0.7141 | 0.5711 |
| Mitra | 0.5828 | 0.5143 | 0.6434 | 0.5054 | 0.6876 | 0.5228 | 0.5797 | 0.5041 | 0.7066 | 0.5668 |
| TabPFNv2 | 0.6265 | 0.5168 | 0.6034 | 0.5047 | 0.7412 | 0.5902 | 0.5708 | 0.5042 | 0.7401 | 0.5877 |
| TabPFNv2.5 | 0.5976 | 0.5161 | 0.6105 | 0.5066 | 0.7269 | 0.5973 | 0.5734 | 0.4814 | 0.7548 | 0.5979 |
| TabPFNv2.5-1estimator | 0.5660 | 0.5124 | 0.5859 | 0.5056 | 0.7223 | 0.5723 | 0.5634 | 0.4851 | 0.7350 | 0.6017 |
| TabICLv1 | 0.6468 | 0.5105 | 0.5995 | 0.5066 | 0.7067 | 0.5754 | 0.5641 | 0.5039 | 0.7737 | 0.5486 |
| TabICLv1.1 | 0.6215 | 0.5047 | 0.5707 | 0.5036 | 0.6793 | 0.5468 | 0.5601 | 0.5133 | 0.7219 | 0.5820 |
| TabICLv1.1-1estimator | 0.6276 | 0.4991 | 0.5270 | 0.5036 | 0.6669 | 0.5462 | 0.5642 | 0.5070 | 0.6961 | 0.5609 |
| LimiX2m | 0.6537 | 0.5144 | 0.6073 | 0.5076 | 0.7044 | 0.5699 | 0.5633 | 0.5159 | 0.7342 | 0.5889 |
| LimiX16m | 0.6234 | 0.5092 | 0.6277 | 0.5105 | 0.7243 | 0.5831 | 0.5802 | 0.4756 | 0.7392 | 0.5872 |
| LimiX16m-1estimator | 0.6170 | 0.5067 | 0.6495 | 0.5093 | 0.7112 | 0.5755 | 0.5792 | 0.4806 | 0.7315 | 0.5846 |
| RDBPFN_single_table | 0.6241 | 0.5007 | 0.6604 | 0.5064 | 0.5911 | 0.5374 | 0.5796 | 0.4684 | 0.6513 | 0.5054 |
| RDBPFN | 0.6286 | 0.5172 | 0.6028 | 0.5106 | 0.7006 | 0.5791 | 0.5674 | 0.5061 | 0.7342 | 0.6055 |

**Part 2: Remaining 9 Datasets and Average**

| Model | Rel F1 Driver Dnf | Rel F1 Driver Top3 | Rel Hm User Churn | Rel Stack User Badge | Rel Stack User Engagement | Rel Trial Study Outcome | Retailrocket Cvr | Stackexchange Churn | Stackexchange Upvote | Average |
|---|---|---|---|---|---|---|---|---|---|---|
| Random Forest | 0.7049 | 0.7709 | 0.6087 | 0.7719 | 0.7654 | 0.5101 | 0.6432 | 0.7763 | 0.8276 | 0.6415 |
| XGBoost | 0.6937 | 0.7733 | 0.5890 | 0.6722 | 0.6331 | 0.5293 | 0.5596 | 0.7186 | 0.8045 | 0.6127 |
| XGBoost-Tuned | 0.6788 | 0.7670 | 0.5383 | 0.5584 | 0.5500 | 0.5209 | 0.5541 | 0.6596 | 0.8275 | 0.5919 |
| AutoGluon-Medium | 0.6653 | 0.7445 | 0.5735 | 0.5927 | 0.5383 | 0.5064 | 0.5075 | 0.6903 | 0.8251 | 0.5936 |
| AutoGluon-Best | 0.6689 | 0.7836 | 0.5804 | 0.7148 | 0.6975 | 0.5374 | 0.5445 | 0.7240 | 0.8315 | 0.6201 |
| Mitra | 0.7068 | 0.7793 | 0.5878 | 0.8105 | 0.8015 | 0.5212 | 0.6525 | 0.7858 | 0.8199 | 0.6463 |
| TabPFNv2 | 0.6952 | 0.7836 | 0.6239 | 0.7804 | 0.7645 | 0.5403 | 0.6602 | 0.7819 | 0.8422 | 0.6557 |
| TabPFNv2.5 | 0.6948 | 0.7734 | 0.6221 | 0.7941 | 0.7674 | 0.5418 | 0.6390 | 0.7960 | 0.8424 | 0.6544 |
| TabPFNv2.5-1estimator | 0.6967 | 0.7775 | 0.6091 | 0.7816 | 0.7595 | 0.5328 | 0.6415 | 0.7882 | 0.8387 | 0.6461 |
| TabICLv1 | 0.6909 | 0.7778 | 0.6067 | 0.8083 | 0.7995 | 0.5261 | 0.6715 | 0.7970 | 0.8150 | 0.6541 |
| TabICLv1.1 | 0.7141 | 0.7686 | 0.5893 | 0.8044 | 0.7875 | 0.5299 | 0.6659 | 0.7662 | 0.8151 | 0.6445 |
| TabICLv1.1-1estimator | 0.7151 | 0.7725 | 0.5846 | 0.7955 | 0.7768 | 0.5077 | 0.6369 | 0.7786 | 0.8049 | 0.6353 |
| LimiX2m | 0.6892 | 0.7767 | 0.6007 | 0.7959 | 0.7874 | 0.5371 | 0.6684 | 0.7997 | 0.8184 | 0.6544 |
| LimiX16m | 0.7069 | 0.7667 | 0.6084 | 0.8107 | 0.7890 | 0.5421 | 0.6786 | 0.7903 | 0.8387 | 0.6575 |
| LimiX16m-1estimator | 0.7026 | 0.7784 | 0.5975 | 0.7992 | 0.7790 | 0.5504 | 0.6622 | 0.7758 | 0.8382 | 0.6541 |
| RDBPFN_single_table | 0.6063 | 0.7581 | 0.5618 | 0.7619 | 0.7220 | 0.5212 | 0.6075 | 0.7544 | 0.7569 | 0.6145 |
| RDBPFN | 0.6932 | 0.7952 | 0.6073 | 0.7729 | 0.7599 | 0.5474 | 0.6503 | 0.7669 | 0.8367 | 0.6517 |

*Table 7.* Raw Results for Context Size 128

**Part 1: First 10 Datasets**

| Model | Amazon Churn | Avs Repeater | Diginetica Ctr | Outbrain Small Ctr | Rel Amazon Item Churn | Rel Amazon User Churn | Rel Avito User Clicks | Rel Avito User Visits | Rel Event User Ignore | Rel Event User Repeat |
|---|---|---|---|---|---|---|---|---|---|---|
| Random Forest | 0.6030 | 0.5234 | 0.5400 | 0.5002 | 0.6953 | 0.5816 | 0.5578 | 0.4754 | 0.7445 | 0.5909 |
| XGBoost | 0.5801 | 0.5154 | 0.5486 | 0.4989 | 0.6814 | 0.5657 | 0.4905 | 0.5635 | 0.7731 | 0.6049 |
| XGBoost-Tuned | 0.5708 | 0.5241 | 0.5158 | 0.4987 | 0.6448 | 0.5531 | 0.4951 | 0.5531 | 0.7021 | 0.6132 |
| AutoGluon-Medium | 0.6071 | 0.5137 | 0.4972 | 0.5082 | 0.6693 | 0.5741 | 0.4922 | 0.5477 | 0.7407 | 0.6012 |
| AutoGluon-Best | 0.5866 | 0.5220 | 0.5422 | 0.5021 | 0.6888 | 0.5758 | 0.5066 | 0.5497 | 0.7545 | 0.6367 |
| Mitra | 0.6421 | 0.5323 | 0.6481 | 0.4933 | 0.7274 | 0.5842 | 0.5576 | 0.5496 | 0.7209 | 0.6203 |
| TabPFNv2 | 0.6609 | 0.5304 | 0.5929 | 0.5043 | 0.7563 | 0.6261 | 0.5782 | 0.5264 | 0.7753 | 0.6274 |
| TabPFNv2.5 | 0.5854 | 0.5331 | 0.5831 | 0.5098 | 0.7558 | 0.6278 | 0.5788 | 0.5083 | 0.7805 | 0.6452 |
| TabPFNv2.5-1estimator | 0.5783 | 0.5250 | 0.5772 | 0.5074 | 0.7464 | 0.6231 | 0.5489 | 0.5277 | 0.7745 | 0.6457 |
| TabICLv1 | 0.6670 | 0.5209 | 0.6093 | 0.4957 | 0.7519 | 0.6141 | 0.5758 | 0.5199 | 0.7977 | 0.5498 |
| TabICLv1.1 | 0.6426 | 0.5214 | 0.5729 | 0.5010 | 0.7431 | 0.5965 | 0.5588 | 0.5408 | 0.7577 | 0.5967 |
| TabICLv1.1-1estimator | 0.6476 | 0.5133 | 0.5243 | 0.4909 | 0.7358 | 0.5908 | 0.5623 | 0.5201 | 0.6989 | 0.6050 |
| LimiX2m | 0.6751 | 0.5320 | 0.5942 | 0.5045 | 0.7512 | 0.6122 | 0.5312 | 0.5603 | 0.7632 | 0.5792 |
| LimiX16m | 0.6241 | 0.5259 | 0.6391 | 0.5083 | 0.7553 | 0.6205 | 0.5393 | 0.5400 | 0.7541 | 0.5709 |
| LimiX16m-1estimator | 0.6051 | 0.5204 | 0.6197 | 0.5069 | 0.7496 | 0.6178 | 0.5361 | 0.5351 | 0.7519 | 0.5839 |
| RDBPFN_single_table | 0.6675 | 0.5105 | 0.6351 | 0.5019 | 0.6228 | 0.5761 | 0.5219 | 0.5252 | 0.6591 | 0.5317 |
| RDBPFN | 0.6408 | 0.5329 | 0.6398 | 0.4957 | 0.7532 | 0.6007 | 0.5605 | 0.5572 | 0.7486 | 0.6455 |

**Part 2: Remaining 9 Datasets and Average**

| Model | Rel F1 Driver Dnf | Rel F1 Driver Top3 | Rel Hm User Churn | Rel Stack User Badge | Rel Stack User Engagement | Rel Trial Study Outcome | Retailrocket Cvr | Stackexchange Churn | Stackexchange Upvote | Average |
|---|---|---|---|---|---|---|---|---|---|---|
| Random Forest | 0.6999 | 0.7822 | 0.6023 | 0.7595 | 0.7943 | 0.5340 | 0.7082 | 0.7853 | 0.8398 | 0.6483 |
| XGBoost | 0.6757 | 0.7824 | 0.5871 | 0.6454 | 0.7275 | 0.5379 | 0.6103 | 0.7383 | 0.8171 | 0.6286 |
| XGBoost-Tuned | 0.7007 | 0.7763 | 0.5982 | 0.6449 | 0.6503 | 0.5285 | 0.5930 | 0.7344 | 0.8384 | 0.6177 |
| AutoGluon-Medium | 0.6932 | 0.7268 | 0.5604 | 0.5954 | 0.6289 | 0.5225 | 0.5588 | 0.6899 | 0.8341 | 0.6085 |
| AutoGluon-Best | 0.6341 | 0.7500 | 0.5918 | 0.7296 | 0.7060 | 0.5275 | 0.5901 | 0.7408 | 0.8357 | 0.6300 |
| Mitra | 0.7125 | 0.7936 | 0.5953 | 0.7794 | 0.7815 | 0.5273 | 0.7268 | 0.7766 | 0.8401 | 0.6636 |
| TabPFNv2 | 0.6966 | 0.7907 | 0.6310 | 0.7569 | 0.7356 | 0.5488 | 0.7091 | 0.8018 | 0.8426 | 0.6680 |
| TabPFNv2.5 | 0.6945 | 0.7924 | 0.6284 | 0.7569 | 0.7507 | 0.5519 | 0.7034 | 0.8179 | 0.8455 | 0.6658 |
| TabPFNv2.5-1estimator | 0.6985 | 0.7840 | 0.6184 | 0.7404 | 0.7425 | 0.5595 | 0.7030 | 0.8006 | 0.8411 | 0.6601 |
| TabICLv1 | 0.7019 | 0.7866 | 0.6174 | 0.7882 | 0.8105 | 0.5426 | 0.7079 | 0.8134 | 0.8434 | 0.6692 |
| TabICLv1.1 | 0.7088 | 0.7936 | 0.6301 | 0.7974 | 0.7990 | 0.5440 | 0.7083 | 0.7986 | 0.8437 | 0.6661 |
| TabICLv1.1-1estimator | 0.7057 | 0.7916 | 0.6173 | 0.7927 | 0.7982 | 0.5311 | 0.6958 | 0.8072 | 0.8395 | 0.6562 |
| LimiX2m | 0.6946 | 0.7950 | 0.6214 | 0.7267 | 0.7789 | 0.5638 | 0.7087 | 0.8101 | 0.8419 | 0.6655 |
| LimiX16m | 0.6839 | 0.7827 | 0.6058 | 0.7864 | 0.7535 | 0.5624 | 0.7000 | 0.7895 | 0.8461 | 0.6625 |
| LimiX16m-1estimator | 0.6765 | 0.7812 | 0.5993 | 0.7606 | 0.7487 | 0.5642 | 0.6907 | 0.7802 | 0.8439 | 0.6564 |
| RDBPFN_single_table | 0.6242 | 0.7653 | 0.5732 | 0.7607 | 0.7385 | 0.5393 | 0.6618 | 0.7902 | 0.8395 | 0.6339 |
| RDBPFN | 0.7000 | 0.8013 | 0.6159 | 0.7464 | 0.7803 | 0.5386 | 0.7127 | 0.7952 | 0.8410 | 0.6688 |

*Table 8.* Raw Results for Context Size 256

**Part 1: First 10 Datasets**

| Model | Amazon Churn | Avs Repeater | Diginetica Ctr | Outbrain Small Ctr | Rel Amazon Item Churn | Rel Amazon User Churn | Rel Avito User Clicks | Rel Avito User Visits | Rel Event User Ignore | Rel Event User Repeat |
|---|---|---|---|---|---|---|---|---|---|---|
| Random Forest | 0.6230 | 0.5189 | 0.5826 | 0.5177 | 0.7135 | 0.5812 | 0.5728 | 0.5250 | 0.7709 | 0.6257 |
| XGBoost | 0.5992 | 0.5163 | 0.5697 | 0.5142 | 0.7056 | 0.5792 | 0.5063 | 0.5631 | 0.8075 | 0.6265 |
| XGBoost-Tuned | 0.6050 | 0.5123 | 0.5331 | 0.5095 | 0.7074 | 0.5791 | 0.5326 | 0.5863 | 0.7642 | 0.6429 |
| AutoGluon-Medium | 0.6075 | 0.5246 | 0.5173 | 0.5055 | 0.7274 | 0.5683 | 0.5006 | 0.5868 | 0.7797 | 0.6122 |
| AutoGluon-Best | 0.5936 | 0.5159 | 0.5534 | 0.5205 | 0.7283 | 0.5924 | 0.5196 | 0.5841 | 0.7914 | 0.6487 |
| Mitra | 0.6864 | 0.5353 | 0.6576 | 0.5018 | 0.7639 | 0.6129 | 0.5498 | 0.5812 | 0.7670 | 0.6438 |
| TabPFNv2 | 0.6972 | 0.5324 | 0.6422 | 0.5284 | 0.7691 | 0.6321 | 0.5856 | 0.6047 | 0.8209 | 0.6797 |
| TabPFNv2.5 | 0.6653 | 0.5356 | 0.6292 | 0.5275 | 0.7727 | 0.6310 | 0.6131 | 0.5585 | 0.8191 | 0.6732 |
| TabPFNv2.5-1estimator | 0.6655 | 0.5274 | 0.6193 | 0.5259 | 0.7697 | 0.6275 | 0.5691 | 0.5760 | 0.8047 | 0.6504 |
| TabICLv1 | 0.7000 | 0.5175 | 0.6341 | 0.5107 | 0.7727 | 0.6212 | 0.6051 | 0.5764 | 0.8145 | 0.5825 |
| TabICLv1.1 | 0.6981 | 0.5147 | 0.5790 | 0.5197 | 0.7701 | 0.6119 | 0.5879 | 0.5910 | 0.7871 | 0.6304 |
| TabICLv1.1-1estimator | 0.7119 | 0.5082 | 0.5536 | 0.5152 | 0.7710 | 0.6040 | 0.5834 | 0.5700 | 0.7326 | 0.6507 |
| LimiX2m | 0.6990 | 0.5184 | 0.6441 | 0.5252 | 0.7681 | 0.6263 | 0.5453 | 0.5899 | 0.7948 | 0.6304 |
| LimiX16m | 0.6416 | 0.5179 | 0.6536 | 0.5280 | 0.7700 | 0.6338 | 0.5351 | 0.5965 | 0.7817 | 0.6239 |
| LimiX16m-1estimator | 0.6293 | 0.5168 | 0.6577 | 0.5267 | 0.7645 | 0.6331 | 0.5248 | 0.6029 | 0.7800 | 0.6268 |
| RDBPFN_single_table | 0.7134 | 0.5073 | 0.6595 | 0.5103 | 0.6624 | 0.5681 | 0.5712 | 0.5900 | 0.7443 | 0.5827 |
| RDBPFN | 0.6604 | 0.5264 | 0.6566 | 0.5290 | 0.7570 | 0.6103 | 0.5968 | 0.6247 | 0.8073 | 0.6629 |

**Part 2: Remaining 9 Datasets and Average**

| Model | Rel F1 Driver Dnf | Rel F1 Driver Top3 | Rel Hm User Churn | Rel Stack User Badge | Rel Stack User Engagement | Rel Trial Study Outcome | Retailrocket Cvr | Stackexchange Churn | Stackexchange Upvote | Average |
|---|---|---|---|---|---|---|---|---|---|---|
| Random Forest | 0.6941 | 0.7765 | 0.5787 | 0.7849 | 0.8279 | 0.5442 | 0.7063 | 0.7846 | 0.8364 | 0.6613 |
| XGBoost | 0.6728 | 0.7907 | 0.5696 | 0.7609 | 0.7819 | 0.5547 | 0.6717 | 0.7638 | 0.8310 | 0.6518 |
| XGBoost-Tuned | 0.6877 | 0.7931 | 0.5962 | 0.7867 | 0.7275 | 0.5517 | 0.6497 | 0.7624 | 0.8375 | 0.6508 |
| AutoGluon-Medium | 0.6797 | 0.7898 | 0.5873 | 0.7046 | 0.7131 | 0.5542 | 0.5962 | 0.7517 | 0.8424 | 0.6394 |
| AutoGluon-Best | 0.6765 | 0.7929 | 0.5720 | 0.7888 | 0.7443 | 0.5643 | 0.6674 | 0.7867 | 0.8438 | 0.6571 |
| Mitra | 0.7075 | 0.8147 | 0.6337 | 0.8193 | 0.8337 | 0.5373 | 0.7148 | 0.8154 | 0.8463 | 0.6854 |
| TabPFNv2 | 0.7113 | 0.7995 | 0.6466 | 0.7624 | 0.8153 | 0.5883 | 0.7510 | 0.8238 | 0.8473 | 0.6967 |
| TabPFNv2.5 | 0.7135 | 0.8056 | 0.6468 | 0.7916 | 0.8254 | 0.5913 | 0.7447 | 0.8422 | 0.8496 | 0.6966 |
| TabPFNv2.5-1estimator | 0.7173 | 0.8038 | 0.6426 | 0.7630 | 0.8095 | 0.5677 | 0.7299 | 0.8306 | 0.8472 | 0.6867 |
| TabICLv1 | 0.7107 | 0.7982 | 0.6318 | 0.8161 | 0.8412 | 0.5842 | 0.7232 | 0.8270 | 0.8502 | 0.6904 |
| TabICLv1.1 | 0.7137 | 0.8056 | 0.6473 | 0.8181 | 0.8438 | 0.5667 | 0.7157 | 0.8054 | 0.8438 | 0.6868 |
| TabICLv1.1-1estimator | 0.7155 | 0.8044 | 0.6386 | 0.8095 | 0.8423 | 0.5500 | 0.7084 | 0.8130 | 0.8466 | 0.6805 |
| LimiX2m | 0.7004 | 0.8098 | 0.6337 | 0.7925 | 0.8295 | 0.5972 | 0.7298 | 0.8297 | 0.8421 | 0.6898 |
| LimiX16m | 0.6960 | 0.7955 | 0.6327 | 0.7909 | 0.8197 | 0.5864 | 0.7240 | 0.8232 | 0.8453 | 0.6840 |
| LimiX16m-1estimator | 0.6799 | 0.7853 | 0.6082 | 0.7691 | 0.8167 | 0.5879 | 0.6974 | 0.8159 | 0.8418 | 0.6771 |
| RDBPFN_single_table | 0.6126 | 0.8079 | 0.5980 | 0.7567 | 0.7614 | 0.5858 | 0.7203 | 0.8285 | 0.8442 | 0.6645 |
| RDBPFN | 0.7171 | 0.8090 | 0.6279 | 0.7864 | 0.8255 | 0.5844 | 0.7418 | 0.8233 | 0.8436 | 0.6942 |

*Table 9.* Raw Results for Context Size 512

**Part 1: First 10 Datasets**

| Model | Amazon Churn | Avs Repeater | Diginetica Ctr | Outbrain Small Ctr | Rel Amazon Item Churn | Rel Amazon User Churn | Rel Avito User Clicks | Rel Avito User Visits | Rel Event User Ignore | Rel Event User Repeat |
|---|---|---|---|---|---|---|---|---|---|---|
| Random Forest | 0.6392 | 0.5277 | 0.6084 | 0.5174 | 0.7292 | 0.6064 | 0.6129 | 0.5016 | 0.7821 | 0.6919 |
| XGBoost | 0.6219 | 0.5289 | 0.6143 | 0.5105 | 0.7289 | 0.5978 | 0.5707 | 0.5799 | 0.8100 | 0.6939 |
| XGBoost-Tuned | 0.6408 | 0.5215 | 0.5756 | 0.5079 | 0.7297 | 0.6116 | 0.5785 | 0.6057 | 0.7848 | 0.7089 |
| AutoGluon-Medium | 0.6479 | 0.5280 | 0.5475 | 0.5223 | 0.7278 | 0.6073 | 0.5666 | 0.5852 | 0.7750 | 0.6731 |
| AutoGluon-Best | 0.6976 | 0.5292 | 0.5616 | 0.5210 | 0.7500 | 0.6144 | 0.5435 | 0.5748 | 0.8046 | 0.7024 |
| Mitra | 0.7106 | 0.5467 | 0.6717 | 0.5056 | 0.7833 | 0.6405 | 0.5941 | 0.6005 | 0.7846 | 0.6980 |
| TabPFNv2 | 0.7220 | 0.5457 | 0.6356 | 0.5257 | 0.7814 | 0.6440 | 0.6130 | 0.6234 | 0.8286 | 0.7125 |
| TabPFNv2.5 | 0.6772 | 0.5503 | 0.6109 | 0.5302 | 0.7841 | 0.6402 | 0.6256 | 0.5702 | 0.8265 | 0.7113 |
| TabPFNv2.5-1estimator | 0.6471 | 0.5478 | 0.6138 | 0.5123 | 0.7789 | 0.6369 | 0.6081 | 0.5954 | 0.8201 | 0.7037 |
| TabICLv1 | 0.7160 | 0.5487 | 0.6539 | 0.5114 | 0.7775 | 0.6441 | 0.6264 | 0.6137 | 0.8163 | 0.6392 |
| TabICLv1.1 | 0.7213 | 0.5446 | 0.5995 | 0.5152 | 0.7767 | 0.6463 | 0.6123 | 0.6169 | 0.7953 | 0.6741 |
| TabICLv1.1-1estimator | 0.7306 | 0.5384 | 0.5544 | 0.5107 | 0.7788 | 0.6420 | 0.6077 | 0.6037 | 0.7458 | 0.6627 |
| LimiX2m | 0.7213 | 0.5508 | 0.6638 | 0.5251 | 0.7731 | 0.6405 | 0.6029 | 0.6068 | 0.8121 | 0.6920 |
| LimiX16m | 0.6499 | 0.5484 | 0.6845 | 0.5279 | 0.7779 | 0.6380 | 0.6040 | 0.6162 | 0.8079 | 0.6901 |
| LimiX16m-1estimator | 0.6467 | 0.5355 | 0.6747 | 0.5259 | 0.7700 | 0.6410 | 0.5937 | 0.6242 | 0.8085 | 0.6883 |
| RDBPFN_single_table | 0.7378 | 0.5300 | 0.6516 | 0.4719 | 0.6667 | 0.6070 | 0.6221 | 0.6142 | 0.7565 | 0.6572 |
| RDBPFN | 0.7040 | 0.5536 | 0.6735 | 0.5256 | 0.7758 | 0.6445 | 0.6113 | 0.6253 | 0.8207 | 0.7276 |

**Part 2: Remaining 9 Datasets and Average**

| Model | Rel F1 Driver Dnf | Rel F1 Driver Top3 | Rel Hm User Churn | Rel Stack User Badge | Rel Stack User Engagement | Rel Trial Study Outcome | Retailrocket Cvr | Stackexchange Churn | Stackexchange Upvote | Average |
|---|---|---|---|---|---|---|---|---|---|---|
| Random Forest | 0.6916 | 0.7661 | 0.6017 | 0.7687 | 0.8149 | 0.5599 | 0.7169 | 0.7922 | 0.8448 | 0.6723 |
| XGBoost | 0.6883 | 0.7866 | 0.5907 | 0.7579 | 0.7910 | 0.5781 | 0.7070 | 0.7757 | 0.8445 | 0.6725 |
| XGBoost-Tuned | 0.6902 | 0.7877 | 0.6022 | 0.7804 | 0.7759 | 0.5769 | 0.6888 | 0.7860 | 0.8508 | 0.6739 |
| AutoGluon-Medium | 0.7057 | 0.7698 | 0.6043 | 0.7465 | 0.7649 | 0.5654 | 0.6522 | 0.7800 | 0.8483 | 0.6641 |
| AutoGluon-Best | 0.6556 | 0.7659 | 0.6066 | 0.6982 | 0.7222 | 0.5800 | 0.6419 | 0.8365 | 0.8484 | 0.6660 |
| Mitra | 0.7138 | 0.8040 | 0.6650 | 0.8202 | 0.8356 | 0.5255 | 0.7529 | 0.8180 | 0.8502 | 0.7011 |
| TabPFNv2 | 0.7190 | 0.8001 | 0.6659 | 0.7546 | 0.8273 | 0.6011 | 0.7766 | 0.8436 | 0.8570 | 0.7093 |
| TabPFNv2.5 | 0.7151 | 0.7994 | 0.6662 | 0.7684 | 0.8315 | 0.6013 | 0.7647 | 0.8514 | 0.8576 | 0.7043 |
| TabPFNv2.5-1estimator | 0.7115 | 0.7950 | 0.6630 | 0.7503 | 0.8236 | 0.5977 | 0.7500 | 0.8461 | 0.8558 | 0.6977 |
| TabICLv1 | 0.7126 | 0.7933 | 0.6535 | 0.8187 | 0.8323 | 0.5984 | 0.7383 | 0.8312 | 0.8546 | 0.7042 |
| TabICLv1.1 | 0.7176 | 0.7887 | 0.6600 | 0.8202 | 0.8296 | 0.5926 | 0.7388 | 0.8115 | 0.8536 | 0.7008 |
| TabICLv1.1-1estimator | 0.7180 | 0.7832 | 0.6568 | 0.8054 | 0.8257 | 0.5717 | 0.7312 | 0.8203 | 0.8524 | 0.6916 |
| LimiX2m | 0.7156 | 0.7968 | 0.6535 | 0.7993 | 0.8206 | 0.6173 | 0.7463 | 0.8376 | 0.8536 | 0.7068 |
| LimiX16m | 0.7110 | 0.7977 | 0.6479 | 0.7780 | 0.8180 | 0.5843 | 0.7419 | 0.8356 | 0.8546 | 0.7007 |
| LimiX16m-1estimator | 0.7093 | 0.7903 | 0.6231 | 0.7527 | 0.8185 | 0.5775 | 0.7387 | 0.8292 | 0.8524 | 0.6947 |
| RDBPFN_single_table | 0.6640 | 0.7937 | 0.6393 | 0.8181 | 0.7985 | 0.5961 | 0.7453 | 0.8465 | 0.8322 | 0.6868 |
| RDBPFN | 0.7219 | 0.8023 | 0.6536 | 0.7816 | 0.8246 | 0.5986 | 0.7554 | 0.8444 | 0.8496 | 0.7102 |

*Table 10.* Raw Results for Context Size 1024

**Part 1: First 10 Datasets**

| Model | Amazon Churn | Avs Repeater | Diginetica Ctr | Outbrain Small Ctr | Rel Amazon Item Churn | Rel Amazon User Churn | Rel Avito User Clicks | Rel Avito User Visits | Rel Event User Ignore | Rel Event User Repeat |
|---|---|---|---|---|---|---|---|---|---|---|
| Random Forest | 0.6360 | 0.5289 | 0.6125 | 0.5183 | 0.7425 | 0.6124 | 0.5973 | 0.5169 | 0.7913 | 0.6807 |
| XGBoost | 0.6200 | 0.5319 | 0.6198 | 0.5127 | 0.7410 | 0.5969 | 0.5409 | 0.5932 | 0.8174 | 0.6947 |
| XGBoost-Tuned | 0.6405 | 0.5307 | 0.6371 | 0.5179 | 0.7448 | 0.6065 | 0.5930 | 0.6167 | 0.8209 | 0.7277 |
| AutoGluon-Medium | 0.6749 | 0.5299 | 0.5953 | 0.5208 | 0.7537 | 0.5952 | 0.5373 | 0.5974 | 0.8101 | 0.7075 |
| AutoGluon-Best | 0.7000 | 0.5352 | 0.5627 | 0.5272 | 0.7709 | 0.5989 | 0.5187 | 0.6096 | 0.8080 | 0.7075 |
| Mitra | 0.7009 | 0.5618 | 0.6959 | 0.5020 | 0.7882 | 0.6417 | 0.5627 | 0.6395 | 0.8059 | 0.7159 |
| TabPFNv2 | 0.7219 | 0.5529 | 0.7057 | 0.5379 | 0.7975 | 0.6481 | 0.6290 | 0.6417 | 0.8299 | 0.7185 |
| TabPFNv2.5 | 0.6693 | 0.5549 | 0.6638 | 0.5383 | 0.7961 | 0.6447 | 0.6326 | 0.6173 | 0.8318 | 0.7313 |
| TabPFNv2.5-1estimator | 0.6380 | 0.5532 | 0.6631 | 0.5233 | 0.7898 | 0.6439 | 0.6113 | 0.6220 | 0.8301 | 0.7200 |
| TabICLv1 | 0.7239 | 0.5547 | 0.6769 | 0.5210 | 0.7870 | 0.6492 | 0.6189 | 0.6377 | 0.8216 | 0.6818 |
| TabICLv1.1 | 0.7278 | 0.5494 | 0.6326 | 0.5256 | 0.7892 | 0.6482 | 0.6176 | 0.6441 | 0.8085 | 0.7004 |
| TabICLv1.1-1estimator | 0.7399 | 0.5446 | 0.6042 | 0.5186 | 0.7883 | 0.6464 | 0.6130 | 0.6395 | 0.7521 | 0.6806 |
| LimiX2m | 0.7117 | 0.5534 | 0.7034 | 0.5357 | 0.7845 | 0.6430 | 0.6159 | 0.6395 | 0.8249 | 0.6930 |
| LimiX16m | 0.6496 | 0.5582 | 0.7116 | 0.5373 | 0.7894 | 0.6394 | 0.6141 | 0.6461 | 0.8213 | 0.7176 |
| LimiX16m-1estimator | 0.6487 | 0.5550 | 0.6894 | 0.5335 | 0.7860 | 0.6428 | 0.5931 | 0.6478 | 0.8212 | 0.7135 |
| RDBPFN_single_table | 0.7455 | 0.5388 | 0.6639 | 0.4760 | 0.6660 | 0.6115 | 0.6200 | 0.6364 | 0.7601 | 0.6738 |
| RDBPFN | 0.7179 | 0.5599 | 0.7004 | 0.5352 | 0.7821 | 0.6479 | 0.6266 | 0.6546 | 0.8273 | 0.7533 |

**Part 2: Remaining 9 Datasets and Average**

| Model | Rel F1 Driver Dnf | Rel F1 Driver Top3 | Rel Hm User Churn | Rel Stack User Badge | Rel Stack User Engagement | Rel Trial Study Outcome | Retailrocket Cvr | Stackexchange Churn | Stackexchange Upvote | Average |
|---|---|---|---|---|---|---|---|---|---|---|
| Random Forest | 0.6779 | 0.7689 | 0.6227 | 0.7801 | 0.8240 | 0.5715 | 0.7454 | 0.7952 | 0.8375 | 0.6768 |
| XGBoost | 0.6948 | 0.7791 | 0.6050 | 0.7566 | 0.8172 | 0.5885 | 0.7269 | 0.7937 | 0.8460 | 0.6777 |
| XGBoost-Tuned | 0.7035 | 0.7554 | 0.6025 | 0.7941 | 0.8025 | 0.5919 | 0.7392 | 0.8160 | 0.8499 | 0.6890 |
| AutoGluon-Medium | 0.6874 | 0.7684 | 0.6368 | 0.7640 | 0.7501 | 0.5748 | 0.6864 | 0.8100 | 0.8454 | 0.6761 |
| AutoGluon-Best | 0.7007 | 0.7805 | 0.6368 | 0.7664 | 0.7751 | 0.5943 | 0.6374 | 0.8401 | 0.8428 | 0.6788 |
| Mitra | 0.7113 | 0.8274 | 0.6673 | 0.8235 | 0.8575 | 0.5624 | 0.7733 | 0.8334 | 0.8538 | 0.7118 |
| TabPFNv2 | 0.7142 | 0.7962 | 0.6685 | 0.8157 | 0.8568 | 0.6045 | 0.7981 | 0.8538 | 0.8606 | 0.7238 |
| TabPFNv2.5 | 0.7165 | 0.8042 | 0.6683 | 0.8208 | 0.8534 | 0.6258 | 0.7912 | 0.8588 | 0.8609 | 0.7200 |
| TabPFNv2.5-1estimator | 0.7200 | 0.7927 | 0.6652 | 0.7860 | 0.8466 | 0.6232 | 0.7792 | 0.8532 | 0.8608 | 0.7117 |
| TabICLv1 | 0.7157 | 0.8082 | 0.6666 | 0.8291 | 0.8626 | 0.6198 | 0.7580 | 0.8441 | 0.8557 | 0.7175 |
| TabICLv1.1 | 0.7174 | 0.8059 | 0.6660 | 0.8297 | 0.8539 | 0.6080 | 0.7653 | 0.8197 | 0.8576 | 0.7140 |
| TabICLv1.1-1estimator | 0.7155 | 0.7765 | 0.6642 | 0.8205 | 0.8517 | 0.5843 | 0.7533 | 0.8237 | 0.8572 | 0.7039 |
| LimiX2m | 0.7122 | 0.8021 | 0.6627 | 0.8148 | 0.8528 | 0.6260 | 0.7735 | 0.8493 | 0.8570 | 0.7187 |
| LimiX16m | 0.7154 | 0.8072 | 0.6607 | 0.8021 | 0.8429 | 0.6102 | 0.7738 | 0.8448 | 0.8610 | 0.7159 |
| LimiX16m-1estimator | 0.7114 | 0.8070 | 0.6456 | 0.7820 | 0.8299 | 0.6050 | 0.7720 | 0.8439 | 0.8586 | 0.7098 |
| RDBPFN_single_table | 0.7086 | 0.7997 | 0.6523 | 0.8228 | 0.8086 | 0.5963 | 0.7700 | 0.8498 | 0.8303 | 0.6963 |
| RDBPFN | 0.7188 | 0.8115 | 0.6648 | 0.8126 | 0.8655 | 0.6159 | 0.7708 | 0.8477 | 0.8527 | 0.7245 |

*Table 11.* Single-Table Benchmark Results

**Part 1: First 12 Datasets**

| Model | Bioresponse | Diabetes130us | Higgs | Magictelescope | Miniboone | Albert | Bank Marketing | California | Compas Two Years | Covertype | Covertype (v2) | Credit |
|---|---|---|---|---|---|---|---|---|---|---|---|---|
| Random Forest | 0.8256 | 0.5861 | 0.7307 | 0.9107 | 0.9620 | 0.6899 | 0.8579 | 0.9318 | 0.6547 | 0.8579 | 0.8222 | 0.8386 |
| XGBoost | 0.8300 | 0.5666 | 0.7329 | 0.9046 | 0.9703 | 0.6647 | 0.8432 | 0.9397 | 0.6470 | 0.8480 | 0.8147 | 0.8148 |
| AutoGluon | 0.8146 | 0.6159 | 0.7208 | 0.9078 | 0.9650 | 0.6845 | 0.8582 | 0.9355 | 0.7210 | 0.8459 | 0.8104 | 0.8455 |
| Mitra | 0.8083 | 0.6341 | 0.7525 | 0.9250 | 0.9735 | 0.6969 | 0.8708 | 0.9475 | 0.7302 | 0.8456 | 0.8252 | 0.8518 |
| TabPFNv2 | 0.8210 | 0.6366 | 0.7627 | 0.9263 | 0.9761 | 0.7026 | 0.8701 | 0.9567 | 0.7297 | 0.8684 | 0.8427 | 0.8511 |
| TabPFNv2.5 | 0.8367 | 0.6356 | 0.7644 | 0.9307 | 0.9779 | 0.7011 | 0.8687 | 0.9530 | 0.7291 | 0.8739 | 0.8426 | 0.8511 |
| TabPFNv2.5-1estimator | 0.8247 | 0.6331 | 0.7574 | 0.9305 | 0.9763 | 0.6942 | 0.8673 | 0.9487 | 0.7317 | 0.8676 | 0.8351 | 0.8537 |
| TabICLv1 | 0.8362 | 0.6335 | 0.7548 | 0.9290 | 0.9759 | 0.7011 | 0.8673 | 0.9459 | 0.7284 | 0.8652 | 0.8347 | 0.8514 |
| TabICLv1.1 | 0.8372 | 0.6344 | 0.7619 | 0.9274 | 0.9777 | 0.7023 | 0.8661 | 0.9480 | 0.7259 | 0.8717 | 0.8430 | 0.8507 |
| TabICLv1.1-1estimator | 0.8160 | 0.6351 | 0.7512 | 0.9269 | 0.9761 | 0.6959 | 0.8687 | 0.9466 | 0.7257 | 0.8678 | 0.8370 | 0.8522 |
| LimiX2m | 0.8374 | 0.6351 | 0.7600 | 0.9261 | 0.9742 | 0.6971 | 0.8745 | 0.9573 | 0.7306 | 0.8762 | 0.8438 | 0.8492 |
| LimiX16m | 0.8393 | 0.6313 | 0.7639 | 0.9313 | 0.9773 | 0.6980 | 0.8740 | 0.9580 | 0.7313 | 0.8829 | 0.8468 | 0.8533 |
| LimiX16m-1estimator | 0.8364 | 0.6292 | 0.7564 | 0.9250 | 0.9753 | 0.6896 | 0.8674 | 0.9555 | 0.7290 | 0.8796 | 0.8360 | 0.8496 |
| RDBPFN_single_table | 0.7582 | 0.6317 | 0.7348 | 0.9100 | 0.9647 | 0.6935 | 0.8675 | 0.9248 | 0.7205 | 0.8225 | 0.8185 | 0.8337 |
| RDBPFN | 0.8107 | 0.6277 | 0.7469 | 0.9125 | 0.9605 | 0.6979 | 0.8683 | 0.9358 | 0.7269 | 0.8498 | 0.8259 | 0.8454 |

**Part 2: Remaining 11 Datasets and Average**

| Model | Default of Credit Card Clients | Default of Credit Card Clients (Cat) | Electricity | Electricity (Cat) | Eye Movements | Eye Movements (Cat) | Heloc | House 16h | Jannis | Pol | Road Safety | Average |
|---|---|---|---|---|---|---|---|---|---|---|---|---|
| Random Forest | 0.7578 | 0.7527 | 0.8789 | 0.8700 | 0.6046 | 0.6025 | 0.7783 | 0.9342 | 0.8152 | 0.9941 | 0.7999 | 0.8024 |
| XGBoost | 0.7388 | 0.7323 | 0.8850 | 0.8713 | 0.6011 | 0.5935 | 0.7613 | 0.9371 | 0.8152 | 0.9949 | 0.7897 | 0.7955 |
| AutoGluon | 0.7593 | 0.7607 | 0.8709 | 0.8659 | 0.5948 | 0.5832 | 0.7697 | 0.9318 | 0.8019 | 0.9943 | 0.7851 | 0.8019 |
| Mitra | 0.7686 | 0.7692 | 0.8767 | 0.8718 | 0.5944 | 0.5905 | 0.7862 | 0.9460 | 0.8272 | 0.9945 | 0.8035 | 0.8126 |
| TabPFNv2 | 0.7731 | 0.7734 | 0.8928 | 0.8824 | 0.6205 | 0.6176 | 0.7939 | 0.9471 | 0.8325 | 0.9976 | 0.8364 | 0.8222 |
| TabPFNv2.5 | 0.7730 | 0.7739 | 0.8731 | 0.8683 | 0.6223 | 0.6179 | 0.7925 | 0.9464 | 0.8352 | 0.9981 | 0.8371 | 0.8219 |
| TabPFNv2.5-1estimator | 0.7734 | 0.7721 | 0.8693 | 0.8617 | 0.6171 | 0.6156 | 0.7864 | 0.9454 | 0.8372 | 0.9974 | 0.8319 | 0.8186 |
| TabICLv1 | 0.7720 | 0.7729 | 0.8676 | 0.8615 | 0.6069 | 0.5983 | 0.7900 | 0.9464 | 0.8212 | 0.9962 | 0.8226 | 0.8165 |
| TabICLv1.1 | 0.7743 | 0.7756 | 0.8768 | 0.8701 | 0.6140 | 0.6018 | 0.7900 | 0.9473 | 0.8305 | 0.9951 | 0.8193 | 0.8192 |
| TabICLv1.1-1estimator | 0.7744 | 0.7757 | 0.8737 | 0.8672 | 0.6146 | 0.6083 | 0.7865 | 0.9466 | 0.8310 | 0.9953 | 0.8128 | 0.8168 |
| LimiX2m | 0.7755 | 0.7749 | 0.8960 | 0.8838 | 0.6369 | 0.6264 | 0.7911 | 0.9470 | 0.8378 | 0.9971 | 0.8263 | 0.8241 |
| LimiX16m | 0.7731 | 0.7716 | 0.9023 | 0.8945 | 0.6526 | 0.6426 | 0.7882 | 0.9481 | 0.8419 | 0.9975 | 0.8305 | 0.8274 |
| LimiX16m-1estimator | 0.7693 | 0.7665 | 0.9009 | 0.8949 | 0.6228 | 0.6189 | 0.7839 | 0.9446 | 0.8384 | 0.9969 | 0.8214 | 0.8212 |
| RDBPFN_single_table | 0.7678 | 0.7688 | 0.8473 | 0.8415 | 0.5781 | 0.5821 | 0.7868 | 0.9402 | 0.8156 | 0.9854 | 0.7987 | 0.7997 |
| RDBPFN | 0.7722 | 0.7704 | 0.8695 | 0.8648 | 0.6050 | 0.5962 | 0.7900 | 0.9423 | 0.8226 | 0.9880 | 0.7974 | 0.8099 |

*Table 12.* Raw PR-AUC Results for Context Size 64

**Part 1: First 10 Datasets**

| Model | Amazon Churn | Avs Repeater | Diginetica Ctr | Outbrain Small Ctr | Rel Amazon Item Churn | Rel Amazon User Churn | Rel Avito User Clicks | Rel Avito User Visits | Rel Event User Ignore | Rel Event User Repeat |
|---|---|---|---|---|---|---|---|---|---|---|
| Random Forest | 0.5394 | 0.3691 | 0.0245 | 0.2002 | 0.7463 | 0.4534 | 0.0205 | 0.1505 | 0.9465 | 0.5481 |
| XGBoost | 0.5306 | 0.3676 | 0.0098 | 0.1943 | 0.7221 | 0.4114 | 0.0151 | 0.1607 | 0.9388 | 0.5372 |
| AutoGluon | 0.5269 | 0.3655 | 0.0098 | 0.1956 | 0.7388 | 0.4162 | 0.0154 | 0.1602 | 0.9401 | 0.5538 |
| Mitra | 0.5259 | 0.3702 | 0.0286 | 0.1973 | 0.7794 | 0.4102 | 0.0206 | 0.1603 | 0.9365 | 0.4980 |
| TabPFNv2.5 | 0.5480 | 0.3730 | 0.0241 | 0.1994 | 0.8155 | 0.5053 | 0.0200 | 0.1497 | 0.9511 | 0.5396 |
| TabICLv1.1 | 0.5618 | 0.3644 | 0.0170 | 0.1942 | 0.7750 | 0.4383 | 0.0192 | 0.1643 | 0.9462 | 0.5332 |
| LimiX16m | 0.5651 | 0.3679 | 0.0235 | 0.1993 | 0.8150 | 0.4789 | 0.0205 | 0.1516 | 0.9483 | 0.5256 |
| RDBPFN | 0.5772 | 0.3745 | 0.0255 | 0.2026 | 0.7986 | 0.4852 | 0.0202 | 0.1554 | 0.9473 | 0.5567 |

**Part 2: Remaining 9 Datasets and Average**

| Model | Rel F1 Driver Dnf | Rel F1 Driver Top3 | Rel Hm User Churn | Rel Stack User Badge | Rel Stack User Engagement | Rel Trial Study Outcome | Retailrocket Cvr | Stackexchange Churn | Stackexchange Upvote | Average |
|---|---|---|---|---|---|---|---|---|---|---|
| Random Forest | 0.4874 | 0.3182 | 0.8696 | 0.9893 | 0.9897 | 0.4244 | 0.0405 | 0.9875 | 0.5981 | 0.5107 |
| XGBoost | 0.4491 | 0.3403 | 0.8408 | 0.9772 | 0.9758 | 0.4336 | 0.0319 | 0.9767 | 0.5905 | 0.5002 |
| AutoGluon | 0.4375 | 0.3196 | 0.8548 | 0.9791 | 0.9748 | 0.4192 | 0.0256 | 0.9790 | 0.5948 | 0.5004 |
| Mitra | 0.4667 | 0.3595 | 0.8590 | 0.9922 | 0.9918 | 0.4249 | 0.0417 | 0.9882 | 0.5803 | 0.5069 |
| TabPFNv2.5 | 0.4593 | 0.3332 | 0.8755 | 0.9916 | 0.9904 | 0.4504 | 0.0422 | 0.9895 | 0.6049 | 0.5191 |
| TabICLv1.1 | 0.4861 | 0.3127 | 0.8641 | 0.9919 | 0.9913 | 0.4331 | 0.0432 | 0.9875 | 0.5867 | 0.5111 |
| LimiX16m | 0.4738 | 0.3065 | 0.8739 | 0.9924 | 0.9914 | 0.4427 | 0.0442 | 0.9891 | 0.6080 | 0.5167 |
| RDBPFN | 0.4446 | 0.3484 | 0.8725 | 0.9902 | 0.9894 | 0.4552 | 0.0416 | 0.9871 | 0.5906 | 0.5191 |

*Table 13.* Raw PR-AUC Results for Context Size 128

**Part 1: First 10 Datasets**

| Model | Amazon Churn | Avs Repeater | Diginetica Ctr | Outbrain Small Ctr | Rel Amazon Item Churn | Rel Amazon User Churn | Rel Avito User Clicks | Rel Avito User Visits | Rel Event User Ignore | Rel Event User Repeat |
|---|---|---|---|---|---|---|---|---|---|---|
| Random Forest | 0.5418 | 0.3760 | 0.0154 | 0.1935 | 0.7807 | 0.4642 | 0.0181 | 0.1503 | 0.9497 | 0.5305 |
| XGBoost | 0.5149 | 0.3744 | 0.0104 | 0.1935 | 0.7291 | 0.4375 | 0.0155 | 0.1733 | 0.9363 | 0.5356 |
| AutoGluon | 0.5478 | 0.3690 | 0.0098 | 0.2008 | 0.7596 | 0.4676 | 0.0154 | 0.1733 | 0.9497 | 0.5424 |
| Mitra | 0.5760 | 0.3821 | 0.0342 | 0.1917 | 0.8114 | 0.4759 | 0.0189 | 0.1760 | 0.9412 | 0.5401 |
| TabPFNv2.5 | 0.5428 | 0.3833 | 0.0221 | 0.2057 | 0.8453 | 0.5606 | 0.0205 | 0.1619 | 0.9563 | 0.5718 |
| TabICLv1.1 | 0.5837 | 0.3757 | 0.0147 | 0.1986 | 0.8336 | 0.5053 | 0.0184 | 0.1741 | 0.9548 | 0.5174 |
| LimiX16m | 0.5734 | 0.3782 | 0.0223 | 0.2005 | 0.8455 | 0.5498 | 0.0175 | 0.1734 | 0.9520 | 0.5119 |
| RDBPFN | 0.5886 | 0.3841 | 0.0259 | 0.1964 | 0.8422 | 0.5223 | 0.0185 | 0.1758 | 0.9489 | 0.5757 |

**Part 2: Remaining 9 Datasets and Average**

| Model | Rel F1 Driver Dnf | Rel F1 Driver Top3 | Rel Hm User Churn | Rel Stack User Badge | Rel Stack User Engagement | Rel Trial Study Outcome | Retailrocket Cvr | Stackexchange Churn | Stackexchange Upvote | Average |
|---|---|---|---|---|---|---|---|---|---|---|
| Random Forest | 0.4594 | 0.3440 | 0.8683 | 0.9891 | 0.9914 | 0.4446 | 0.0505 | 0.9887 | 0.6131 | 0.5142 |
| XGBoost | 0.4703 | 0.3679 | 0.8645 | 0.9827 | 0.9829 | 0.4412 | 0.0356 | 0.9826 | 0.6072 | 0.5082 |
| AutoGluon | 0.4429 | 0.3305 | 0.8525 | 0.9789 | 0.9804 | 0.4360 | 0.0303 | 0.9795 | 0.6021 | 0.5089 |
| Mitra | 0.4802 | 0.3984 | 0.8605 | 0.9904 | 0.9905 | 0.4323 | 0.0545 | 0.9873 | 0.6001 | 0.5232 |
| TabPFNv2.5 | 0.4446 | 0.3836 | 0.8793 | 0.9888 | 0.9894 | 0.4581 | 0.0583 | 0.9907 | 0.6098 | 0.5302 |
| TabICLv1.1 | 0.4583 | 0.3696 | 0.8796 | 0.9912 | 0.9915 | 0.4503 | 0.0536 | 0.9895 | 0.6117 | 0.5248 |
| LimiX16m | 0.4353 | 0.3722 | 0.8710 | 0.9910 | 0.9896 | 0.4661 | 0.0503 | 0.9891 | 0.6144 | 0.5265 |
| RDBPFN | 0.4629 | 0.3835 | 0.8751 | 0.9885 | 0.9906 | 0.4444 | 0.0606 | 0.9892 | 0.6009 | 0.5302 |

*Table 14.* Raw PR-AUC Results for Context Size 256

**Part 1: First 10 Datasets**

| Model | Amazon Churn | Avs Repeater | Diginetica Ctr | Outbrain Small Ctr | Rel Amazon Item Churn | Rel Amazon User Churn | Rel Avito User Clicks | Rel Avito User Visits | Rel Event User Ignore | Rel Event User Repeat |
|---|---|---|---|---|---|---|---|---|---|---|
| Random Forest | 0.5677 | 0.3745 | 0.0173 | 0.2024 | 0.7893 | 0.4703 | 0.0195 | 0.1668 | 0.9571 | 0.5513 |
| XGBoost | 0.5472 | 0.3694 | 0.0115 | 0.2014 | 0.7816 | 0.4551 | 0.0179 | 0.1879 | 0.9487 | 0.5659 |
| AutoGluon | 0.5612 | 0.3796 | 0.0104 | 0.1994 | 0.8168 | 0.4720 | 0.0158 | 0.1815 | 0.9564 | 0.5286 |
| Mitra | 0.6271 | 0.3860 | 0.0302 | 0.1979 | 0.8478 | 0.5165 | 0.0184 | 0.1845 | 0.9533 | 0.5544 |
| TabPFNv2.5 | 0.6098 | 0.3886 | 0.0294 | 0.2266 | 0.8553 | 0.5590 | 0.0236 | 0.1765 | 0.9646 | 0.5899 |
| TabICLv1.1 | 0.6415 | 0.3692 | 0.0155 | 0.2131 | 0.8549 | 0.5269 | 0.0214 | 0.1890 | 0.9609 | 0.5432 |
| LimiX16m | 0.5953 | 0.3726 | 0.0230 | 0.2226 | 0.8567 | 0.5683 | 0.0180 | 0.1916 | 0.9571 | 0.5498 |
| RDBPFN | 0.6106 | 0.3808 | 0.0269 | 0.2256 | 0.8449 | 0.5255 | 0.0225 | 0.2005 | 0.9609 | 0.5844 |

**Part 2: Remaining 9 Datasets and Average**

| Model | Rel F1 Driver Dnf | Rel F1 Driver Top3 | Rel Hm User Churn | Rel Stack User Badge | Rel Stack User Engagement | Rel Trial Study Outcome | Retailrocket Cvr | Stackexchange Churn | Stackexchange Upvote | Average |
|---|---|---|---|---|---|---|---|---|---|---|
| Random Forest | 0.4445 | 0.3360 | 0.8584 | 0.9903 | 0.9931 | 0.4606 | 0.0542 | 0.9885 | 0.6044 | 0.5182 |
| XGBoost | 0.4440 | 0.3623 | 0.8637 | 0.9901 | 0.9866 | 0.4526 | 0.0414 | 0.9859 | 0.5985 | 0.5164 |
| AutoGluon | 0.4330 | 0.3745 | 0.8622 | 0.9850 | 0.9854 | 0.4600 | 0.0344 | 0.9851 | 0.6180 | 0.5189 |
| Mitra | 0.4575 | 0.4059 | 0.8744 | 0.9923 | 0.9930 | 0.4398 | 0.0510 | 0.9901 | 0.6132 | 0.5333 |
| TabPFNv2.5 | 0.4602 | 0.3684 | 0.8861 | 0.9906 | 0.9931 | 0.4870 | 0.0744 | 0.9919 | 0.6201 | 0.5418 |
| TabICLv1.1 | 0.4653 | 0.3638 | 0.8859 | 0.9921 | 0.9936 | 0.4637 | 0.0537 | 0.9896 | 0.6073 | 0.5342 |
| LimiX16m | 0.4455 | 0.3544 | 0.8817 | 0.9906 | 0.9927 | 0.4834 | 0.0574 | 0.9908 | 0.6119 | 0.5349 |
| RDBPFN | 0.4777 | 0.3794 | 0.8779 | 0.9906 | 0.9927 | 0.4819 | 0.0707 | 0.9907 | 0.6087 | 0.5396 |

*Table 15.* Raw PR-AUC Results for Context Size 512

| | | | | Part 1: First 10 Datasets | | | | | |
|---|---|---|---|---|---|---|---|---|---|
| Model | Amazon Churn | Avs Repeater | Diginetica Ctr | Outbrain Small Ctr | Rel Amazon Item Churn | Rel Amazon User Churn | Rel Avito User Clicks | Rel Avito User Visits | Rel Event User Ignore | Rel Event User Repeat |
| Random Forest | 0.5799 | 0.3811 | 0.0190 | 0.2029 | 0.8101 | 0.5081 | 0.0242 | 0.1591 | 0.9591 | 0.6023 |
| XGBoost | 0.5815 | 0.3760 | 0.0137 | 0.2002 | 0.8032 | 0.4917 | 0.0207 | 0.1956 | 0.9583 | 0.6223 |
| AutoGluon | 0.5812 | 0.3817 | 0.0113 | 0.2109 | 0.8180 | 0.5167 | 0.0195 | 0.1848 | 0.9559 | 0.5858 |
| Mitra | 0.6553 | 0.3913 | 0.0304 | 0.1972 | 0.8622 | 0.5713 | 0.0225 | 0.1894 | 0.9574 | 0.6023 |
| TabPFNv2.5 | 0.6214 | 0.4037 | 0.0230 | 0.2296 | 0.8662 | 0.5785 | 0.0272 | 0.1782 | 0.9656 | 0.6354 |
| TabICLv1.1 | 0.6682 | 0.3998 | 0.0198 | 0.2135 | 0.8615 | 0.5821 | 0.0249 | 0.1975 | 0.9628 | 0.5955 |
| LimiX16m | 0.6021 | 0.4014 | 0.0301 | 0.2276 | 0.8634 | 0.5775 | 0.0249 | 0.1970 | 0.9627 | 0.6082 |
| RDBPFN | 0.6519 | 0.4071 | 0.0281 | 0.2224 | 0.8588 | 0.5748 | 0.0257 | 0.2002 | 0.9658 | 0.6467 |

| | | | | Part 2: Remaining 9 Datasets and Average | | | | | |
|---|---|---|---|---|---|---|---|---|---|
| Model | Rel F1 Driver Dnf | Rel F1 Driver Top3 | Rel Hm User Churn | Rel Stack User Badge | Rel Stack User Engagement | Rel Trial Study Outcome | Retailrocket Cvr | Stackexchange Churn | Stackexchange Upvote | Average |
| Random Forest | 0.4331 | 0.3123 | 0.8681 | 0.9894 | 0.9922 | 0.4767 | 0.0577 | 0.9891 | 0.6176 | 0.5254 |
| XGBoost | 0.4378 | 0.3561 | 0.8661 | 0.9896 | 0.9899 | 0.4791 | 0.0447 | 0.9874 | 0.6232 | 0.5283 |
| AutoGluon | 0.4670 | 0.3241 | 0.8669 | 0.9876 | 0.9892 | 0.4634 | 0.0411 | 0.9878 | 0.6215 | 0.5271 |
| Mitra | 0.4738 | 0.3877 | 0.8893 | 0.9925 | 0.9932 | 0.4327 | 0.0602 | 0.9901 | 0.6187 | 0.5430 |
| TabPFNv2.5 | 0.4725 | 0.3587 | 0.8928 | 0.9895 | 0.9932 | 0.4986 | 0.0910 | 0.9922 | 0.6420 | 0.5505 |
| TabICLv1.1 | 0.4677 | 0.3364 | 0.8906 | 0.9922 | 0.9926 | 0.4894 | 0.0625 | 0.9897 | 0.6308 | 0.5462 |
| LimiX16m | 0.4609 | 0.3569 | 0.8868 | 0.9895 | 0.9926 | 0.4849 | 0.0712 | 0.9913 | 0.6329 | 0.5454 |
| RDBPFN | 0.4804 | 0.3623 | 0.8874 | 0.9901 | 0.9926 | 0.5004 | 0.0863 | 0.9916 | 0.6174 | 0.5521 |

*Table 16.* Raw PR-AUC Results for Context Size 1024

| | | | | Part 1: First 10 Datasets | | | | | |
|---|---|---|---|---|---|---|---|---|---|
| Model | Amazon Churn | Avs Repeater | Diginetica Ctr | Outbrain Small Ctr | Rel Amazon Item Churn | Rel Amazon User Churn | Rel Avito User Clicks | Rel Avito User Visits | Rel Event User Ignore | Rel Event User Repeat |
| Random Forest | 0.5817 | 0.3826 | 0.0226 | 0.2012 | 0.8194 | 0.5157 | 0.0222 | 0.1671 | 0.9612 | 0.6042 |
| XGBoost | 0.5921 | 0.3815 | 0.0176 | 0.2050 | 0.8190 | 0.5065 | 0.0217 | 0.1983 | 0.9648 | 0.6444 |
| AutoGluon | 0.6157 | 0.3844 | 0.0140 | 0.2100 | 0.8401 | 0.5072 | 0.0184 | 0.1884 | 0.9630 | 0.6174 |
| Mitra | 0.6510 | 0.4054 | 0.0333 | 0.1969 | 0.8642 | 0.5708 | 0.0193 | 0.2081 | 0.9628 | 0.6241 |
| TabPFNv2.5 | 0.6121 | 0.4063 | 0.0334 | 0.2367 | 0.8743 | 0.5819 | 0.0282 | 0.2006 | 0.9674 | 0.6618 |
| TabICLv1.1 | 0.6783 | 0.4025 | 0.0211 | 0.2263 | 0.8693 | 0.5827 | 0.0251 | 0.2110 | 0.9657 | 0.6118 |
| LimiX16m | 0.6002 | 0.4058 | 0.0308 | 0.2406 | 0.8711 | 0.5778 | 0.0255 | 0.2142 | 0.9645 | 0.6437 |
| RDBPFN | 0.6645 | 0.4106 | 0.0302 | 0.2318 | 0.8621 | 0.5766 | 0.0277 | 0.2173 | 0.9662 | 0.6714 |

| | | | | Part 2: Remaining 9 Datasets and Average | | | | | |
|---|---|---|---|---|---|---|---|---|---|
| Model | Rel F1 Driver Dnf | Rel F1 Driver Top3 | Rel Hm User Churn | Rel Stack User Badge | Rel Stack User Engagement | Rel Trial Study Outcome | Retailrocket Cvr | Stackexchange Churn | Stackexchange Upvote | Average |
| Random Forest | 0.4139 | 0.3155 | 0.8728 | 0.9900 | 0.9926 | 0.4842 | 0.0657 | 0.9897 | 0.6099 | 0.5270 |
| XGBoost | 0.4565 | 0.3195 | 0.8658 | 0.9904 | 0.9916 | 0.4992 | 0.0532 | 0.9902 | 0.6197 | 0.5335 |
| AutoGluon | 0.4488 | 0.3192 | 0.8792 | 0.9889 | 0.9884 | 0.4853 | 0.0466 | 0.9900 | 0.6157 | 0.5327 |
| Mitra | 0.4728 | 0.4363 | 0.8902 | 0.9925 | 0.9942 | 0.4558 | 0.0677 | 0.9912 | 0.6239 | 0.5506 |
| TabPFNv2.5 | 0.4691 | 0.3618 | 0.8926 | 0.9922 | 0.9942 | 0.5267 | 0.0931 | 0.9928 | 0.6477 | 0.5565 |
| TabICLv1.1 | 0.4722 | 0.3589 | 0.8920 | 0.9926 | 0.9939 | 0.5028 | 0.0764 | 0.9905 | 0.6379 | 0.5532 |
| LimiX16m | 0.4702 | 0.3584 | 0.8899 | 0.9912 | 0.9939 | 0.5056 | 0.0784 | 0.9921 | 0.6467 | 0.5527 |
| RDBPFN | 0.4801 | 0.3726 | 0.8913 | 0.9918 | 0.9948 | 0.5176 | 0.0911 | 0.9921 | 0.6287 | 0.5589 |

*Table 17.* **Fine-tuning and cross-task adaptation results.** We compare RDB-PFN fine-tuned on the target task, RDB-PFN cross-task fine-tuned on other RDB tasks excluding the target task, and TabPFNv2.5 target-task fine-tuning. The fine-tuning data can be spread across the training set, but the context size remains limited to 1,024.

| | | | | Part 1: First 10 Datasets | | | | | |
|---|---|---|---|---|---|---|---|---|---|
| Model | Amazon Churn | Avs Repeater | Diginetica Ctr | Outbrain Small Ctr | Rel Amazon Item Churn | Rel Amazon User Churn | Rel Avito User Clicks | Rel Avito User Visits | Rel Event User Ignore | Rel Event User Repeat |
| RDBPFN-Finetune | 0.7337 | 0.5561 | 0.7600 | 0.5322 | 0.8054 | 0.6576 | 0.6459 | 0.6598 | 0.8282 | 0.7456 |
| RDBPFN-Cross-RDB-Finetune | 0.7153 | 0.5630 | 0.7193 | 0.5305 | 0.7858 | 0.6455 | 0.6398 | 0.6486 | 0.8182 | 0.7572 |
| TabPFNv2.5-Finetune | 0.7280 | 0.5520 | 0.6887 | 0.5357 | 0.7982 | 0.6475 | 0.5884 | 0.6346 | 0.8228 | 0.6445 |

| | | | | Part 2: Remaining 9 Datasets and Average | | | | | |
|---|---|---|---|---|---|---|---|---|---|
| Model | Rel F1 Driver Dnf | Rel F1 Driver Top3 | Rel Hm User Churn | Rel Stack User Badge | Rel Stack User Engagement | Rel Trial Study Outcome | Retailrocket Cvr | Stackexchange Churn | Stackexchange Upvote | Average |
| RDBPFN-Finetune | 0.7229 | 0.7360 | 0.6744 | 0.8449 | 0.8829 | 0.6432 | 0.7959 | 0.8607 | 0.8652 | 0.7342 |
| RDBPFN-Cross-RDB-Finetune | 0.7289 | 0.8317 | 0.6611 | 0.7993 | 0.8564 | 0.6130 | 0.7834 | 0.8494 | 0.8497 | 0.7261 |
| TabPFNv2.5-Finetune | 0.7204 | 0.7975 | 0.6693 | 0.8152 | 0.8522 | 0.6261 | 0.7899 | 0.8559 | 0.8602 | 0.7172 |

*Table 18.* Raw Results for Effective Context Size 2048 via Chunked-Support Ensembling

**Part 1: First 10 Datasets**

| Model | Amazon Churn | Avs Repeater | Diginetica Ctr | Outbrain Small Ctr | Rel Amazon Item Churn | Rel Amazon User Churn | Rel Avito User Clicks | Rel Avito User Visits | Rel Event User Ignore | Rel Event User Repeat |
|---|---|---|---|---|---|---|---|---|---|---|
| Random Forest | 0.6532 | 0.5289 | 0.6454 | 0.5207 | 0.7650 | 0.6186 | 0.6036 | 0.5340 | 0.8017 | 0.6963 |
| XGBoost | 0.6383 | 0.5341 | 0.6494 | 0.5118 | 0.7624 | 0.6006 | 0.5563 | 0.6058 | 0.8233 | 0.6947 |
| AutoGluon | 0.6850 | 0.5324 | 0.6058 | 0.5274 | 0.7658 | 0.6269 | 0.5417 | 0.6115 | 0.8082 | 0.7020 |
| Mitra | 0.7118 | 0.5571 | 0.7205 | 0.5123 | 0.7943 | 0.6495 | 0.5604 | 0.6436 | 0.8116 | 0.7304 |
| TabPFNv2.5 | 0.6676 | 0.5500 | 0.6816 | 0.5322 | 0.8043 | 0.6534 | 0.6386 | 0.6323 | 0.8407 | 0.7389 |
| TabICLv1.1 | 0.7335 | 0.5472 | 0.6851 | 0.5252 | 0.7978 | 0.6572 | 0.6296 | 0.6538 | 0.8186 | 0.7135 |
| LimiX16m | 0.6763 | 0.5591 | 0.7185 | 0.5322 | 0.7980 | 0.6431 | 0.6287 | 0.6524 | 0.8288 | 0.7241 |
| RDBPFN | 0.7147 | 0.5563 | 0.7215 | 0.5278 | 0.7882 | 0.6454 | 0.6369 | 0.6536 | 0.8329 | 0.7474 |

**Part 2: Remaining 9 Datasets and Average**

| Model | Rel F1 Driver Dnf | Rel F1 Driver Top3 | Rel Hm User Churn | Rel Stack User Badge | Rel Stack User Engagement | Rel Trial Study Outcome | Retailrocket Cvr | Stackexchange Churn | Stackexchange Upvote | Average |
|---|---|---|---|---|---|---|---|---|---|---|
| Random Forest | 0.6756 | 0.7642 | 0.6279 | 0.8062 | 0.8572 | 0.5999 | 0.7606 | 0.8078 | 0.8458 | 0.6901 |
| XGBoost | 0.6794 | 0.7963 | 0.6205 | 0.7843 | 0.8469 | 0.6096 | 0.7551 | 0.7948 | 0.8502 | 0.6902 |
| AutoGluon | 0.7008 | 0.7900 | 0.6460 | 0.7770 | 0.8214 | 0.6261 | 0.7147 | 0.8132 | 0.8560 | 0.6922 |
| Mitra | 0.7144 | 0.8255 | 0.6683 | 0.8430 | 0.8736 | 0.5986 | 0.7873 | 0.8341 | 0.8538 | 0.7205 |
| TabPFNv2.5 | 0.7142 | 0.7995 | 0.6737 | 0.8376 | 0.8707 | 0.6428 | 0.8066 | 0.8580 | 0.8631 | 0.7266 |
| TabICLv1.1 | 0.7143 | 0.8021 | 0.6693 | 0.8435 | 0.8707 | 0.6320 | 0.7845 | 0.8210 | 0.8606 | 0.7242 |
| LimiX16m | 0.7103 | 0.7988 | 0.6682 | 0.8189 | 0.8646 | 0.6430 | 0.7887 | 0.8483 | 0.8628 | 0.7245 |
| RDBPFN | 0.7258 | 0.8045 | 0.6682 | 0.8283 | 0.8762 | 0.6335 | 0.7696 | 0.8543 | 0.8546 | 0.7284 |

*Table 19.* Raw Results for Effective Context Size 4096 via Chunked-Support Ensembling

**Part 1: First 10 Datasets**

| Model | Amazon Churn | Avs Repeater | Diginetica Ctr | Outbrain Small Ctr | Rel Amazon Item Churn | Rel Amazon User Churn | Rel Avito User Clicks | Rel Avito User Visits | Rel Event User Ignore | Rel Event User Repeat |
|---|---|---|---|---|---|---|---|---|---|---|
| Random Forest | 0.6620 | 0.5378 | 0.6826 | 0.5207 | 0.7737 | 0.6207 | 0.6131 | 0.5362 | 0.8026 | 0.6809 |
| XGBoost | 0.6529 | 0.5424 | 0.6715 | 0.5111 | 0.7711 | 0.6040 | 0.5571 | 0.6153 | 0.8222 | 0.6675 |
| AutoGluon | 0.6968 | 0.5446 | 0.6488 | 0.5213 | 0.7797 | 0.6258 | 0.5523 | 0.6056 | 0.8240 | 0.7371 |
| Mitra | 0.7146 | 0.5618 | 0.7448 | 0.5184 | 0.7959 | 0.6415 | 0.5578 | 0.6496 | 0.8126 | 0.7119 |
| TabPFNv2.5 | 0.6796 | 0.5559 | 0.7138 | 0.5296 | 0.8097 | 0.6537 | 0.6448 | 0.6202 | 0.8397 | 0.7477 |
| TabICLv1.1 | 0.7383 | 0.5523 | 0.7148 | 0.5255 | 0.8028 | 0.6582 | 0.6389 | 0.6566 | 0.8204 | 0.7277 |
| LimiX16m | 0.6866 | 0.5682 | 0.7436 | 0.5316 | 0.8034 | 0.6408 | 0.6432 | 0.6528 | 0.8343 | 0.7305 |
| RDBPFN | 0.7271 | 0.5584 | 0.7349 | 0.5250 | 0.7955 | 0.6538 | 0.6445 | 0.6562 | 0.8328 | 0.7471 |

**Part 2: Remaining 9 Datasets and Average**

| Model | Rel F1 Driver Dnf | Rel F1 Driver Top3 | Rel Hm User Churn | Rel Stack User Badge | Rel Stack User Engagement | Rel Trial Study Outcome | Retailrocket Cvr | Stackexchange Churn | Stackexchange Upvote | Average |
|---|---|---|---|---|---|---|---|---|---|---|
| Random Forest | 0.6416 | 0.7642 | 0.6390 | 0.7967 | 0.8398 | 0.6065 | 0.7763 | 0.8150 | 0.8478 | 0.6925 |
| XGBoost | 0.6653 | 0.7963 | 0.6229 | 0.7905 | 0.8477 | 0.6250 | 0.7750 | 0.8075 | 0.8500 | 0.6945 |
| AutoGluon | 0.6787 | 0.7900 | 0.6477 | 0.7736 | 0.8356 | 0.6249 | 0.7516 | 0.8213 | 0.8561 | 0.7008 |
| Mitra | 0.7184 | 0.8243 | 0.6743 | 0.8425 | 0.8738 | 0.6245 | 0.7959 | 0.8406 | 0.8545 | 0.7241 |
| TabPFNv2.5 | 0.7139 | 0.7995 | 0.6768 | 0.8429 | 0.8693 | 0.6490 | 0.8119 | 0.8617 | 0.8634 | 0.7307 |
| TabICLv1.1 | 0.7119 | 0.8021 | 0.6751 | 0.8417 | 0.8740 | 0.6378 | 0.7880 | 0.8298 | 0.8615 | 0.7293 |
| LimiX16m | 0.7072 | 0.7988 | 0.6751 | 0.8179 | 0.8547 | 0.6580 | 0.7990 | 0.8527 | 0.8635 | 0.7296 |
| RDBPFN | 0.7248 | 0.8045 | 0.6730 | 0.8386 | 0.8790 | 0.6383 | 0.7707 | 0.8549 | 0.8532 | 0.7322 |

*Table 20.* Raw Results for Effective Context Size 8192 via Chunked-Support Ensembling

**Part 1: First 10 Datasets**

| Model | Amazon Churn | Avs Repeater | Diginetica Ctr | Outbrain Small Ctr | Rel Amazon Item Churn | Rel Amazon User Churn | Rel Avito User Clicks | Rel Avito User Visits | Rel Event User Ignore | Rel Event User Repeat |
|---|---|---|---|---|---|---|---|---|---|---|
| Random Forest | 0.6629 | 0.5401 | 0.6918 | 0.5167 | 0.7783 | 0.6240 | 0.6157 | 0.5362 | 0.8027 | 0.6809 |
| XGBoost | 0.6577 | 0.5412 | 0.6853 | 0.5123 | 0.7739 | 0.6131 | 0.5519 | 0.6199 | 0.8165 | 0.6675 |
| AutoGluon | 0.7134 | 0.5448 | 0.6515 | 0.5205 | 0.7952 | 0.6348 | 0.5703 | 0.6353 | 0.8179 | 0.7358 |
| TabPFNv2.5 | 0.6856 | 0.5593 | 0.7127 | 0.5296 | 0.8144 | 0.6624 | 0.6489 | 0.6061 | 0.8404 | 0.7477 |
| TabICLv1.1 | 0.7421 | 0.5559 | 0.7283 | 0.5264 | 0.8068 | 0.6636 | 0.6471 | 0.6594 | 0.8224 | 0.7277 |
| RDBPFN | 0.7338 | 0.5591 | 0.7527 | 0.5259 | 0.7959 | 0.6592 | 0.6446 | 0.6582 | 0.8343 | 0.7471 |

**Part 2: Remaining 9 Datasets and Average**

| Model | Rel F1 Driver Dnf | Rel F1 Driver Top3 | Rel Hm User Churn | Rel Stack User Badge | Rel Stack User Engagement | Rel Trial Study Outcome | Retailrocket Cvr | Stackexchange Churn | Stackexchange Upvote | Average |
|---|---|---|---|---|---|---|---|---|---|---|
| Random Forest | 0.6298 | 0.7642 | 0.6301 | 0.8089 | 0.8376 | 0.6359 | 0.7844 | 0.7982 | 0.8457 | 0.6939 |
| XGBoost | 0.6646 | 0.7963 | 0.6087 | 0.8027 | 0.8577 | 0.6484 | 0.7869 | 0.8018 | 0.8529 | 0.6979 |
| AutoGluon | 0.6704 | 0.7900 | 0.6445 | 0.8162 | 0.8553 | 0.6503 | 0.7962 | 0.8354 | 0.8602 | 0.7125 |
| TabPFNv2.5 | 0.7121 | 0.7995 | 0.6759 | 0.8483 | 0.8821 | 0.6597 | 0.8152 | 0.8600 | 0.8649 | 0.7329 |
| TabICLv1.1 | 0.7048 | 0.8021 | 0.6738 | 0.8461 | 0.8802 | 0.6556 | 0.7966 | 0.8305 | 0.8647 | 0.7334 |
| RDBPFN | 0.7269 | 0.8045 | 0.6721 | 0.8452 | 0.8804 | 0.6450 | 0.7812 | 0.8550 | 0.8560 | 0.7356 |

*Table 21.* Raw Relational Prior Ablation Results for Context Size 64

**Part 1: First 10 Datasets**

| Variant | Amazon Churn | Avs Repeater | Diginetica Ctr | Outbrain Small Ctr | Rel Amazon Item Churn | Rel Amazon User Churn | Rel Avito User Clicks | Rel Avito User Visits | Rel Event User Ignore | Rel Event User Repeat |
|---|---|---|---|---|---|---|---|---|---|---|
| Original | 0.6295 | 0.5142 | 0.5646 | 0.5152 | 0.6747 | 0.5674 | 0.5692 | 0.4936 | 0.7407 | 0.6132 |
| Simpler DAG Generator | 0.5984 | 0.5050 | 0.5594 | 0.5075 | 0.6694 | 0.5674 | 0.5554 | 0.5080 | 0.7191 | 0.5768 |
| Remove Temporal Component | 0.6051 | 0.5104 | 0.5639 | 0.5073 | 0.6612 | 0.5706 | 0.5703 | 0.5031 | 0.7240 | 0.6019 |
| Simpler Edge Generation | 0.6126 | 0.5084 | 0.5464 | 0.5036 | 0.6757 | 0.5746 | 0.5776 | 0.4989 | 0.7218 | 0.5873 |
| Merge Aggregation From Any Rows | 0.6046 | 0.5077 | 0.5734 | 0.5124 | 0.6424 | 0.5679 | 0.5740 | 0.4901 | 0.7417 | 0.5543 |

**Part 2: Remaining 9 Datasets and Average**

| Variant | Rel F1 Driver Dnf | Rel F1 Driver Top3 | Rel Hm User Churn | Rel Stack User Badge | Rel Stack User Engagement | Rel Trial Study Outcome | Retailrocket Cvr | Stackexchange Churn | Stackexchange Upvote | Average |
|---|---|---|---|---|---|---|---|---|---|---|
| Original | 0.7167 | 0.7967 | 0.5734 | 0.7704 | 0.7583 | 0.5215 | 0.6222 | 0.7467 | 0.8292 | 0.6430 |
| Simpler DAG Generator | 0.7008 | 0.7955 | 0.5836 | 0.7769 | 0.7685 | 0.5255 | 0.6348 | 0.7546 | 0.8390 | 0.6392 |
| Remove Temporal Component | 0.7200 | 0.7810 | 0.5901 | 0.7826 | 0.7713 | 0.5215 | 0.6267 | 0.7578 | 0.8368 | 0.6424 |
| Simpler Edge Generation | 0.6517 | 0.7870 | 0.5873 | 0.7735 | 0.7587 | 0.5492 | 0.6227 | 0.7343 | 0.8409 | 0.6375 |
| Merge Aggregation From Any Rows | 0.6896 | 0.8012 | 0.5799 | 0.7732 | 0.7816 | 0.5326 | 0.6149 | 0.7583 | 0.8373 | 0.6388 |

*Table 22.* Raw Relational Prior Ablation Results for Context Size 128

**Part 1: First 10 Datasets**

| Variant | Amazon Churn | Avs Repeater | Diginetica Ctr | Outbrain Small Ctr | Rel Amazon Item Churn | Rel Amazon User Churn | Rel Avito User Clicks | Rel Avito User Visits | Rel Event User Ignore | Rel Event User Repeat |
|---|---|---|---|---|---|---|---|---|---|---|
| Original | 0.6598 | 0.5293 | 0.5727 | 0.5079 | 0.7211 | 0.5875 | 0.5597 | 0.5530 | 0.7486 | 0.6211 |
| Simpler DAG Generator | 0.6014 | 0.5213 | 0.5711 | 0.5069 | 0.7341 | 0.6005 | 0.5258 | 0.5559 | 0.7331 | 0.6116 |
| Remove Temporal Component | 0.5736 | 0.5261 | 0.5710 | 0.5065 | 0.7316 | 0.5974 | 0.5517 | 0.5513 | 0.7166 | 0.6258 |
| Simpler Edge Generation | 0.6470 | 0.5282 | 0.5502 | 0.5019 | 0.7370 | 0.6031 | 0.5609 | 0.5416 | 0.7315 | 0.6207 |
| Merge Aggregation From Any Rows | 0.6351 | 0.5258 | 0.5655 | 0.5013 | 0.7278 | 0.5974 | 0.5585 | 0.5319 | 0.7336 | 0.6057 |

**Part 2: Remaining 9 Datasets and Average**

| Variant | Rel F1 Driver Dnf | Rel F1 Driver Top3 | Rel Hm User Churn | Rel Stack User Badge | Rel Stack User Engagement | Rel Trial Study Outcome | Retailrocket Cvr | Stackexchange Churn | Stackexchange Upvote | Average |
|---|---|---|---|---|---|---|---|---|---|---|
| Original | 0.7077 | 0.7970 | 0.5987 | 0.7698 | 0.7539 | 0.5333 | 0.6643 | 0.7680 | 0.8419 | 0.6576 |
| Simpler DAG Generator | 0.6945 | 0.8032 | 0.6178 | 0.7758 | 0.7576 | 0.5446 | 0.7004 | 0.7774 | 0.8443 | 0.6567 |
| Remove Temporal Component | 0.7075 | 0.7947 | 0.6094 | 0.7883 | 0.7341 | 0.5407 | 0.6969 | 0.7821 | 0.8429 | 0.6552 |
| Simpler Edge Generation | 0.6785 | 0.7907 | 0.5870 | 0.7930 | 0.7609 | 0.5544 | 0.6985 | 0.7619 | 0.8448 | 0.6575 |
| Merge Aggregation From Any Rows | 0.6915 | 0.8059 | 0.5964 | 0.7813 | 0.7780 | 0.5300 | 0.6852 | 0.7645 | 0.8438 | 0.6557 |

*Table 23.* Raw Relational Prior Ablation Results for Context Size 256

**Part 1: First 10 Datasets**

| Variant | Amazon Churn | Avs Repeater | Diginetica Ctr | Outbrain Small Ctr | Rel Amazon Item Churn | Rel Amazon User Churn | Rel Avito User Clicks | Rel Avito User Visits | Rel Event User Ignore | Rel Event User Repeat |
|---|---|---|---|---|---|---|---|---|---|---|
| Original | 0.6770 | 0.5224 | 0.6312 | 0.5274 | 0.7451 | 0.6096 | 0.5844 | 0.5990 | 0.7979 | 0.6810 |
| Simpler DAG Generator | 0.6503 | 0.5133 | 0.6180 | 0.5245 | 0.7523 | 0.6246 | 0.5629 | 0.5911 | 0.7731 | 0.6508 |
| Remove Temporal Component | 0.6615 | 0.5203 | 0.6225 | 0.5227 | 0.7498 | 0.6171 | 0.5698 | 0.5966 | 0.7562 | 0.6744 |
| Simpler Edge Generation | 0.6660 | 0.5239 | 0.5956 | 0.5227 | 0.7486 | 0.6209 | 0.5775 | 0.6072 | 0.8005 | 0.6485 |
| Merge Aggregation From Any Rows | 0.6542 | 0.5185 | 0.6116 | 0.5290 | 0.7418 | 0.6135 | 0.5770 | 0.5902 | 0.7818 | 0.6588 |

**Part 2: Remaining 9 Datasets and Average**

| Variant | Rel F1 Driver Dnf | Rel F1 Driver Top3 | Rel Hm User Churn | Rel Stack User Badge | Rel Stack User Engagement | Rel Trial Study Outcome | Retailrocket Cvr | Stackexchange Churn | Stackexchange Upvote | Average |
|---|---|---|---|---|---|---|---|---|---|---|
| Original | 0.6984 | 0.8033 | 0.6045 | 0.8133 | 0.8240 | 0.5647 | 0.6840 | 0.7995 | 0.8421 | 0.6847 |
| Simpler DAG Generator | 0.6958 | 0.8056 | 0.6215 | 0.8002 | 0.8261 | 0.5835 | 0.7169 | 0.8174 | 0.8487 | 0.6830 |
| Remove Temporal Component | 0.7026 | 0.7921 | 0.6241 | 0.8024 | 0.8255 | 0.5685 | 0.7100 | 0.8123 | 0.8388 | 0.6825 |
| Simpler Edge Generation | 0.6858 | 0.7995 | 0.6033 | 0.8161 | 0.8097 | 0.5793 | 0.7058 | 0.8202 | 0.8440 | 0.6829 |
| Merge Aggregation From Any Rows | 0.7058 | 0.8163 | 0.6036 | 0.8037 | 0.8183 | 0.5719 | 0.7190 | 0.7968 | 0.8406 | 0.6817 |

*Table 24.* Raw Relational Prior Ablation Results for Context Size 512

**Part 1: First 10 Datasets**

| Variant | Amazon Churn | Avs Repeater | Diginetica Ctr | Outbrain Small Ctr | Rel Amazon Item Churn | Rel Amazon User Churn | Rel Avito User Clicks | Rel Avito User Visits | Rel Event User Ignore | Rel Event User Repeat |
|---|---|---|---|---|---|---|---|---|---|---|
| Original | 0.7313 | 0.5431 | 0.6081 | 0.5271 | 0.7543 | 0.6306 | 0.6069 | 0.6192 | 0.8010 | 0.7398 |
| Simpler DAG Generator | 0.6868 | 0.5365 | 0.6121 | 0.5247 | 0.7612 | 0.6409 | 0.5985 | 0.6165 | 0.7858 | 0.7001 |
| Remove Temporal Component | 0.6991 | 0.5425 | 0.6163 | 0.5203 | 0.7713 | 0.6420 | 0.6057 | 0.6123 | 0.7643 | 0.7117 |
| Simpler Edge Generation | 0.6979 | 0.5388 | 0.6130 | 0.5203 | 0.7659 | 0.6428 | 0.6199 | 0.6189 | 0.8037 | 0.7009 |
| Merge Aggregation From Any Rows | 0.6891 | 0.5380 | 0.6138 | 0.5304 | 0.7512 | 0.6234 | 0.6074 | 0.6145 | 0.7816 | 0.7263 |

**Part 2: Remaining 9 Datasets and Average**

| Variant | Rel F1 Driver Dnf | Rel F1 Driver Top3 | Rel Hm User Churn | Rel Stack User Badge | Rel Stack User Engagement | Rel Trial Study Outcome | Retailrocket Cvr | Stackexchange Churn | Stackexchange Upvote | Average |
|---|---|---|---|---|---|---|---|---|---|---|
| Original | 0.7138 | 0.7976 | 0.6376 | 0.8128 | 0.8197 | 0.5651 | 0.7322 | 0.8286 | 0.8514 | 0.7011 |
| Simpler DAG Generator | 0.7127 | 0.7932 | 0.6461 | 0.8086 | 0.8211 | 0.6002 | 0.7445 | 0.8271 | 0.8513 | 0.6983 |
| Remove Temporal Component | 0.7003 | 0.7745 | 0.6489 | 0.8106 | 0.8201 | 0.5862 | 0.7462 | 0.8326 | 0.8506 | 0.6977 |
| Simpler Edge Generation | 0.7135 | 0.7905 | 0.6369 | 0.8044 | 0.8060 | 0.5945 | 0.7495 | 0.8372 | 0.8516 | 0.7003 |
| Merge Aggregation From Any Rows | 0.7185 | 0.8128 | 0.6359 | 0.8128 | 0.8141 | 0.5812 | 0.7631 | 0.8134 | 0.8500 | 0.6988 |

*Table 25.* Raw Relational Prior Ablation Results for Context Size 1024

**Part 1: First 10 Datasets**

| Variant | Amazon Churn | Avs Repeater | Diginetica Ctr | Outbrain Small Ctr | Rel Amazon Item Churn | Rel Amazon User Churn | Rel Avito User Clicks | Rel Avito User Visits | Rel Event User Ignore | Rel Event User Repeat |
|---|---|---|---|---|---|---|---|---|---|---|
| Original | 0.7360 | 0.5454 | 0.6707 | 0.5351 | 0.7564 | 0.6308 | 0.6065 | 0.6503 | 0.8012 | 0.7510 |
| Simpler DAG Generator | 0.6969 | 0.5428 | 0.6397 | 0.5349 | 0.7673 | 0.6401 | 0.6142 | 0.6434 | 0.7822 | 0.7223 |
| Remove Temporal Component | 0.6965 | 0.5464 | 0.6650 | 0.5355 | 0.7782 | 0.6439 | 0.6193 | 0.6463 | 0.7547 | 0.7383 |
| Simpler Edge Generation | 0.6961 | 0.5395 | 0.6598 | 0.5216 | 0.7663 | 0.6419 | 0.6251 | 0.6482 | 0.8049 | 0.7086 |
| Merge Aggregation From Any Rows | 0.6898 | 0.5510 | 0.6597 | 0.5390 | 0.7520 | 0.6227 | 0.6243 | 0.6453 | 0.7818 | 0.7452 |

**Part 2: Remaining 9 Datasets and Average**

| Variant | Rel F1 Driver Dnf | Rel F1 Driver Top3 | Rel Hm User Churn | Rel Stack User Badge | Rel Stack User Engagement | Rel Trial Study Outcome | Retailrocket Cvr | Stackexchange Churn | Stackexchange Upvote | Average |
|---|---|---|---|---|---|---|---|---|---|---|
| Original | 0.7127 | 0.7943 | 0.6564 | 0.8321 | 0.8459 | 0.5736 | 0.7713 | 0.8323 | 0.8518 | 0.7134 |
| Simpler DAG Generator | 0.7093 | 0.7932 | 0.6583 | 0.8299 | 0.8457 | 0.6203 | 0.7700 | 0.8341 | 0.8535 | 0.7104 |
| Remove Temporal Component | 0.6984 | 0.7454 | 0.6535 | 0.8296 | 0.8468 | 0.6022 | 0.7784 | 0.8366 | 0.8469 | 0.7085 |
| Simpler Edge Generation | 0.7165 | 0.7948 | 0.6583 | 0.8132 | 0.8338 | 0.6025 | 0.7758 | 0.8329 | 0.8550 | 0.7103 |
| Merge Aggregation From Any Rows | 0.7183 | 0.8138 | 0.6506 | 0.8271 | 0.8379 | 0.5982 | 0.7833 | 0.8235 | 0.8433 | 0.7109 |

*Table 26.* Raw Backbone and Model-Size Ablation Results for Context Size 64

**Part 1: First 10 Datasets**

| Variant | Amazon Churn | Avs Repeater | Diginetica Ctr | Outbrain Small Ctr | Rel Amazon Item Churn | Rel Amazon User Churn | Rel Avito User Clicks | Rel Avito User Visits | Rel Event User Ignore | Rel Event User Repeat |
|---|---|---|---|---|---|---|---|---|---|---|
| Original | 0.6005 | 0.5086 | 0.5645 | 0.5104 | 0.6717 | 0.5604 | 0.5753 | 0.4934 | 0.7337 | 0.5816 |
| Cols First | 0.5202 | 0.5138 | 0.5672 | 0.4982 | 0.4928 | 0.4889 | 0.5862 | 0.4712 | 0.5866 | 0.5327 |
| Rows First | 0.4461 | 0.4884 | 0.4810 | 0.4995 | 0.4711 | 0.5265 | 0.6197 | 0.3851 | 0.7061 | 0.4739 |
| Small Capacity | 0.5643 | 0.5081 | 0.5834 | 0.5030 | 0.6571 | 0.5506 | 0.5781 | 0.4909 | 0.7218 | 0.6289 |
| Tiny Capacity | 0.5785 | 0.5111 | 0.5556 | 0.5115 | 0.6314 | 0.5230 | 0.5733 | 0.5176 | 0.6663 | 0.5890 |

**Part 2: Remaining 9 Datasets and Average**

| Variant | Rel F1 Driver Dnf | Rel F1 Driver Top3 | Rel Hm User Churn | Rel Stack User Badge | Rel Stack User Engagement | Rel Trial Study Outcome | Retailrocket Cvr | Stackexchange Churn | Stackexchange Upvote | Average |
|---|---|---|---|---|---|---|---|---|---|---|
| Original | 0.6828 | 0.7934 | 0.5693 | 0.7458 | 0.7608 | 0.5273 | 0.6169 | 0.7464 | 0.8331 | 0.6356 |
| Cols First | 0.5953 | 0.6369 | 0.5266 | 0.7502 | 0.5791 | 0.5590 | 0.6208 | 0.7365 | 0.6262 | 0.5731 |
| Rows First | 0.7031 | 0.7109 | 0.5317 | 0.8141 | 0.8023 | 0.4407 | 0.6231 | 0.7937 | 0.2366 | 0.5660 |
| Small Capacity | 0.6595 | 0.7795 | 0.5991 | 0.7846 | 0.6788 | 0.5329 | 0.5984 | 0.7508 | 0.8369 | 0.6319 |
| Tiny Capacity | 0.6962 | 0.7696 | 0.6129 | 0.7824 | 0.6339 | 0.5579 | 0.6330 | 0.7318 | 0.8468 | 0.6275 |

*Table 27.* Raw Backbone and Model-Size Ablation Results for Context Size 128

**Part 1: First 10 Datasets**

| Variant | Amazon Churn | Avs Repeater | Diginetica Ctr | Outbrain Small Ctr | Rel Amazon Item Churn | Rel Amazon User Churn | Rel Avito User Clicks | Rel Avito User Visits | Rel Event User Ignore | Rel Event User Repeat |
|---|---|---|---|---|---|---|---|---|---|---|
| Original | 0.6150 | 0.5252 | 0.5471 | 0.5098 | 0.7178 | 0.5981 | 0.5743 | 0.5301 | 0.7551 | 0.6127 |
| Cols First | 0.5083 | 0.5049 | 0.5722 | 0.5189 | 0.4945 | 0.4745 | 0.5806 | 0.4886 | 0.6092 | 0.5461 |
| Rows First | 0.4664 | 0.4874 | 0.4900 | 0.5013 | 0.4787 | 0.5451 | 0.6199 | 0.3841 | 0.7002 | 0.4457 |
| Small Capacity | 0.5813 | 0.5185 | 0.5495 | 0.5113 | 0.7062 | 0.5776 | 0.5549 | 0.5252 | 0.7287 | 0.6595 |
| Tiny Capacity | 0.6029 | 0.5184 | 0.5611 | 0.5054 | 0.6759 | 0.5586 | 0.5582 | 0.5233 | 0.6820 | 0.6079 |

**Part 2: Remaining 9 Datasets and Average**

| Variant | Rel F1 Driver Dnf | Rel F1 Driver Top3 | Rel Hm User Churn | Rel Stack User Badge | Rel Stack User Engagement | Rel Trial Study Outcome | Retailrocket Cvr | Stackexchange Churn | Stackexchange Upvote | Average |
|---|---|---|---|---|---|---|---|---|---|---|
| Original | 0.6987 | 0.8004 | 0.5912 | 0.7400 | 0.7438 | 0.5393 | 0.6722 | 0.7702 | 0.8428 | 0.6518 |
| Cols First | 0.5800 | 0.6537 | 0.5492 | 0.7827 | 0.7120 | 0.5329 | 0.6252 | 0.7367 | 0.7264 | 0.5893 |
| Rows First | 0.6955 | 0.7008 | 0.5471 | 0.8163 | 0.7925 | 0.4283 | 0.5911 | 0.7923 | 0.2739 | 0.5661 |
| Small Capacity | 0.6612 | 0.7786 | 0.6031 | 0.7597 | 0.7120 | 0.5483 | 0.6502 | 0.7653 | 0.8432 | 0.6439 |
| Tiny Capacity | 0.6647 | 0.7711 | 0.6218 | 0.7850 | 0.6648 | 0.5791 | 0.6847 | 0.7664 | 0.8510 | 0.6412 |

*Table 28.* Raw Backbone and Model-Size Ablation Results for Context Size 256

| Part 1: First 10 Datasets | | | | | | | | | |
|---|---|---|---|---|---|---|---|---|---|
| Variant | Amazon Churn | Avs Repeater | Diginetica Ctr | Outbrain Small Ctr | Rel Amazon Item Churn | Rel Amazon User Churn | Rel Avito User Clicks | Rel Avito User Visits | Rel Event User Ignore | Rel Event User Repeat |
| Original | 0.6665 | 0.5196 | 0.5986 | 0.5292 | 0.7358 | 0.6127 | 0.5880 | 0.5824 | 0.7906 | 0.6512 |
| Cols First | 0.4753 | 0.5116 | 0.5619 | 0.5290 | 0.4965 | 0.4879 | 0.6025 | 0.5004 | 0.6261 | 0.5083 |
| Rows First | 0.4833 | 0.4859 | 0.4885 | 0.5019 | 0.4872 | 0.5302 | 0.6216 | 0.3837 | 0.7038 | 0.3970 |
| Small Capacity | 0.6141 | 0.5123 | 0.5823 | 0.5282 | 0.7295 | 0.5688 | 0.5560 | 0.5694 | 0.7567 | 0.6797 |
| Tiny Capacity | 0.6223 | 0.5215 | 0.5936 | 0.5245 | 0.6709 | 0.5254 | 0.5931 | 0.5743 | 0.6964 | 0.6121 |

| Part 2: Remaining 9 Datasets and Average | | | | | | | | | |
|---|---|---|---|---|---|---|---|---|---|
| Variant | Rel F1 Driver Dnf | Rel F1 Driver Top3 | Rel Hm User Churn | Rel Stack User Badge | Rel Stack User Engagement | Rel Trial Study Outcome | Retailrocket Cvr | Stackexchange Churn | Stackexchange Upvote | Average |
| Original | 0.7131 | 0.8081 | 0.5905 | 0.7883 | 0.8067 | 0.5755 | 0.6999 | 0.8130 | 0.8408 | 0.6795 |
| Cols First | 0.5600 | 0.6717 | 0.5385 | 0.7882 | 0.6476 | 0.5702 | 0.6237 | 0.7644 | 0.7435 | 0.5899 |
| Rows First | 0.7116 | 0.7087 | 0.5276 | 0.8190 | 0.8003 | 0.4283 | 0.6023 | 0.7851 | 0.3436 | 0.5689 |
| Small Capacity | 0.6805 | 0.7920 | 0.6234 | 0.7778 | 0.7872 | 0.5817 | 0.6794 | 0.7957 | 0.8477 | 0.6664 |
| Tiny Capacity | 0.6857 | 0.8032 | 0.6468 | 0.7874 | 0.7105 | 0.5878 | 0.7240 | 0.7918 | 0.8509 | 0.6591 |

*Table 29.* Raw Backbone and Model-Size Ablation Results for Context Size 512

| Part 1: First 10 Datasets | | | | | | | | | |
|---|---|---|---|---|---|---|---|---|---|
| Variant | Amazon Churn | Avs Repeater | Diginetica Ctr | Outbrain Small Ctr | Rel Amazon Item Churn | Rel Amazon User Churn | Rel Avito User Clicks | Rel Avito User Visits | Rel Event User Ignore | Rel Event User Repeat |
| Original | 0.7131 | 0.5388 | 0.5836 | 0.5352 | 0.7445 | 0.6333 | 0.6187 | 0.6021 | 0.8024 | 0.7132 |
| Cols First | 0.5110 | 0.5141 | 0.5759 | 0.5315 | 0.4985 | 0.4931 | 0.6120 | 0.5274 | 0.6704 | 0.4972 |
| Rows First | 0.5047 | 0.4843 | 0.4919 | 0.5038 | 0.5037 | 0.5538 | 0.6233 | 0.3828 | 0.7042 | 0.4200 |
| Small Capacity | 0.6257 | 0.5388 | 0.5722 | 0.5327 | 0.7484 | 0.6112 | 0.5945 | 0.5691 | 0.7669 | 0.7076 |
| Tiny Capacity | 0.6424 | 0.5355 | 0.5860 | 0.5363 | 0.7017 | 0.5503 | 0.6196 | 0.5737 | 0.7108 | 0.6430 |

| Part 2: Remaining 9 Datasets and Average | | | | | | | | | |
|---|---|---|---|---|---|---|---|---|---|
| Variant | Rel F1 Driver Dnf | Rel F1 Driver Top3 | Rel Hm User Churn | Rel Stack User Badge | Rel Stack User Engagement | Rel Trial Study Outcome | Retailrocket Cvr | Stackexchange Churn | Stackexchange Upvote | Average |
| Original | 0.7204 | 0.8035 | 0.6270 | 0.8093 | 0.7981 | 0.5852 | 0.7391 | 0.8301 | 0.8507 | 0.6973 |
| Cols First | 0.6151 | 0.6570 | 0.5298 | 0.7617 | 0.6937 | 0.5944 | 0.6541 | 0.7448 | 0.7330 | 0.6008 |
| Rows First | 0.7176 | 0.7047 | 0.5266 | 0.8199 | 0.7956 | 0.4315 | 0.5955 | 0.7846 | 0.3216 | 0.5721 |
| Small Capacity | 0.7051 | 0.7922 | 0.6429 | 0.8098 | 0.8152 | 0.5885 | 0.7394 | 0.7981 | 0.8517 | 0.6847 |
| Tiny Capacity | 0.6552 | 0.8002 | 0.6575 | 0.7972 | 0.6541 | 0.5953 | 0.7481 | 0.7999 | 0.8510 | 0.6662 |

*Table 30.* Raw Backbone and Model-Size Ablation Results for Context Size 1024

| Part 1: First 10 Datasets | | | | | | | | | |
|---|---|---|---|---|---|---|---|---|---|
| Variant | Amazon Churn | Avs Repeater | Diginetica Ctr | Outbrain Small Ctr | Rel Amazon Item Churn | Rel Amazon User Churn | Rel Avito User Clicks | Rel Avito User Visits | Rel Event User Ignore | Rel Event User Repeat |
| Original | 0.7166 | 0.5494 | 0.6638 | 0.5402 | 0.7482 | 0.6369 | 0.6265 | 0.6376 | 0.8003 | 0.7301 |
| Cols First | 0.5365 | 0.5099 | 0.5890 | 0.5137 | 0.4954 | 0.5106 | 0.6121 | 0.5706 | 0.6681 | 0.5115 |
| Rows First | 0.5241 | 0.4822 | 0.4905 | 0.5006 | 0.5087 | 0.5465 | 0.6231 | 0.3826 | 0.7072 | 0.4001 |
| Small Capacity | 0.6410 | 0.5265 | 0.6089 | 0.5359 | 0.7516 | 0.6090 | 0.5879 | 0.6131 | 0.7464 | 0.7196 |
| Tiny Capacity | 0.6543 | 0.5358 | 0.6255 | 0.5362 | 0.6747 | 0.5502 | 0.6241 | 0.5915 | 0.7135 | 0.5860 |

| Part 2: Remaining 9 Datasets and Average | | | | | | | | | |
|---|---|---|---|---|---|---|---|---|---|
| Variant | Rel F1 Driver Dnf | Rel F1 Driver Top3 | Rel Hm User Churn | Rel Stack User Badge | Rel Stack User Engagement | Rel Trial Study Outcome | Retailrocket Cvr | Stackexchange Churn | Stackexchange Upvote | Average |
| Original | 0.7197 | 0.8038 | 0.6541 | 0.8238 | 0.8248 | 0.6034 | 0.7751 | 0.8349 | 0.8483 | 0.7125 |
| Cols First | 0.6623 | 0.6837 | 0.5397 | 0.7979 | 0.6665 | 0.6066 | 0.6638 | 0.7710 | 0.7488 | 0.6136 |
| Rows First | 0.7242 | 0.7075 | 0.5262 | 0.8219 | 0.7972 | 0.4311 | 0.5948 | 0.7827 | 0.3123 | 0.5718 |
| Small Capacity | 0.7025 | 0.7975 | 0.6617 | 0.8167 | 0.8295 | 0.5836 | 0.7635 | 0.8082 | 0.8531 | 0.6924 |
| Tiny Capacity | 0.6931 | 0.8004 | 0.6544 | 0.7977 | 0.6542 | 0.5894 | 0.7561 | 0.8033 | 0.8509 | 0.6680 |

