# OpenReview forum: "Relational In-Context Learning via Synthetic Pre-training with Structural Prior"
_ICML.cc/2026/Conference — ICML 2026 regular_

### Official Review · Reviewer_Jk5v · 2026-03-08

**Soundness:** 3
**Presentation:** 4
**Significance:** 3
**Originality:** 3
**Overall Recommendation:** 5
**Confidence:** 3

**Summary:**

The paper introduces a synthetic data prior for relational databases. To create a foundation model for relational databases, it uses a single-table TabPFN-style architecture together with deep feature synthesis (DFS) for featurizing databases into single tables. The resulting model, RDB-PFN, achieves state-of-the-art results compared to other tabular foundation models on DFS-encoded databases while being smaller, faster, and using fewer pre-training steps. On single tables, it falls behind other foundation models, but the multi-table pre-training still shows benefits.

**Compliance With Llm Reviewing Policy:**

Affirmed.

**Final Justification:**

The proposed method is interesting, is well-presented and comes with many ablations. The authors have addressed my concerns in the rebuttal. I can't assess the comparison to relational methods in detail as I'm not an expert on that.

**Key Questions For Authors:**

No questions (minor ones see comments).

**Limitations:**

There is no dedicated discussion of limitations, but some of them are contained in the assumptions in Section 4.1. A discussion regarding limitations would be nice (e.g., only binary classification).

**Strengths And Weaknesses:**

**Strengths**:
- Significance: Foundation models for relational databases are an emerging research topic with high potential for practical impact. The creation of good synthetic data generators is relevant due to the lack of public high-quality real-world data, and the combination with DFS builds connections with tabular foundation models and potential ways to generate more data for them.
- Originality: The prior contains new ideas as far as I know, with some connections to preferential attachment, and there is theory studying its universality in some sense (I did not check it). It seems lower quality for single tables (I'm not sure whether there is a graph structure between the columns inside a single table).
- Presentation: The paper is well-written and has nice plots.

**Weaknesses**:

(W1) Part of the evaluation:
- There is only one "native" relational baseline in the appendix, and IMHO it should be present in the main paper. The authors mention that they were unable to establish a fair comparison protocol for RT, would it be possible to elaborate? Is the problem that the test rows are not handled independently of each other? Also, would a comparison to RelGT be feasible? (Of course, the results are interesting in their own right, but especially for making claims like "a rigorously defined synthetic generator is all you need to solve relational reasoning" I would expect more evidence.)
- Some non-foundational baselines use weak settings and some details on their evaluation are lacking, drawing into question the claim that "RDB-PFN [...] consistently outperform[s] traditional methods".
	- For AutoGluon, the version (1.5?) and time limit are not specified and the worst setting (medium_quality) was chosen, which uses holdout instead of cross-validation. The authors justify this by training efficiency, but the training efficiency is controlled through the time limit, and unless that is very low, better presets should improve the benchmark performance. The best setting should be extreme_quality (with GPU), which includes foundation models in the ensemble. If the authors want to make a point about the results that can be achieved without foundation models and/or on CPU, they could use the best_quality preset instead. The lowest time limit I know from the literature is 5 minutes (https://arxiv.org/abs/2504.01222), which might be okay with the small datasets in the paper. (For me, the label in the figure should be something like "AutoGluon 1.5 extreme (5m)")
	- For XGBoost, is cross-validation and early-stopping used? At these dataset sizes, it would also be feasible and cheap to tune hyperparameters, and I think taking the setup / search space from something like TabArena would be the least controversial way to use XGBoost as a baseline. In terms of the fixed hyperparameters, more estimators + smaller learning rate is good, but from my experience (without DFS) the depth seems way too large especially for these small datasets. There is a paper (https://arxiv.org/abs/2411.04324) that optimizes LightGBM for few-shot settings and it limits the number of leaves to 4, which means the maximum depth is 3.

---

> ### Author Rebuttal · Authors · 2026-03-31
>
> Thank you for the positive and careful review. We especially appreciate that you recognized the significance of the problem and the novelty of the synthetic relational prior. In response, we added new experiments and clarifications; due to the response-length limit, detailed results are placed in the anonymous supplement [https://anonymous.4open.science/r/RDBPFN_rebuttal/Results_README.md].
>
> On your concern that there is only one “native” relational baseline in the appendix, and that stronger relational baselines such as RT / RelGT should be discussed more explicitly:
>
> We agree that the paper should position itself more clearly relative to native supervised relational methods. At the same time, we would like to clarify that our target regime is **strict few-shot / in-context adaptation for relational databases**, rather than full-data supervised relational learning with task-specific graph pipelines. In that sense, Griffin was included because it is one of the few relational baselines that is at least closer to this low-label adaptation regime (although it still requires fine-tuning). We do not claim that RDB-PFN universally surpasses strong supervised relational GNN/GT pipelines such as GraphSAGE or RelGT under their native protocols; rather, our claim is that a well-designed **relational synthetic prior** can make a lightweight PFN-style model highly effective in the low-label, low-engineering regime. To make this positioning clearer, we also include contextual comparisons from 4DBInfer, which systematically compare DFS+single-table methods and GNN baselines in the supervised setting, and we additionally add a fine-tuning study for RDB-PFN so readers can inspect its behavior outside the strict ICL regime.
>
> On your question about why RT is awkward to compare fairly, and whether the issue is that test rows are not handled independently:
>
> Yes, the main difficulty is a **supervision-interface mismatch**, not simply a difference in model family. In those temporal forecasting-style tasks, RT can incorporate entity-specific historical label information through additional relational inputs, while our strict ICL protocol uses a single **global support set** shared across test predictions and does not inject per-instance historical labels in that way. We believe both settings can be useful depending on the available data/labels and the willingness to build a task-specific graph pipeline; however, because the accessible supervision differs, a direct comparison is difficult to interpret as apples-to-apples. We will make this distinction more explicit in the revision. Regarding the phrase “a rigorously defined synthetic generator is all you need to solve relational reasoning,” we agree that this wording is too strong, especially because it may mislead readers into thinking that scalable synthetic data alone replaces the benefit of stronger native relational modeling. We will revise it to focus more precisely on the advantage of scalable synthetic relational pretraining.
>
> On your concern that some non-foundational baselines use weak settings, especially AutoGluon and XGBoost:
>
> We agree that stronger baseline settings are important for fairness. In the revision, we strengthen the classical baselines by adding tuning and using a much larger-performance preset for AutoGluon (5 minutes with the best preset). Even with these stronger settings, they still remain behind the strongest TFMs in the few-shot regime, which is consistent with recent low-data tabular benchmark observations (like TabArena).
>
> On your suggestion to add a clearer limitations discussion:
>
> Thank you for this helpful suggestion. We add an explicit limitations section that states more clearly that the current paper focuses primarily on binary classification in a strict few-shot relational setting, and does not fully cover stronger native supervised relational baselines or all task families. We would also like to clarify that we mainly evaluate binary classification because the benchmark composition is overwhelmingly binary (all RelBench classification tasks are binary, and all but one in 4DBInfer are binary), and this is mainly a head choice rather than a pipeline limitation; similarly, many TFMs support multiclass classification through a shared higher-dimensional classification head rather than a fundamentally different architecture. We also include a small multiclass feasibility extension to support multiclass tasks.

---

> > ### Author Rebuttal · Reviewer_Jk5v · 2026-04-01
> >
> > I thank the authors for their detailed response. My concerns regarding the tabular baselines are mostly addressed (I don't know what tuning spaces were used so I can't judge them, but it should not matter much given that it didn't matter much for AutoGluon). I also appreciate the new ablations. Regarding ICL vs RT, do I understand correctly that there are two settings, one with shared context for each test sample and one with separate context for each test sample, and that in principle all methods could handle both settings but for TabPFN-style ICL the second setting would require one forward pass per test sample (with sample-dependent context) and for RT the first setting would require separate forward passes per test sample? (At least from the explanation of the authors I don't understand why a fair comparison would be impossible, just that it would perhaps be too computationally expensive to adapt a method to the other setting.) Anyway, from RDBLearn it seems that ICL performs quite well in a setting that compares both, and there are enough other interesting conclusions to be drawn from the paper. Overall, I raise my score from 4 to 5.
> >
> > I was wondering if the authors plan to release their code, but at least some code is included in the rebuttal material.
> >
> > I accidentally forgot to post my minor comments in the review, I'll provide them here but I'm not expecting a detailed response (but it might be worth addressing the questions in the paper).
> >
> > - l. 237 right: What are the different relation types?
> > - l. 381 right: Ensembling uses the same parameters for all ensemble models. It therefore doesn't decrease the "storage-efficiency", unless you are referring to the size of the KV cache.
> > - Figure 2: There is only one tick on the x-axis, which means the reader can't tell how big the differences are. Moreover, some of the parameter counts are wrong - TabICL has around 27M parameters and TabPFN around 10M (checkpoint size should be divided by 4 bytes per parameter).
> > - Paragraph "The PFN Insight": I find this a bit overexaggerated. "As long as the statistical structure of the real world falls within the support of this synthetic prior, the model generalizes automatically." That's quite an optimistic assumption. Also, the 5,000x speedup on small data is likely just because they set an excessive time limit for the hyperparameter tuning that probably doesn't help, and neglects that TabPFN runs on a GPU while trees run on CPUs.
> > - Explain: What is a "grounded foreign key link"?
> > - Section 5.2: The terms column-wise and row-wise can be understood in the opposite way (as intra-column/intra-row rather than inter-column/inter-row), and have been used in the opposite way at least by TabICL (column-wise = tokens within each column attend to each other, etc.), though it seems that there is no established standard. I wonder if a different terminology would be less confusing.
> > - Figure 5: Is there no higher-dimensional real-world dataset that would allow a more direct comparison to the DFS dataset?
> > - Do you only subsample the table with the target column to generate the different dataset sizes? And is only the train set subsampled or also the test set?
> > - l. 794: The default TabPFN-2.5 checkpoint is RealTabPFN-2.5. It would be great to clarify which one you used (maybe via the detailed checkpoint name or so).

---

> > > ### Author Response · Authors · 2026-04-07
> > >
> > > Thank you very much for the careful follow-up and for raising the score. We are glad that the stronger baselines and additional ablations addressed the main concerns.
> > >
> > > Thank you for this helpful clarification question. Yes, we think your summary is broadly correct: there are effectively two context protocols here, a **shared/global context** and a **target-specific local context**. Our point is not that methods such as RT and TabPFN-style ICL cannot in principle be adapted across these protocols, but that the resulting comparison is hard to make **really fair**, especially in the few-shot setting where **how the context is constructed** and **how much supervision is accessible** are themselves part of the method design. In RT, labels are materialized into the RDB itself as an additional relational table (e.g., a timestamped label table linked back to entities by foreign keys). For example, in a temporal churn-style task over user / item / review-history tables, labels are timestamped outcomes rather than one static label per user, so the same user can have multiple historical labels at different times; once these are materialized in the graph, the sampled subgraph for a target user can naturally expose that user’s own historical labels, and potentially labels of nearby connected entities. Across test instances, the union of accessible labels can also become much larger than a single shared global support set. In contrast, the strict ICL setting uses one global support set shared across test predictions and does not inject per-target historical labels into the relational structure. Thus, moving RT to a shared-global protocol or moving ICL to a graph-conditioned local protocol is not just a computational change; it also changes what label information is accessible and how much supervision is effectively used across test instances. We also agree that intermediate protocols are possible. For example, giving ICL methods larger or more structure-aware contexts, or modifying RT-style methods to expose broader/non-local context, but those would still correspond to different few-shot regimes rather than a truly apples-to-apples comparison. So the comparison is not impossible, but it is difficult to make fully fair without changing the scientific setting each method is designed for. We will revise the wording to make this distinction more precise, and we will also report RT’s referenced performance in the appendix for reference.
> > >
> > > We will also maintain a public code release for reproducibility and broader use.
> > >
> > > We also thank you for the additional minor comments; they are very helpful, and we will address them in the revision. In particular:
> > >
> > > - clarify that **“relation types”** refers to the distinct foreign-key-based table connections incident to the target table;
> > > - remove the unfair **parameter-efficiency** comparison to ensembled baselines and keep the discussion focused on inference efficiency;
> > > - correct the mixed and partly inaccurate parameter reporting (TabPFNv2.5 to 10.7M, TabICLv1.1 to 27.1M, Mitra to 72M, Limix16M to 16M, RDBPFN to 0.7M), improve the x-axis presentation, and report the precise checkpoint details;
> > > - moderate the wording in **“The PFN Insight”** paragraph and revise the speedup phrasing accordingly; specifically, change it to *“The broad coverage of statistical patterns in the synthetic data gives the model the potential to generalize to real data”* and clarify that *“With GPU acceleration, current state-of-the-art solutions can achieve stronger performance, and in some settings reduce runtime by thousands-fold”*;
> > > - remove **“grounded foreign key link”** and simply use **“foreign key”**;
> > > - revise the **row-wise / column-wise terminology** to less ambiguous wording as attention **within the same row / within the same column**;
> > > - add a higher-dimensional benchmark example from TabArena for context in the revised discussion (now updated in the anonymous supplement);
> > > - clarify that we first run DFS on the full RDB, then uniformly downsample the generated task table to 600 rows; dataset difficulty is varied by changing the train/test ratio (0.5–0.9), rather than by separately subsampling raw train/test tables;
> > > - explicitly state that the **TabPFN-2.5 checkpoint** used is `tabpfn-v2.5-classifier-v2.5_default.ckpt`.

---

### Official Review · Reviewer_34Ht · 2026-03-09

**Soundness:** 3
**Presentation:** 3
**Significance:** 3
**Originality:** 3
**Overall Recommendation:** 5
**Confidence:** 2

**Summary:**

This paper studies foundation models for relational databases (RDBs), which are important for modern data science and decision-making. The authors argue that high-quality RDBs are typically private and scarce. While synthetic data can potentially replace the role of real-world training data, existing prior data-fitted networks (PFNs) are typically designed for the single-table domains, and are not compatible with RDBs. To address this limitation, the authors propose a novel framework called RDB-PFN that extends the concept of PFNs to RDBs. Specifically, they introduce a three-step procedure (Schema, Structural Skeleton, and Dependent Content) for synthetic RDB generation, and design an adapted Transformer with a two-stage process (DFS and Bi-Attention Reasoning). They conduct comprehensive empirical evaluations to show that their proposed RDB-PFN outperforms classical ML models and existing tabular foundation models, across various scenarios. In the appendix, they state that their proposed generative framework can approximate any RDB distribution satisfying the assumptions.

**Compliance With Llm Reviewing Policy:**

Affirmed.

**Final Justification:**

The rebuttal adequately addresses my concerns regarding robustness and scalability, and I recommend accepting the paper.

**Key Questions For Authors:**

1. Could the authors provide more intuition or empirical evidence that explains why training on synthetic RDB data alone can outperform larger tabular foundation model trained on real data? Also, it would be interesting to know whether a small amount of real RDB data could further improve the model performance (e.g., through fine-tuning).
2. The synthetic RDB data is generated according to several structural assumptions. I wonder how sensitive the model performance is to the violations of these assumptions in synthetic training data generation process, and in the testing data.
3. The current model has only 2.6M parameters and is relatively small in scale. Could the authors comment on how it scales to larger RDBs and larger model sizes in practice?

**Limitations:**

Yes.

**Strengths And Weaknesses:**

Strengths
1. The authors provide a technically detailed framework for applying the PFNs concept to RDBs. The assumptions for the synthetic data generation are clearly stated, and the theoretical guarantee appears solid (although in the appendix). The experimental comparisons are also fairly comprehensive and include several strong baselines.
2. The paper is overall clearly written and well organized. The notations and algorithms are explained clearly, and the algorithm visualization in Figure 1 helps illustrate the overall framework.
3. The authors study an important topic of learning from relational databases, which is important in many real-world applications, including machine learning and e-commerce platforms.
4. To the best of my knowledge, this is one of the first works that builds a foundation model for learning from synthetic relational databases. It could potentially improve automatic decision-making that relies on relational data.

Weaknesses
1. Although the empirical evaluation of the RDB-PFN model is comprehensive, it still remains unclear why a model trained entirely on synthetic RDB data (without using real data) can outperform larger tabular foundation models that are trained on real data. It would also be important to understand whether the performance could be further improved by fine-tuning the model on a small amount of real RDB data.
2. The paper generates synthetic RDB data under certain assumptions (Section 4.1). It is unclear how the model performs when the (synthetic) test data satisfies all the assumptions, or violates some of these assumptions (e.g., when the schema graph is not a DAG).
3. The organization of the paper can be improved. Sections 1 and 2 are relatively too long, and some of the background could potentially be shortened and leave more space for the method details and theoretical analysis (currently placed in the appendix).

---

> ### Author Rebuttal · Authors · 2026-03-31
>
> Thank you for the positive and thoughtful review. We especially appreciate that you recognized the importance of the problem and the overall technical clarity of the paper. In response, we added new experiments and clarifications; due to the response-length limit, detailed results are placed in the anonymous supplement [https://anonymous.4open.science/r/RDBPFN_rebuttal/Results_README.md].
>
> Q1 / W1 (why synthetic relational pretraining works; effect of real-data fine-tuning):
> We believe the key reason is **prior match rather than raw model scale**. The downstream tasks here come from **multi-table relational systems**, where useful predictive signals arise from cross-table dependencies that are absent from standard single-table priors. After DFS-style linearization, these dependencies appear as structured groups of correlated features that are not typical of purely single-table synthetic training. In this sense, a relationally matched prior can be more valuable than simply scaling up a model trained on a mismatched single-table prior. We also agree that real-data adaptation is important, so we add a **fine-tuning experiment** and observe that downstream fine-tuning can indeed further improve performance. We additionally explore continued training on other real RDBs; the trend there is less clear, likely because the available real-RDB collection is much smaller and more heterogeneous than the synthetic corpus.
>
> Q2 / W2 (sensitivity to structural assumptions):
> We agree that robustness to assumption mismatch is important. To address this, we add a lightweight **playground sensitivity study** in which we alter the structural assumptions, especially locality. In particular, we compare the intended local relational prior against more overly global variants, where some columns can receive information not only from local neighbors but also from randomly selected distant rows. Another extreme violation is to discard the relational structure entirely and degenerate to the single-table case, which is already covered by our existing comparison. Across these variants, the intended local relational prior performs better, suggesting that the structural assumptions are not arbitrary but provide a useful inductive bias. We agree that further study of assumption sensitivity would be valuable.
>
> Q3 (scaling to larger RDBs and larger models):
> We add two lightweight scaling studies. First, we study **effective context scaling** using chunked-support ensembling to test whether the method continues benefiting from more support examples beyond the pretraining context budget. Second, we study **model-size scaling** by comparing reduced variants. Both show gains in the expected direction, although the gains are somewhat marginal rather than dramatic. We present these as an initial characterization rather than a full scaling-law study.
>
> W3 (paper organization):
> Thank you for the helpful suggestion. Our original intent was to make the paper accessible to readers coming from different backgrounds, including relational databases, graph-based and tabular methods, and synthetic data generation. In the revision, we will streamline parts of the background especially the longer related-work discussion, and move some material to the appendix, so that we can make more space for method details and the newly added ablations.

---

> > ### Author Rebuttal · Reviewer_34Ht · 2026-04-04
> >
> > Thank you for the detailed rebuttal and additional experiments, which help address my concerns regarding robustness and scalability. I will adjust my score accordingly.

---

> > > ### Author Response · Authors · 2026-04-07
> > >
> > > Thank you very much for the positive follow-up. We are glad that the added robustness, fine-tuning, and scaling experiments helped address your concerns, and we will make sure these clarifications are reflected clearly in the revised paper.

---

### Official Review · Reviewer_5ThT · 2026-03-12

**Soundness:** 3
**Presentation:** 3
**Significance:** 3
**Originality:** 3
**Overall Recommendation:** 5
**Confidence:** 4

**Summary:**

This paper introduces RDB-PFN, a foundation model for relational databases (RDBs) that extends Prior-Data Fitted Networks (PFNs) from a single-table setting to a multi-table relational setting. The key challenge is the lack of sufficient high-quality pretraining data for relational deep learning, which the authors address by designing a synthetic relational prior — a three-stage generative procedure that produces realistic multi-table datasets. Stage 1 generates schema graphs using a LayerDAG autoregressive diffusion model, Stage 2 generates the structural skeleton of inter-table relationships via an attention-based mechanism, and Stage 3 fills in table contents using Gaussian and Softmax heads over an MLP decoder. The model is pretrained on approximately 2 million synthetic tasks and uses a simplified TabPFN architecture augmented with schema attention and a DFS-based graph linearization strategy for flattening multi-table inputs. Experiments on RelBench and 4DBInfer benchmarks show competitive performance against classical baselines, single-table foundation models (TabPFN, TabICL, LimiX), and graph-based methods, while being more parameter-efficient and faster at inference.

**Compliance With Llm Reviewing Policy:**

Affirmed.

**Final Justification:**

My final recommendation is to accept this paper. The rebuttal meaningfully addressed the main concerns from my original review and shifted my assessment upward.

**Soundness.** The rebuttal addressed the three main experimental gaps: stronger tuned baselines (W3), PR-AUC alongside ROC-AUC (W2), and target-task/cross-task finetuning (W4). In each case the qualitative conclusions held, which supports my confidence in the results. The relational prior ablations (Q1-Q3, Q7) confirmed that the generative procedure contributes beyond the tabular pretraining stage. The remaining concern is DFS linearization sensitivity (W1), which I consider minor since the contribution centers on the prior design, not the linearization strategy.

**Originality.** Extending PFNs to relational databases via a synthetic relational prior is a natural but non-trivial task. The three-stage decomposition is well-motivated by the assumptions in Section 4.1, and the universality result via Theorem D.1 provides formal grounding. The rebuttal revealed that some implementation details were inaccurate, but the architectural choices are reasonable and the overall system delivers strong performance.

**Significance.** Foundation models for multi-table settings remain underexplored. RDB-PFN demonstrates that a compact model trained on synthetic data can outperform larger single-table foundation models in the few-shot regime across 19 relational tasks, with notable resource efficiency. The rebuttal's finetuning and cross-task transfer results further add to the practical relevance. The restriction to binary classification limits the scope but is acknowledged as a current limitation, not a fundamental constraint.

**Clarity.** The paper is well-written with a clear progression from problem to assumptions to method to experiments. The description of some implementation details needs updating in the camera-ready. The supplementary materials are thorough and well-organized.

**Impact of the rebuttal.** The rebuttal changed my evaluation. The original submission had notable weaknesses in the experimental protocol. The additional experiments resolved them without altering the qualitative conclusions. The relational prior ablations, which were entirely absent from the original submission, provided the missing direct evidence that the proposed prior design matters. The release of source code and checkpoints in the anonymous repository addressed the reproducibility concern. I raised my score accordingly.

**Key Questions For Authors:**

1. Why was LayerDAG, an autoregressive diffusion model, chosen for schema graph generation in Stage 1 (Section 5.1)? Is there a particular reason why a simpler generative model would not suffice? The motivation appears to be the need to capture the distribution of realistic industrial topologies, but has this been validated against simpler alternatives?

2. Regarding Stage 2 (Section 5.1), did you experiment with alternative mechanisms for connecting child rows to candidate parent rows, beyond the attention-based approach with query and key matrices $W_Q$ and $W_K$? Are these matrices randomly initialized, and if so, from what distribution? Additionally, when you state that "the final connection is sampled from a softmax distribution over compatibility scores", does this mean the connection is stochastic — i.e., any candidate parent can be selected, not just the most probable one?

3. In the same section, you mention modulating "the frequency and amplitude of this feedback" between child and parent tables during content generation. How is this modulation implemented? What specific mechanism controls the strength and frequency of the feedback signal?

4. You also state that "continuous values are generated via Gaussian heads, while categorical values are sampled via Softmax distributions". Does this mean that the decoder MLP predicts the mean and variance of a Gaussian (for continuous features) or logits of a categorical distribution (for discrete features), and exact values are then sampled from these distributions?

5. You describe using "a simplified TabPFN architecture" (Section 5.2), which is supposed to be similar to TabPFNv1 without feature-level attention. However, you also introduce schema attention that operates at the column level. This appears functionally closer to the feature-level attention in TabPFNv2. Could you clarify the relationship between your architecture and the variants of TabPFN?

6. Why did you not initialize from a pretrained checkpoint of a modern single-table foundation model such as TabPFNv2 or LimiX, rather than training a new backbone from scratch? This would potentially eliminate the need for the tabular warm-up pretraining stage (Section 5.3, Stage 1). Additionally, what are the "synthetic single-table datasets" used for this warm-up, and how are they constructed?

7. To the best of my knowledge, RelBench datasets contain a temporal component, with predictions required for future time points based on historical data. Does the relational synthetic prior incorporate temporal signals? If so, how?

8. Why is the evaluation conducted in a few-shot regime rather than using standard predefined data splits with more conventional split ratios (e.g., 60/20/20 or 10/10/80)?

9. Can you explain the main source of the inference speed-up reported in Section 6.2 and Figure 2? What architectural or design choices contribute most to the reduced inference latency compared to other foundation models?

10. How is the normalized score defined in the performance comparisons (e.g., Figures 3 and 4)? Specifically, for each dataset, are individual method scores linearly normalized to the $[0, 1]$ range between the worst and best scores achieved on that task?

11. In the single-table benchmark (Figure 4), there appear to be both "single-table" and "full" versions of RDB-PFN. If the "single-table" version uses only the target table without DFS compression, and the "full" version includes DFS aggregation, what accounts for the difference between these two versions on single-table datasets that contain only one table and therefore do not require DFS?

I am willing to increase my score if the authors address the questions and concerns raised above.

**Limitations:**

The paper does not include an explicit limitations section. The most significant unacknowledged limitations include: the restriction to binary classification tasks, the absence of finetuning evaluation, the lack of ablation over the DFS linearization strategy, and the missing analysis of the sufficient amount of pretraining data. The absence of source code and pretrained checkpoints further limits the reproducibility and practical impact of the work. Societal impact is not a concern for this work.

**Strengths And Weaknesses:**

## Strengths

1. **Clear introduction and problem statement.** The introduction (Section 1) immediately establishes the problem (the lack of pretraining data for relational deep learning) and explains why extending PFNs to multi-table settings is a meaningful direction. The progression from the foundation model discrepancy to the gap in relational synthetic prior and the proposed approach is logical and easy to follow.

2. **Notable contribution in the scope of foundation models.** The core idea of generalizing PFNs from single-table to relational databases is a natural and practically important extension. The observation that designing a relational synthetic prior is the key missing ingredient (as opposed to architectural novelty) is well-argued and positions the contribution clearly within the existing PFN literature.

3. **Properly motivated relational assumptions.** The assumptions in Section 4.1 (Assumptions 4.1-4.3) are meaningful and grounded in practical observations about real-world relational databases. They provide a principled foundation for the generative procedure design and are supported by concrete examples.

4. **Resource efficiency analysis.** The inclusion of inference time comparisons and parameter counts (Section 6.2, Figure 3) is valuable. It is informative to see that RDB-PFN achieves competitive performance while being substantially lighter and faster than other tabular foundation models such as TabPFN, TabICL, and LimiX.

5. **Informative ablation of the relational prior.** The analysis of the linearized relational prior in Section 6.4 (Figure 5) helps confirm that the relational component of the synthetic prior is a significant contributor to performance, and the reported gains can not be attributed solely to the tabular pretraining stage or architectural choices.

## Weaknesses

1. **DFS linearization is not ablated and may unfairly disadvantage baselines.** The DFS-based graph linearization (Section 5.2) is presented as the sole method for flattening multi-table RDBs into a single sequence, without comparison to alternative linearization strategies. This is a critical preprocessing step that directly affects what information is available to the model, yet it is not ablated. Additionally, the associated steps of feature standardization, over-generation, and undersampling described in Appendix C.2 introduce further design choices that are not systematically evaluated. For the single-table foundation model baselines, which were not designed to consume DFS-linearized inputs, this preprocessing may introduce an unfair disadvantage, making it difficult to isolate the contribution of the relational prior from the effects of the data conversion pipeline.

2. **Restrictive task formulation and flawed evaluation metric.** The standardization of all tasks to binary classification (Appendix A.2) is a notable limitation that narrows the scope of the evaluation. Furthermore, the paper's claim that "ROC-AUC is threshold-independent and robust to significant class imbalance" is not accurate — under severe imbalance, a model can achieve high ROC-AUC by correctly ranking easy majority-class examples yet still failing on the minority class that matters most. Metrics such as PR-AUC would be more informative for imbalanced tasks like fraud detection.

3. **No hyperparameter tuning for classical baselines.** Appendix B.1 reports a single fixed configuration for each classical baseline — XGBoost is run with one hand-picked set of hyperparameters. No tuning search space or tuning procedure is described, suggesting that no hyperparameter optimization was performed at all. It is well-known that GBDTs are highly sensitive to hyperparameter tuning and can achieve substantially different performance with an appropriate tuning budget. Without proper tuning, the reported gaps between RDB-PFN and classical baselines may be partly attributable to suboptimal baseline configurations rather than a true advantage of the proposed method.

4. **No investigation of finetuning.** The paper evaluates all foundation models in a purely in-context learning setting without investigating finetuning on downstream tasks (Section 6.1). It is known to provide significant performance gains for tabular foundation models such as TabPFNv2, and its omission makes the comparison incomplete. It is possible that the relative ranking of methods would change under a finetuning protocol.

5. **No source code provided.** The absence of source code and pretrained checkpoints is a significant limitation. For foundation models, which are primarily useful when practitioners can readily apply them to new tasks, open and accessible implementations are particularly important. Otherwise, the results are not independently reproducible.

---

> ### Author Rebuttal · Authors · 2026-03-31
>
> Thank you for the detailed and constructive review. We added new experiments and clarifications, detailed results and materials are in the link [https://anonymous.4open.science/r/RDBPFN_rebuttal/Results_README.md].
>
> Q1–Q3, Q7 (schema, link, update, temporal component):
> We now add a lightweight playground ablation study. We compare the original LayerDAG generator with a simpler DAG generator, and the original edge-generation mechanism with simpler alternatives like predetermined fixed-probability variants. Additional ablations are reported in the link. Across these comparisons, the full relational prior retains a slight but consistent advantage over the simplified variants. We also clarify the implementation details: the edge-generation module contains variants including attention-based scoring (with matrices using default Kaiming-uniform initialization) and concatenation followed by MLP scoring; the signal update is applied to the selected parent embeddings. Temporal information is incorporated both in timestamp generation and latent initialization, and we additionally include an ablation with this component removed.
>
> Q4–Q6 (decoder, backbone, pretrained TFM):
> We apologize that the decoder explain is inaccurate. In the final implementation, we do not use the complex decoder in the text; instead, the SCM/MLP predicts a scalar latent per cell, which is then converted into a continuous or categorical value through simple post-processing (e.g., clipping / normalization / binning). We also revise the “simplified TabPFN”: the backbone includes interleaved cross-row and cross-column attention, and is closer to a lightweight TabPFNv2-style design. Finally, the single-table generation is inherited from TabICL’s pipeline, but with different activations and hyperparameters controlling the complexity. We train from scratch because our main goal is to isolate the effect of the relational synthetic prior; continued pretraining of third-party TFMs, whose own pretraining pipelines may be only partially revealed, would introduce an additional confound.
>
> Q8, W4 (few-shot regime and fine-tuning):
> The strict few-shot setting is an intentional scientific target: the paper studies rapid adaptation to a new RDB under limited labels and minimal graph-model engineering. At the same time, we agree that complementary adaptation settings are useful, so we additionally include (i) **larger-context ensembling**, (ii) **target-task fine-tuning**, which both further improves performance, and (iii) **cross-task fine-tuning on other RDB tasks**, which yields a modest additional improvement, suggesting partial transferability across real relational tasks.
>
> Q9–Q11 (speedup, definition, “single-table” vs “full”):
> The main source of the speed-up is the absence of expensive default ensembling used by some tabular FMs, with the smaller backbone contributing an additional but smaller factor. Regarding the normalized score, your understanding is correct, and we now make the definition explicit. Finally, the “single-table” and “full” variants use the same test-time interface and backbone; they differ only in pretraining curriculum (single-table-only vs single-table+relational synthetic pretraining), so on a true single-table benchmark the comparison is about transfer from relational pretraining.
>
> W1 (DFS linearization / preprocessing fairness):
> DFS is used here as the shared relational-to-tabular interface needed to make raw RDB inputs accessible to single-table methods at all; after these DFS features are computed, each single-table baseline still uses its own downstream preprocessing pipeline. We do not claim that DFS is the only possible interface, but in the benchmark setting used here it is the most practical standardized way to compare single-table methods on relational data. The sampling strategy in Appendix C.2 is mainly an engineering standardization choice rather than a claim about an optimal DFS configuration: because DFS produces variable numbers of features while our PFN-style training requires a fixed input shape for efficiency, we first computing a larger pool of DFS features (different configs for different DFS hops) and then sampling fixed columns for each training instance. This also interacts with task augmentation, allowing broader reuse of the generated data with different sampled columns and tasks under a fixed data generation budget.
>
> W2, W3, W5 (scope, metrics, stronger baselines, reproducibility):
> The current evaluation is binary-focused because the benchmark composition is overwhelmingly binary; we clarify that this is indeed an limitation but a head choice rather than a pipeline limitation, and include a small multiclass feasibility extension using one-vs-rest. We add PR-AUC alongside ROC-AUC and observe that the comparison remains similar. We also strengthen the classical baselines by adding tuning and a much larger-performance preset for AutoGluon. We additionally provide supporting implementation materials in the link.

---

> > ### Author Rebuttal · Reviewer_5ThT · 2026-04-03
> >
> > I have carefully studied the supplementary rebuttal materials and thank the authors for the substantial additional experiments and clarifications provided during the discussion period.
> >
> > **Addressed concerns.** Most of the weaknesses and questions raised in my original review have been adequately addressed:
> >
> > - W2 (metrics) and W3 (baselines): Stronger classical baselines with hyperparameter tuning and a higher performing AutoGluon preset (Figure 1) together with PR-AUC results (Figure 2) are commendable. The qualitative conclusions remain consistent under both changes, which strengthens the evaluation.
> > - W4 (finetuning): The investigation of target-task finetuning and cross-task adaptation (Figure 3) addresses one of the key missing pieces in the original evaluation. The observation that cross-task finetuning yields modest additional improvements is interesting and suggests partial transferability across relational tasks.
> > - Q1-Q3, Q7 (relational prior ablation): The playground ablation study (Figure 6) comparing the full relational prior against simplified variants (simpler DAG generator, alternative edge-generation mechanisms, temporal component removal) provides useful evidence that the proposed prior design contributes meaningfully to performance. I also appreciate the clarifications on initialization, stochastic connection sampling, the feedback modulation mechanism, and the incorporation of temporal signals.
> > - Q4-Q6 (architecture, training). The correction that the actual decoder is simpler than originally described in the text (scalar latent per cell with post-processing) is noted. The revised description of the backbone as interleaved cross-row and cross-column attention, closer to a TabPFNv2 design, resolves my earlier confusion. The rationale for training from scratch to isolate the effect of the relational prior is reasonable.
> > - Q8-Q11 (evaluation details). The clarifications on the few-shot regime as an intentional scientific target, the inference speedup being primarily due to the absence of default ensembling, the normalized score definition, and the single-table vs full version distinction (differing only in pretraining schedule) are all satisfactory.
> >
> > **Requests for the revised paper.** I ask the authors to address the following in the camera-ready version:
> >
> > - W1 (DFS linearization): The argument that DFS is a practical shared interface is reasonable. It would still be useful to see at least a brief discussion of how sensitive the results are to the specific DFS configuration (e.g., aggregation choices, hop depth), even if a full ablation is not feasible.
> > - W5 (reproducibility): I note that the anonymous repository now contains source code and pretrained checkpoints, which addresses the reproducibility concern from the original review. I encourage the authors to maintain a public release upon acceptance.
> > - Q4, Q5 (implementation details): The revised descriptions of the decoder and the backbone architecture should be reflected in the paper.
> > - New results: The updated baselines (Figure 1), the reported PR-AUC values (Figure 2), finetuning results (Figure 3), and relational prior ablations (Figure 6) from the supplementary materials are important for the completeness of the evaluation. I ask the authors to incorporate these into the main paper or appendix.
> >
> > **Overall assessment.** The additional experiments, particularly the relational prior ablations, the finetuning investigation, and the stronger baselines, notably strengthen the evaluation. The core contribution of designing a synthetic relational prior for extending PFNs to relational databases is clearly motivated and practically relevant. I increase my score accordingly.

---

> > > ### Author Response · Authors · 2026-04-07
> > >
> > > Thank you very much for the careful follow-up and for studying the supplementary materials in such detail. We are very glad that the additional experiments and clarifications addressed most of your original concerns.
> > >
> > > We will make sure that the key new materials you highlighted are incorporated into the revised paper and appendix. We will also maintain a public code release for reproducibility and broader use. We appreciate your suggestion on briefly discussing DFS sensitivity as well. We have already run a lightweight DFS-sensitivity check focusing on hop depth: in a simplified setting using only 1-hop DFS features (and therefore less relational information than the default 2-hop configuration), performance drops clearly relative to the richer setup, suggesting that the model does benefit from the additional relational information exposed by the default DFS configuration. One caveat is that the strength of this effect is task-dependent and may vary with dataset properties. We will incorporate this discussion and the corresponding results into the revision, and we will also extend this analysis to additional DFS hop settings, more datasets, and more aggregation functions.
> > >
> > > ---
> > >
> > > Table: Some small datasets' Performance of RDBPFN with context size 1024.
> > > |       | diginetica/ctr | oubrain-small/ctr | rel-f1/driver-dnf | rel-f1/driver-top3 | rel-hm/user-churn | stackexhange/churn | stackexchange/upvote |
> > > | ----- | -------------- | ----------------- | ----------------- | ------------------ | ----------------- | ------------------ | -------------------- |
> > > | DFS-1 | 0.5225         | 0.4914            | 0.4955            | 0.5445             | 0.5294            | 0.5592             | 0.8555               |
> > > | DFS-2 | 0.7004         | 0.5352            | 0.7188            | 0.8115             | 0.6648            | 0.8477             | 0.8527               |

---

### Official Review · Reviewer_ksBP · 2026-03-13

**Soundness:** 3
**Presentation:** 3
**Significance:** 2
**Originality:** 3
**Overall Recommendation:** 4
**Confidence:** 4

**Summary:**

While foundational models for vision and text are widely successful, we still lack a competitive foundational model for relational databases. This is largely due to the lack of massive scale datasets with diverse enough samples for generalization and assumptions of independence that do not apply to relational databases. To address these limitations, the authors introduce Relational Data Bases Prior-Data Fitted Networks (RDB-PFN). The model is trained with synthetic data while providing a relational prior that allows to generate diverse data while preserving the causal relationships between tables. The backbone utilizes a graph neural network with a stacked linearization layer, which is able to adapt via in-context learning. The model was validated across 19 datasets in binary classification tasks and compared their results to state-of-the-art models. The results show competitive performance while improving inference times and data training efficiency.

**Compliance With Llm Reviewing Policy:**

Affirmed.

**Final Justification:**

The new results substantially address my initial concerns, and I will adjust the score accordingly. The extensions to other tasks, the additional ablation studies, and stress tests support the claims and validate the work presented.

**Key Questions For Authors:**

There are scale constraints associated with in-context learning, a question worth answering is how to overcome this limitation. For example, did you explore dynamic context sizing or hierarchical sampling? If not, is there performance degradation for larger datasets. Separately, did you explore alternative model architectures? How does performance change with model size and our linear projection? While the authors state that tasks other than binary classification are left for future work, can the current design extend to other tasks?

**Limitations:**

Limitations on dataset scale and limitation to binary classification tasks should be highlighted as a limitation of the work, since they defy the claim of a true foundation model.

**Strengths And Weaknesses:**

## Strengths
The manuscript is well-presented and technically sound. This work introduces a new paradigm for relational, tabular data generation towards building a foundational model for relational databases. Evaluation across 19 binary classification tasks showed performance gains of 2-3% compared to alternative models in relational databases, with few-shot examples, and lower but competitive performance in single-table w.r.t. foundational models for tabular data; but allowing for faster inference.
## Weaknesses
The performance shown in relational databases is compelling but fails in single-table settings, raising questions about the capabilities of the presented model as a true foundation model for tabular data. Additionally, this performance is only reported for binary classification tasks, limiting the application to multi-task and regression tasks. Moreover, while the data generation and model assumptions are clearly motivated, ablation studies are needed to understand the need for each individual component and illustrate the associated caveats of each assumption. For example, the linear projection could result in information loss. Moreover, few-shot in context learning is needed for performance setting a clear limitation on scale that is not fully described. While the model size and fast inference speeds are highlighted as a benefit, the tradeoffs on generalization and ability to learn complex relationships is not fully described. If so, comparison to alignment but inexpensive solutions like multiset CCA would be relevant to understand the impact of the proposed solution.

---

> ### Author Rebuttal · Authors · 2026-03-31
>
> Thank you for the thoughtful review and for recognizing the technical soundness of the paper. New results are placed in [https://anonymous.4open.science/r/RDBPFN_rebuttal/Results_README.md].
>
> On your concern that the model “fails in single-table settings,” raising questions about it as “a true foundation model for tabular data”:
>
> We would like to clarify that the single-table setting is not the primary target of the paper. Our claim is not a universal foundation model for all tabular settings, but rather a **foundation model for relational databases**, where cross-table interactions absent from standard single-table priors. Under that scope, we view the single-table result as a transfer check rather than a failure case. Our single-table-only variant serves as a lightweight baseline with much simpler architecture, preprocessing, and pretraining than specialized single-table TFMs, so it is expected to trail them in their home regime. The more important observation is that additional relational pretraining on linearized RDBs improves not only the target RDB setting, but also our own single-table-only variant on single-table tasks, suggesting the relational prior’s benefits.
>
> On your point that the evaluation is “only reported for binary classification tasks,” and whether “the current design extends to other tasks”:
>
> The current checkpoint is binary mainly because the benchmark suite is overwhelmingly binary (all classification tasks in RelBench and all but one in 4DBInfer). We now state this explicitly as a limitation. At the same time, this is primarily a **task-head** choice rather than a limitation of the relational prior or pipeline. More broadly, tabular foundation models often use a shared classification head for all classification tasks (e.g., a fixed 10-way output head that naturally covers tasks with up to 10 classes), rather than changing the core pipeline itself. Our current setup follows the same philosophy, although we instantiate only the binary case. As a feasibility check, we also include a small multiclass extension using one-vs-rest, which already handles a multiclass task without retraining the full model.
>
> On your point that “ablation studies are needed,” including the concern that “the linear projection could result in information loss,” and your related questions about model size and architecture:
>
> To address this, we build a lightweight playground that allows us to retrain smaller ablated versions efficiently. We add ablations over **data generation** (simplifying individual stages of the relational prior) and **model design** (varying model size and lightweight architecture variants), to separate the contributions of the relational prior, backbone design, and model capacity. We would also like to clarify that possible information loss from linearization is not unique to our method: single-table methods cannot consume raw RDB inputs directly, so some relational-to-tabular interface is necessary; following prior relational benchmarks, we use DFS as the standard practical bridge that keeps the comparison close to the tabular FM regime.
>
> On your concern that “few-shot in-context learning is needed for performance,” creating “a clear limitation on scale that is not fully described”:
>
> Few-shot ICL is an intentional scientific setting: the goal is rapid adaptation to a new relational database under limited downstream labels and minimal task-specific engineering, rather than full-data supervised optimization. At the same time, bounded context is a limitation of ICL in general. We therefore add a **larger-context scaling study** using chunked-support ensembling, which approximates much larger effective support sizes and tests whether performance continues improving as support grows. We also report a fine-tuning result as a complementary adaptation setting when more training samples are available.
>
> On your concern that “the tradeoffs on generalization and ability to learn complex relationships”:
>
> Regarding both the effect of RDB-oriented pretraining on single-table tasks and the role of model design, the current results already partially answer both questions: on single-table tasks, relational pretraining improves over our own single-table-only baseline, while on relational benchmarks the model remains strongly competitive despite its much smaller size. We additionally include reduced-size and architecture ablations, and observe that reducing the model size still yields a similar performance trend, while switching to a weaker and harder-to-train architecture design (e.g., replacing interleaved row/column attention with a blockwise row-first/column-second scheme) causes a substantial drop on RDB tasks. These results suggest that, although a tradeoff is possible in principle, we do not observe a clear contradiction in the current model design where improving relational performance comes at the expense of single-table generalization.

---

> > ### Author Rebuttal · Reviewer_ksBP · 2026-04-02
> >
> > I thank the authors for their detailed response and additional experiments. The new results substantially address my initial concerns, and I will adjust the score accordingly. The extensions to other tasks, the additional ablation studies, and stress tests support the claims and validate the work presented.

---

> > > ### Author Response · Authors · 2026-04-07
> > >
> > > Thank you very much for the careful follow-up. We are very glad that the additional experiments and clarifications addressed your concerns.
> > >
> > > We also noticed that your comment mentions adjusting the score accordingly; if that was your intention, we would be grateful if you could please check whether the updated score was successfully recorded in the review system.

---

### Decision · Program_Chairs · 2026-04-30

**Decision:**

Accept (regular)

**Comment:**

The proposed method provides a way to generate synthetic data for training foundation models for relational databases.  Reviewers appreciated the method and its presentation.  Several reviewers appreciated the novelty (and practicality) of trying to create synthetic data for training relational foundation models, since detailed multi-table relational DBs are often proprietary.  The authors ran a number of additional experiments (which will hopefully be added to the paper) to provide more experimental evaluation.